# DIFFERENTIALLY PRIVATE TWO-STAGE GRADIENT DESCENT FOR INSTRUMENTAL VARIABLE REGRESSION

**Haodong Liang, Yanhao Jin, Krishnakumar Balasubramanian, Lifeng Lai**
University of California, Davis
Davis, CA 95616, USA
`{hdliang,yahjin,kbala,lflai}@ucdavis.edu`

## ABSTRACT

We study *instrumental variable regression* (IVaR) under *differential privacy* constraints. Classical IVaR methods (like two-stage least squares regression) rely on solving moment equations that directly use sensitive covariates and instruments, creating significant risks of privacy leakage and posing challenges in designing algorithms that are both statistically efficient and differentially private. We propose a *noisy two-stage gradient descent* algorithm that ensures $\rho$-zero-concentrated differential privacy by injecting carefully calibrated noise into the gradient updates. Our analysis establishes finite-sample convergence rates for the proposed method, showing that the algorithm achieves consistency while preserving privacy. In particular, we derive precise bounds quantifying the trade-off among optimization, privacy, and sampling error. To the best of our knowledge, this is the first work to provide both privacy guarantees and provable convergence rates for instrumental variable regression in linear models. We further validate our theoretical findings with experiments on both synthetic and real datasets, demonstrating that our method offers practical accuracy-privacy trade-offs.

## 1 INTRODUCTION

Instrumental variable regression (IVaR) is a foundational tool in causal inference, designed to recover structural parameters when standard estimators fail due to endogeneity. In many observational settings, covariates are influenced by unobserved confounders, causing naive methods (such as the ordinary least squares (OLS) in the context of linear regression) to produce biased and inconsistent estimates. IVaR circumvents this by leveraging *instruments*, which are variables that are predictive of the endogenous regressors but independent of hidden confounders, to enable consistent estimation of causal effects (Hausman, 2001; Wooldridge, 2010; Angrist & Krueger, 2001). This perspective is increasingly important in machine learning, for example in recommendation systems where user exposure is confounded by prior preferences (Si et al., 2022), or in reinforcement learning where actions and rewards are jointly influenced by unobserved context (Xu et al., 2023). In such settings, IVaR provides a principled way to disentangle causal effects from spurious correlations, enabling more reliable decision making.

However, many applications of IVaR involve sensitive data, such as individual health records, financial transactions, or user interactions, where protecting privacy is of paramount importance. In such settings, releasing model estimates or even intermediate statistics can leak information about individuals in the dataset. Differential privacy (DP) (Dwork et al., 2006) provides a mathematically rigorous framework to ensure that an algorithm's output does not reveal sensitive information about any single data point. Despite the importance of IVaR in causal inference, to the best of our knowledge, there are *no prior works* addressing the problem of performing IVaR under differential privacy. This gap motivates the central question of this paper:

*Can we design differentially private algorithms for instrumental variable models*
*that achieve statistically efficient convergence rates?*

Our work focuses on answering this question in the context of linear IVaR models. To situate our contributions, we briefly review existing work on DP methods for OLS regression, with additional discussion in Section 1.1. Several predominant approaches have emerged in the literature: (i) perturbion methods, where the empirical covariance and cross-covariance matrices are privatized before solving the normal equations; (ii) consensus-based methods, including propose-test-release and exponential mechanism approaches, which directly privatize the estimator through carefully designed randomized output rules; and (iii) gradient perturbation methods, where iterative optimization algorithms are made private by clipping gradients and injecting calibrated Gaussian noise. While all three approaches ensure differential privacy, gradient perturbation combined with clipping has been shown to yield the sharpest statistical rates in OLS regression, particularly in high-dimensional and finite-sample regimes (Bassily et al., 2014; Brown et al., 2024a).

Given the centrality of IVaR in causal inference, it is natural to explore whether the aforementioned techniques can be adapted to this setting. Unlike OLS, however, IVaR is based on moment conditions involving both covariates and instruments, making it less straightforward to design private algorithms. In particular, sufficient-statistics perturbation and consensus-based methods have not been explored, and their adaptation is non-trivial due to the inherent ill-posedness of IVaR under weak instruments and the sensitivity of the moment equations. These challenges stem from the fact that the closed-form 2SLS estimator depends on nested matrix multiplications and inversions, whose sensitivities are difficult to characterize directly. To address this, we design a two-stage gradient-descent-based algorithm that enables injecting noise at the iteration level in a principled manner. This structure allows us to rigorously control sensitivity, calibrate the noise, and derive non-asymptotic utility guarantees. Specifically, we make the following **contributions** in this work:

- We introduce `DP-2S-GD` (Algorithm 1), the first differentially private algorithm for instrumental variable regression, built on noisy gradient descent with gradient clipping.
- We establish finite-sample non-asymptotic error bound for `DP-2S-GD` (Theorem 3.1), explicitly characterizing the trade-off between optimization, privacy, and sampling error. The main technical challenge is to carefully control the interaction between privacy-induced noise and the contraction of the gradient dynamics across iterations, with the privacy guarantee ensured by Proposition 3.1.
- We validate our theoretical analysis with experiments on synthetic and real-world datasets, demonstrating practical accuracy-privacy trade-offs (Section 4).

By developing differentially private algorithm for IVaR and establishing its theoretical bound, we enable researchers to perform robust causal analyses without compromising the confidentiality of individuals in the dataset.

## 1.1 RELATED WORK

**Differential Privacy for Regression.** One can group private regression methods into the following broad families. (1) Output/objective perturbation (private empirical risk minimization (ERM)): add noise to the final estimator (output perturbation) or inject a random linear/quadratic term into a strongly convex loss before optimizing (objective perturbation); these one-shot mechanisms give $(\varepsilon, \delta)$-DP guarantees and excess-risk bounds for convex ERM (Chaudhuri et al. (2011); Kifer et al. (2012); Bassily et al. (2014)). Recent refinements, e.g. Redberg et al. (2023), leverage subsampling and tighter accounting to improve accuracy. (2) Sufficient-statistics (matrix) perturbation: release noisy surrogates of $(\mathbf{X}^\top \mathbf{X}, \mathbf{X}^\top \mathbf{y})$ (or related second-moment structures) and then solve the (regularized) normal equations; this route enables OLS-specific inference but can suffer under ill-conditioning because noise is injected at the Gram-matrix level (Dwork et al. (2014); Sheffet (2017)). Tsfadia et al. (2022) proposes a subsample-and-aggregate framework that can, in principle, be adapted to regression settings by privately estimating the relevant sufficient statistics on carefully selected data subsets. Further developments in this direction include Bernstein & Sheldon (2019) and Ferrando & Sheldon (2024). (3) Exponential mechanism: privately selects an output by randomly choosing among candidates with probabilities that grow exponentially with their quality score, with parameters controlling how strongly it favors the higher-scoring options. This mechanism is frequently applied in constructing algorithm to privately select a regression model from a pool of non-private OLS fits on subsets of the data (Ramsay & Chenouri (2021), Cumings-Menon (2022), Amin et al. (2022)). (4) Gradient perturbation (DP-(S)GD): clip per-example (mini-batch or full) gradients and add Gaussian noise at each step, tracking privacy with bounded log moment generating function of privacy loss random variable Wang et al. (2019), Rényi DP, and subsampled-

RDP-which yields tight composition for many small releases and scales well to large $n, p$ without forming $\mathbf{X}^\top \mathbf{X}$. (Abadi et al. (2016); Bun & Steinke (2016); Mironov (2017); Wang et al. (2019)).

We favor gradient perturbation for multi-stage estimators like IVaR because it (i) composes tightly across many noisy steps using modern privacy accountants, (ii) avoids spectrum-dependent blow-ups from noising $\mathbf{X}^\top \mathbf{X}$ (Sheffet (2017)) and (iii) yields strong convergence rates while fitting standard training pipelines (including using minibatches, streaming, early stopping) and enabling modular, stage-wise design, which is preferable for practice (Bassily et al. (2014), Abadi et al. (2016)). Although there exists DP techniques for estimating gram matrices that avoid blow-ups, e.g., via carefully calibrated noise or regularization (Brown et al. (2023), Kamath et al. (2019)), the purely sufficient statistics-based pipelines require larger sample sizes (polynomial to condition number) than gradient-based approaches to reach a comparable accuracy in high dimensions (Brown et al., 2024b). That said, we note that the convergence of private first-order gradient methods still depend heavily on the condition number Varshney et al. (2022); Liu et al. (2023).

**Instrumental Variable Regression (IVaR)** has been extensively studied in econometrics (Angrist & Krueger, 2001; Angrist & Pischke, 2009). Classical methods such as two-stage least squares (2SLS) admit closed-form solutions but face limitations in modern applications: they do not scale well to high-dimensional or streaming data, cannot easily incorporate regularization, and are restricted to linear models. This has motivated optimization-based approaches, including convex–concave formulations of nonlinear IV (Muandet et al., 2020), stochastic optimization methods for scalable and online estimation (Della Vecchia & Basu, 2023; Chen et al., 2024; Fonseca et al., 2024), and bi-level gradient descent algorithms with convergence guarantees (Liang et al., 2025). Extensions to nonlinear IV include kernel-based methods (Singh et al., 2019) and DeepIV (Hartford et al., 2017). Despite these advances, prior work assumes unrestricted access to the data and does not provide end-to-end differential privacy guarantees, which are increasingly critical in sensitive domains such as healthcare, finance, and online platforms. To our knowledge, no existing method offers DP guarantees with finite-sample convergence rates for linear IV/2SLS that explicitly account for instrument strength, sample size, dimension, and iteration complexity.

*Notations:* Throughout this paper, unless otherwise specified, we use lower-case letters to denote random variable or individual data samples, and upper-case letters to denote datasets, i.e. collections of samples. Bolded letters represent vectors and matrices, whereas unbolded letters represent scalars.

## 2 PRELIMINARIES

### 2.1 PRIVACY NOTIONS

We first review widely used notions of privacy in the literature. Two datasets $D$ and $D'$ are said to be *neighbors* if they differ in exactly one entry. The concept of neighboring datasets allows us to formally quantify the level of differential privacy. The two most common notions are $(\varepsilon, \delta)$-differential privacy and zero-concentrated differential privacy (zCDP).

**Definition 2.1** (($\varepsilon, \delta$)-Differential Privacy (Dwork et al., 2006))**.** A randomized mechanism $M$ satisfies $(\varepsilon, \delta)$-differential privacy if for all neighboring datasets $D, D'$ and all measurable sets $S$, we have $\Pr[M(D) \in S] \le e^\varepsilon \Pr[M(D') \in S] + \delta$. Here $\varepsilon \ge 0$ controls the multiplicative privacy loss, while $\delta \in [0, 1]$ allows for a small probability of arbitrary deviation.

**Definition 2.2** (Zero-Concentrated Differential Privacy (zCDP) (Dwork & Rothblum, 2016; Bun & Steinke, 2016))**.** A randomized mechanism $M$ satisfies $\rho$-zero-concentrated differential privacy ($\rho$-zCDP) if for all neighboring datasets $D, D'$ and all $\alpha > 1$, we have the $D_\alpha(M(D) \| M(D')) \le \rho\alpha$, where $D_\alpha(P\|Q)$ denotes the Rényi divergence (see Appendix A for the definition) of order $\alpha$ between distributions $P$ and $Q$.

While $(\varepsilon, \delta)$-DP is the most widely used notion of privacy, it can be too coarse for analyzing iterative mechanisms, as composition accumulates $\varepsilon$ and $\delta$ linearly[1]. In contrast, zero-concentrated differential privacy (zCDP) characterizes privacy loss through Rényi divergences, which ensures that the privacy loss random variable enjoys a sub-Gaussian concentration property. This yields two key benefits: (i) *tighter composition*, since zCDP parameters add under composition, and (ii) *smooth*

---

[1] $(\epsilon, \delta)$-DP also satisfies an advanced composition that asymptotically matches zCDP, but the composition is more cumbersome and typically less practical than the clean additive composition offered by zCDP.

*conversion*, since $\rho$-zCDP implies $(\varepsilon, \delta)$-DP with $\varepsilon = \rho + 2\sqrt{\rho \log(1/\delta)}$; see Bun & Steinke (2016, Proposition 1.3). As a result, we choose zCDP for technical convenience since it provides simple additive composition rule and leaner formulas in our context where we compose a large number of identical Gaussian mechanisms across both stages in 2SLS algorithm.

## 2.2 IVaR Model and Assumptions

Endogeneity is a central challenge in linear regression. Suppose we aim to estimate the causal effect of the regressor $\mathbf{x} \in \mathbb{R}^p$ on the outcome $y \in \mathbb{R}$. However, there exists an unobserved confounder $\mathbf{u}$ that affects both $\mathbf{x}$ and $y$, thereby violating the standard exogeneity assumption that $\mathbf{x}$ is uncorrelated with the noise. As a result, the OLS estimator becomes biased and inconsistent. Instrumental variable regression (IVaR) is a widely adopted method to handle endogeneity by including $\mathbf{z} \in \mathbb{R}^q$, an instrumental variable (IV), to the model (Angrist & Krueger, 2001):

$$y = \boldsymbol{\beta}^\top \mathbf{x} + \epsilon_1, \qquad \mathbf{x} = \boldsymbol{\Theta}^\top \mathbf{z} + \boldsymbol{\epsilon}_2, \tag{1}$$

where the error terms $\epsilon_1$ and $\boldsymbol{\epsilon}_2$ are correlated due to the common confounder $\boldsymbol{u}$; see Figure 1 for an illustration. Given the dataset $(\mathbf{Z}, \mathbf{X}, \mathbf{Y}) = \{(\mathbf{z}_i, \mathbf{x}_i, y_i)\}_{i=1}^n$ [2], the objective of the IVaR model is to solve the following bi-level optimization problem:

$$\hat{\boldsymbol{\beta}} = \arg\min_{\boldsymbol{\beta} \in \mathbb{R}^p} \left\{ \mathcal{L}(\boldsymbol{\beta}) = \frac{1}{n} \sum_{i=1}^n \left( y_i - \boldsymbol{\beta}^\top \hat{\boldsymbol{\Theta}}^\top \mathbf{z}_i \right)^2 \right\}, \text{s.t. } \hat{\boldsymbol{\Theta}} = \arg\min_{\boldsymbol{\Theta} \in \mathbb{R}^{q \times p}} \left\{ \frac{1}{n} \sum_{j=1}^n \|\mathbf{x}_j - \boldsymbol{\Theta}^\top \mathbf{z}_j\|^2 \right\}. \tag{2}$$

Optimization problem 2 admits a closed-form solution. A classical approach to solve equation 2 is the two-stage least squares (2SLS) estimator; see Definition 2.3.

**Definition 2.3** (2SLS estimator). Given observational data $(\mathbf{Z}, \mathbf{X}, \mathbf{Y}) = \{(\mathbf{z}_i, \mathbf{x}_i, y_i)\}_{i=1}^n$, the 2SLS estimator $\hat{\boldsymbol{\beta}}_{\text{2SLS}}$ is obtained through two consecutive OLS regressions:

    i. *First stage*: Regress $\mathbf{X}$ on $\mathbf{Z}$ to obtain $\hat{\boldsymbol{\Theta}}$

$$\hat{\boldsymbol{\Theta}} = (\mathbf{Z}^\top \mathbf{Z})^{-1} \mathbf{Z}^\top \mathbf{X}.$$

    ii. *Second stage*: Regress $\mathbf{Y}$ on $\hat{\mathbf{X}} := \mathbf{Z}\hat{\boldsymbol{\Theta}}$ to obtain $\hat{\boldsymbol{\beta}}_{\text{2SLS}}$:

$$\hat{\boldsymbol{\beta}}_{\text{2SLS}} = (\hat{\boldsymbol{\Theta}}^\top \mathbf{Z}^\top \mathbf{Z} \hat{\boldsymbol{\Theta}})^{-1} \hat{\boldsymbol{\Theta}}^\top \mathbf{Z}^\top \mathbf{Y}.$$

In the following sections, we will use $\hat{\boldsymbol{\beta}}$ to denote the 2SLS estimator for simplicity. We impose the following standard assumptions for IVaR model.

**Assumption 1** (IVaR Assumptions). A random variable $\mathbf{z} \in \mathbb{R}^q$ is a valid IV, if it satisfies:

    (i) Fully identification: $q \geq p$ (without loss of generality, we assume data $\mathbf{Z}, \mathbf{X}$ are full rank).
    (ii) Correlation to $\boldsymbol{x}$: $\mathsf{Corr}(\mathbf{z}, \mathbf{x}) \neq \mathbf{0}$.
    (iii) Exclusion to $y$: $\mathsf{Corr}(\mathbf{z}, \epsilon_1) = \mathbf{0}$.

In Assumption 1, condition (i) ensures the existence of the unique solution $\hat{\boldsymbol{\beta}}$ in equation 2, condition (ii) guarantees that the instrument explains nontrivial variation in the endogenous regressor $\mathbf{x}$, and condition (iii) ensures that the instrument affects the outcome $y$ only through $\mathbf{x}$. These conditions are crucial for eliminating endogeneity and achieving consistent estimation for $\boldsymbol{\beta}$. See Stock & Watson (2011, Chapter 12) for a detailed discussion. We further impose the following assumptions to establish non-asymptotic rates.

**Assumption 2.** We assume the following conditions hold:

    (i) $\mathbf{z}$ is a mean-zero isotropic sub-Gaussian random vector. That is, $\mathbb{E}[\mathbf{z}] = \mathbf{0}$, $\mathbb{E}[\mathbf{z}\mathbf{z}^\top] = \mathbf{I}_q$, and for some $\sigma_z > 0$, $\mathbb{E}[e^{u\langle \mathbf{z}_i, \mathbf{v} \rangle}] \leq \exp\{\frac{u^2 \sigma_z^2 \|\mathbf{v}\|^2}{2}\}, \forall u \in \mathbb{R}, \mathbf{v} \in \mathbb{R}^q$.
    (ii) $\epsilon_1, \boldsymbol{\epsilon}_2$ are mean-zero sub-Gaussian. That is, $\mathbb{E}[\epsilon_1] = 0$, $\mathbb{E}[\boldsymbol{\epsilon}_2] = \mathbf{0}$, and for some $\sigma_1, \sigma_2 > 0$, $\mathbb{E}[e^{u\epsilon_1}] \leq \exp\{\frac{u^2 \sigma_1^2}{2}\}$, and $\mathbb{E}[e^{u\langle \boldsymbol{\epsilon}_2, \mathbf{v} \rangle}] \leq \exp\{\frac{u^2 \sigma_2^2 \|\mathbf{v}\|^2}{2}\}, \forall u \in \mathbb{R}, \mathbf{v} \in \mathbb{R}^p$.

---

[2]Throughout this paper, we assume each entry of the dataset is independently and identically distributed (i.i.d.).

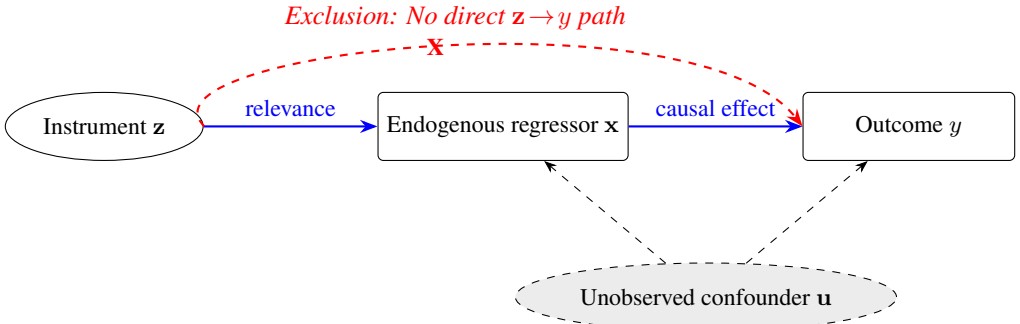

Figure 1: IVaR model: Instrument $\mathbf{z}$ is correlated with the endogenous regressor $\mathbf{x}$ and influences the outcome $y$ only indirectly through $\mathbf{x}$, while an unobserved confounder $\mathbf{u}$ affects both $\mathbf{x}$ and $y$.

Assumption 2 provides the minimal conditions required to leverage concentration results from high-dimensional random design analysis (Vershynin, 2018). Specifically, with condition (i), we have the high-probability concentration bound for the empirical covariance matrix $\frac{\mathbf{Z}^\top \mathbf{Z}}{n}$ (see Lemma D.2). Condition (ii) further ensures high-probability concentration of the cross terms $\frac{\mathbf{Z}^\top \boldsymbol{\mathcal{E}}_1}{n}$ and $\frac{\mathbf{Z}^\top \boldsymbol{\mathcal{E}}_2}{n}$ (see Lemma D.3), where $(\boldsymbol{\mathcal{E}}_1, \boldsymbol{\mathcal{E}}_2) = \{(\epsilon_{1,i}, \boldsymbol{\epsilon}_{2,i})\}_{i=1}^n$ denotes the sample realization of errors. With these conditions, we derive high-probability concentration bound for the sample covariance matrix of $\hat{\mathbf{X}} := \mathbf{Z}\hat{\boldsymbol{\Theta}}$ (see Lemma D.6), and finally establish the non-asymptotic error bound $\|\hat{\boldsymbol{\beta}} - \boldsymbol{\beta}\|$ (see Lemma D.7).

Privacy in IVaR may be required at different levels depending on the application. In some cases, protecting only the causal effect $\boldsymbol{\beta}$ is sufficient, for instance when the first-stage compliance relation $\boldsymbol{\Theta}$ is public, secondary, or not sensitive. In other cases, privacy must also extend to the first-stage parameter $\boldsymbol{\Theta}$, such as when instruments involve sensitive behavioral data, proprietary mechanisms, or institutional policies. To ensure end-to-end privacy in the IVaR model, we adopt the framework of zCDP. We allocate two privacy parameters: $\rho_1$ for the first-stage parameter estimates $\{\boldsymbol{\Theta}^{(t)}\}_{t=1}^T$, and $\rho_2$ for the second-stage parameter estimates $\{\boldsymbol{\beta}^{(t)}\}_{t=1}^T$. By the composition property of zCDP, the overall procedure satisfies $(\rho_1 + \rho_2)$-zCDP.

## 3 Algorithm and Theoretical Guarantees

We begin with a baseline two-stage gradient descent algorithm, denoted as `2S-GD`, for solving the IVaR problem equation 2. The detailed procedure is deferred to Appendix A, Algorithm 2. The method alternates between two coupled updates at each iteration: (i) updating the first-stage projection matrix $\boldsymbol{\Theta}^{(t)}$, which maps instruments $\mathbf{Z}$ to covariates $\mathbf{X}$, and (ii) updating the second-stage regression parameter $\boldsymbol{\beta}^{(t)}$ based on the predicted covariates. This iterative procedure can be viewed as a gradient-based analogue of the classical two-stage least squares estimator.

In this section, we propose a differentially private two-stage gradient descent algorithm, termed `DP-2S-GD`, to solve the IVaR problem equation 2 while ensuring rigorous privacy guarantees. The algorithm is summarized in Algorithm 1. Compared with `2S-GD`, `DP-2S-GD` incorporates two key modifications: (i) per-sample clipping is applied to gradients in both stages to bound the sensitivity of each update, ensuring that no single datapoint can disproportionately affect the results, and (ii) Gaussian perturbations are injected into both the $\boldsymbol{\Theta}$- and $\boldsymbol{\beta}$-updates at every iteration, with noise scales calibrated to the target privacy budgets $\rho_1$ and $\rho_2$.

The privacy analysis proceeds by treating the two stages as separate Gaussian mechanisms with sensitivity controlled by clipping parameters $\gamma_1$ and $\gamma_2$. By the properties of zero-concentrated differential privacy, the choice of noise scales $\lambda_1, \lambda_2$ uniquely determines the effective privacy losses $\rho_1, \rho_2$, which compose additively across iterations. Consequently, for any pre-specified privacy budgets $(\rho_1, \rho_2)$, one can calibrate $(\lambda_1, \lambda_2)$ to ensure that `DP-2S-GD` achieves the desired privacy guarantees. We next establish formal theoretical results, including both privacy accounting and utility bounds for the resulting estimators.

---

**Algorithm 1** DP-2S-GD

---

1: **Input:** Data $\mathbf{Z} \in \mathbb{R}^{n \times q}$, $\mathbf{X} \in \mathbb{R}^{n \times p}$, $\mathbf{Y} \in \mathbb{R}^n$, target privacy budgets $\rho_1, \rho_2 > 0$, step sizes $\eta, \alpha > 0$, number of iterations $T$
2: **Parameters:** Noise scales $\lambda_1, \lambda_2 > 0$, clipping thresholds $\gamma_1, \gamma_2 > 0$
3: Initialize $\boldsymbol{\beta}^{(0)} = \mathbf{0}_p$, $\boldsymbol{\Theta}^{(0)} = \mathbf{0}_{q \times p}$
4: **for** $t = 0, 1, \ldots, T-1$ **do**
5:    Draw $\boldsymbol{\Xi}^{(t)}$ with $\text{vec}(\boldsymbol{\Xi}^{(t)}) \sim \mathcal{N}(\mathbf{0}, \lambda_1^2 \mathbf{I}_q \otimes \mathbf{I}_p)$
6:    Draw $\boldsymbol{\nu}^{(t)} \sim \mathcal{N}(\mathbf{0}, \lambda_2^2 \mathbf{I}_p)$
7:    $\boldsymbol{\Theta}^{(t+1)} = \boldsymbol{\Theta}^{(t)} - \frac{\eta}{n} \sum_{i=1}^{n} \text{CLIP}_{\gamma_1}\left(\mathbf{z}_i(\mathbf{z}_i^\top \boldsymbol{\Theta}^{(t)} - \mathbf{x}_i^\top)\right) + \eta \boldsymbol{\Xi}^{(t)}$
8:    $\boldsymbol{\beta}^{(t+1)} = \boldsymbol{\beta}^{(t)} - \frac{\alpha}{n} \sum_{i=1}^{n} \text{CLIP}_{\gamma_2}\left(\boldsymbol{\Theta}^{(t)\top} \mathbf{z}_i(\mathbf{z}_i^\top \boldsymbol{\Theta}^{(t)} \boldsymbol{\beta}^{(t)} - y_i)\right) + \alpha \boldsymbol{\nu}^{(t)}$
9: **end for**
10: **return** $\{\boldsymbol{\Theta}^{(t)}\}_{t=1}^T$, $\{\boldsymbol{\beta}^{(t)}\}_{t=1}^T$

---

**Proposition 3.1.** *If we set* $\lambda_1 = \frac{2\gamma_1}{n}\sqrt{\frac{T}{\rho_1}}$ *and* $\lambda_2 = \frac{2\gamma_2}{n}\sqrt{\frac{T}{\rho_2}}$*, Algorithm 1 is* $\rho$*-zCDP, where* $\rho := \rho_1 + \rho_2 = \frac{2T}{n^2}\left(\frac{\gamma_1^2}{\lambda_1^2} + \frac{\gamma_2^2}{\lambda_2^2}\right)$.

The proof of Proposition 3.1 is provided in Appendix B.

**Remark 3.1.** Proposition 3.1 highlights several tradeoffs among the parameters. To preserve the same privacy levels $\rho_1, \rho_2$, the noise scales $\lambda_1, \lambda_2$ must increase with larger clipping thresholds $\gamma_1, \gamma_2$, or with larger number of iterations $T$. Conversely, a larger sample size $n$ allows for smaller noise scales while maintaining the same privacy guarantees.

**Theorem 3.1.** *For any fixed* $\boldsymbol{\Theta} \in \mathbb{R}^{q \times p}$ *and* $\boldsymbol{\beta} \in \mathbb{R}^p$*, consider the Algorithm 1 with fixed step sizes satisfying*

$$0 < \eta < \frac{2}{(1 + \delta(\tau))^2}, \quad 0 < \alpha < \frac{4}{2\bar{\gamma}(\tau) + \underline{\gamma}(\tau)}, \tag{3}$$

*under Assumption 2, with parameters*

$$\lambda_1 = \frac{2\gamma_1}{n}\sqrt{\frac{T}{\rho_1}}, \quad \lambda_2 = \frac{2\gamma_2}{n}\sqrt{\frac{T}{\rho_2}}, \quad \gamma_1 = \gamma_2 = c_0\left(\sqrt{q} + \sqrt{\tau + \log(nT)}\right)^2, \tag{4}$$

*and number of iterations*

$$T \lesssim \frac{\rho_1 n^{2-\epsilon}}{p(\sqrt{q} + \sqrt{\tau})^6}, \tag{5}$$

*where* $\epsilon > 0$ *is a small constant. If*

$$n \geq c_1 \max\left\{pq(\tau + \log(pq))^2, \frac{(\sqrt{q} + \sqrt{\tau})^3}{\sqrt{\min\{\rho_1, \rho_2\}}}\right\}, \tag{6}$$

*for any fixed* $\tau$*, with probability* $1 - c_2 e^{-\tau}$*, we have*

$$\|\boldsymbol{\beta}^{(T)} - \hat{\boldsymbol{\beta}}\| \lesssim \kappa(\tau)^{\frac{T}{2}} + \frac{\sqrt{p}(\sqrt{q} + \sqrt{\tau})^3}{n\sqrt{\min\{\rho_1, \rho_2\}}}\sqrt{T} + \frac{\sqrt{pq}(\tau + \log(pq))}{\sqrt{n}}, \tag{7}$$

*where* $0 < \kappa(\tau) < 1$ *is the contraction rate,* $\delta(\tau) > 0$ *is a numerically small term, and* $\bar{\gamma}(\tau), \underline{\gamma}(\tau)$ *are the high-probability upper/lower bounds on the eigenvalues of* $\frac{\hat{\boldsymbol{\Theta}}^\top \mathbf{Z}^\top \mathbf{Z} \hat{\boldsymbol{\Theta}}}{n}$*. The specific definitions of* $\delta(\tau), \bar{\gamma}(\tau), \underline{\gamma}(\tau)$*, and* $\kappa(\tau)$ *are deferred to equation 10.*

The proof of Theorem 3.1 is presented in Appendix C. We now offer several remarks regarding this theorem. In the presentation of Theorem 3.1, all constants $c_0, c_1, c_2$ and scaling factors hidden in "$\lesssim$" are independent of major parameters $n, p, q, T, \rho_1, \rho_2, \tau$. These constants only depend on problem-specific parameters $\boldsymbol{\beta}, \boldsymbol{\Theta}, \sigma_z, \sigma_1, \sigma_2$.

**Remark 3.2.** Consider the population optimization problem $\min_{\boldsymbol{\beta}} \tilde{\mathcal{L}}(\boldsymbol{\beta}) = \mathbb{E}\left[(y - \mathbf{z}^\top \boldsymbol{\Theta} \boldsymbol{\beta})^2\right]$, and the (deterministic) two-stage gradient descent algorithm:

$$\boldsymbol{\Theta}^{(t+1)} = \boldsymbol{\Theta}^{(t)} - \eta_{GD} \mathbb{E}\left[\mathbf{z}(\mathbf{z}^\top \boldsymbol{\Theta}^{(t)} - \mathbf{x}^\top)\right], \qquad \boldsymbol{\beta}^{(t+1)} = \boldsymbol{\beta}^{(t)} - \alpha_{GD} \mathbb{E}\left[\boldsymbol{\Theta}^\top \mathbf{z}(\mathbf{z}^\top \boldsymbol{\Theta} \boldsymbol{\beta}^{(t)} - y)\right].$$

It can be easily shown that under Assumption 2, the sufficient condition for learning rates to guarantee *monotonic* convergence are

$$0 < \eta_{GD} < 2, \quad 0 < \alpha_{GD} < \frac{2}{\|\boldsymbol{\Theta}\|^2}.$$

We note that in our learning rate condition equation 3, we introduce $\delta(\tau)$ and $\psi(\tau)$ to account for the randomness in data. If we have infinite samples, the condition equation 3 becomes

$$0 < \eta < 2, \quad 0 < \alpha < \frac{4}{2\|\boldsymbol{\Theta}\|^2 + \sigma_{\min}^2(\boldsymbol{\Theta})}.$$

Comparing to $\eta_{GD}$ and $\alpha_{GD}$, notice that we have the same $\eta$ condition. However, the $\alpha$ condition is slightly tighter to control the randomness introduced by the first-stage estimates $\boldsymbol{\Theta}^{(t)}$.

**Remark 3.3.** From equation 10, the optimal contraction rate $\kappa^\star(\tau)$ is achieved when the learning rates are set as

$$\eta_{\text{approx}}^\star = \frac{2}{(1 + \delta(\tau))^2 + (1 - \delta(\tau))^2}, \quad \alpha_{\text{approx}}^\star = \frac{2}{\bar{\gamma}(\tau) + \underline{\gamma}(\tau)}. \tag{8}$$

In this case, we have

$$\kappa_{\boldsymbol{\beta}}^\star(\tau) = \frac{\bar{\gamma}(\tau)}{\bar{\gamma}(\tau) + \underline{\gamma}(\tau)}, \quad \kappa_{\boldsymbol{\Theta}}^\star(\tau) = \frac{(1 + \delta(\tau))^2 - (1 - \delta(\tau))^2}{(1 + \delta(\tau))^2 + (1 - \delta(\tau))^2}, \kappa^\star(\tau) = \max\left\{\kappa_{\boldsymbol{\beta}}^\star(\tau), \kappa_{\boldsymbol{\Theta}}^\star(\tau)\right\}.$$

We emphasize that although $\eta_{\text{approx}}^\star$ and $\alpha_{\text{approx}}^\star$ minimize the contraction rate, they should be viewed as approximately optimal step sizes, as the scaling constants in the bound equation 7 vary with different choices of step sizes. Empirically, we also observe that the estimator's error is fairly insensitive in a neighborhood of $\eta_{\text{approx}}^\star, \alpha_{\text{approx}}^\star$. See Appendix G.1 for empirical results.

**Remark 3.4.** From Proposition 3.1, the choice of $\lambda_1, \lambda_2$ in equation 4 guarantees that Algorithm 1 is $\rho$-zCDP. The parameters $\gamma_1$ and $\gamma_2$ are selected so that, with high probability, the clipping operation does not alter the gradients; see Lemma D.1 for details.

**Remark 3.5.** The error bound equation 7 consists of three dominant terms. The first term $\kappa(\tau)^{\frac{T}{2}}$ characterizes the convergence of the gradient descent algorithm, which decays exponentially with $T$. The second term $\frac{\sqrt{p}(\sqrt{q}+\sqrt{\tau})^3}{n\sqrt{\min\{\rho_1,\rho_2\}}}\sqrt{T}$ captures the cumulative effect of the injected Gaussian noise, which grows with $\sqrt{T}$ due to the parameter choices in equation 4 that ensure privacy. The third term $\frac{\sqrt{pq}(\tau+\log(pq))}{\sqrt{n}}$ represents the inherent statistical error in estimating $\hat{\boldsymbol{\beta}}$ via noiseless gradient descent, which decreases with larger sample size $n$. This decomposition highlights the trade-offs between convergence phase and privacy requirement, while also accounting for the structural statistical accuracy attainable from gradient descent.

**Remark 3.6.** The condition for $T$ in equation 4 is necessary to control the noise scale $\lambda_1$ in Proposition 3.1, since the derivation of equation 7 relies on the high-probability concentration of $\|\boldsymbol{\Theta}^{(T)} - \hat{\boldsymbol{\Theta}}\|$. With limited sample size $n$, if $\rho_1$ is small, i.e. we want high privacy on $\boldsymbol{\Theta}^{(1)}, \ldots, \boldsymbol{\Theta}^{(T)}$, we can only set a moderate number of iterations $T$, otherwise the bound equation 7 doesn't hold. See Section 4 for experiments.

**Remark 3.7.** For given sample size $n$, the dominating terms for each $T$ range are:

$$\|\boldsymbol{\beta}^{(T)} - \hat{\boldsymbol{\beta}}\| \lesssim \begin{cases} \kappa(\tau)^{\frac{T}{2}}, & \text{if } T \leq \dfrac{\log\left(\frac{n}{pq(\tau+\log(pq))^2}\right)}{\log\left(\frac{1}{\kappa(\tau)}\right)}, \\[3mm] \dfrac{\sqrt{pq}\,(\tau + \log(pq))}{\sqrt{n}}, & \text{if } \dfrac{\log\left(\frac{n}{pq(\tau+\log(pq))^2}\right)}{\log\left(\frac{1}{\kappa(\tau)}\right)} < T \leq \dfrac{n\min\{\rho_1,\rho_2\}q(\tau+\log(pq))^2}{(\sqrt{q}+\sqrt{\tau})^6}, \\[3mm] \dfrac{\sqrt{p}(\sqrt{q}+\sqrt{\tau})^3}{n\sqrt{\min\{\rho_1,\rho_2\}}}\sqrt{T}, & \text{if } \dfrac{n\min\{\rho_1,\rho_2\}q(\tau+\log(pq))^2}{(\sqrt{q}+\sqrt{\tau})^6} < T \lesssim \dfrac{\rho_1 n^{2-\epsilon}}{p(\sqrt{q}+\sqrt{\tau})^6}. \end{cases}$$

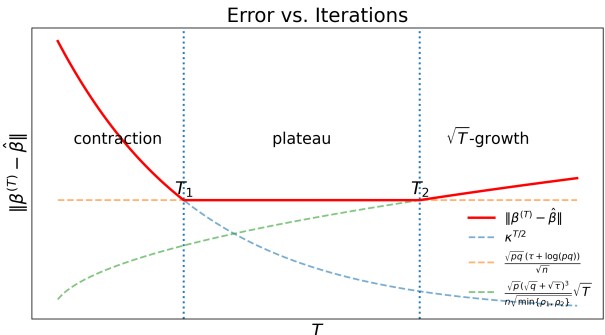

Figure 2: Qualitative trend of the error bound equation 7 as a function of $T$.

Hence, the optimum number of iterations $T$ is sub-linear but super-logarithmic to $n$. Figure 2 qualitatively illustrates the trend of the error bound equation 7 as a function of $T$. This is consistent with our experimental observations in Section 4.

**Corollary 3.1.** *Consider running Algorithm 1 with $\rho_1 = \infty$ and $\rho_2 = \infty$ (i.e. no privacy provided). For any $T > 0$, the bound equation 7 is dominated by*

$$\|\boldsymbol{\beta}^{(T)} - \hat{\boldsymbol{\beta}}\| \lesssim \kappa(\tau)^{\frac{T}{2}} + \frac{\sqrt{pq}(\tau + \log(pq))}{\sqrt{n}}, \tag{9}$$

*which is exactly the convergence rate of the `2S-GD` algorithm 2.*

**Remark 3.8.** We note that when $\rho_1, \rho_2 = \infty$ and $T \to \infty$, the error rate equation 9 still has an additional $\sqrt{p}$ factor compared to the error rate of 2SLS estimator $\|\hat{\boldsymbol{\beta}} - \boldsymbol{\beta}\|$ (see Lemma D.7 for the precise statement). This performance gap is an inherent limitation of gradient-based approximations to 2SLS, which is further confirmed by simulations in Appendix G.3. As an intuitive explanation, the closed-form 2SLS estimator solves both stages in equation 2 using sample moments, whereas the gradient-descent procedure has to approximate the second-stage moment condition. In the population, the optimality condition for 2SLS ensures $\mathbb{E}[\mathbf{z}(\mathbf{y} - \mathbf{x}^\top \boldsymbol{\beta}^\star)] = 0$. However, in the finite-sample gradient-descent iteration, the update direction will eventually depend on $\frac{1}{n}\mathbf{Z}^\top(\mathbf{Y} - \mathbf{Z}\hat{\boldsymbol{\Theta}}\hat{\boldsymbol{\beta}})$, which involves the empirical residual $\mathbf{r} := \mathbf{Y} - \mathbf{Z}\hat{\boldsymbol{\Theta}}\hat{\boldsymbol{\beta}}$. In particular, the quantity $\|\frac{1}{n}\mathbf{Z}^\top\mathbf{r}\|$ has an error rate $\frac{\sqrt{pq}(\tau + \log(pq))}{\sqrt{n}}$ (see Lemma D.8), which yields the additional $\sqrt{p}$ factor.

**Remark 3.9.** In practice, the intermediate estimates $\{\boldsymbol{\Theta}^{(t)}\}_{t=1}^T$ are not always required to be released, so in some settings it suffices to ensure privacy only for $\{\boldsymbol{\beta}^{(t)}\}_{t=1}^T$. In Algorithm 1, setting $\rho_1 = \infty$ implies that no noise $\boldsymbol{\Xi}^{(t)}$ needs to be injected in the first stage, and we can simply return $\{\boldsymbol{\beta}^{(t)}\}_{t=1}^T$ under privacy budget $\rho_2$. Under this regime, the error bound equation 7 continues to hold, except that the condition on $T$ in equation 5 is no longer required. See Appendix F.1 for further details.

## 4 EXPERIMENTS

We conduct experiments using both synthetic data and real data to validate our theoretical findings. For all experiments, we set $\tau = 5$, and step sizes $\eta = \frac{1}{(1+\delta(\tau))^2}$, $\alpha = \frac{2}{2\overline{\gamma}(\tau) + \underline{\gamma}(\tau)}$. As a practical guideline, $\rho = 0.1$ is considered as strong privacy, $\rho = 1$ is considered as moderate privacy, and $\rho = 10$ is considered as weak privacy[3].

### 4.1 SYNTHETIC DATA SIMULATIONS

We generate synthetic data according to the IVaR model in equation 1. To simulate the correlation between $\epsilon_1$ and $\epsilon_2$, we include a confounder $\mathbf{u} \in \mathbb{R}^r$, and set $\epsilon_1 = \boldsymbol{\Phi}^\top \mathbf{u}_i + \boldsymbol{\epsilon}_x$ and $\epsilon_2 = \boldsymbol{\phi}^\top \mathbf{u} + \epsilon_y$,

---

[3]The corresponding $(\epsilon, \delta)$-DP values using the conversion formula $\epsilon = \rho + 2\sqrt{\rho \log(1/\delta)}$ (with $\delta = 10^{-5}$): $\rho = 0.1 \Leftrightarrow (\epsilon, \delta) = (2.25, 10^{-5})$, $\rho = 1 \Leftrightarrow (\epsilon, \delta) = (7.79, 10^{-5})$, and $\rho = 10 \Leftrightarrow (\epsilon, \delta) = (31.47, 10^{-5})$.

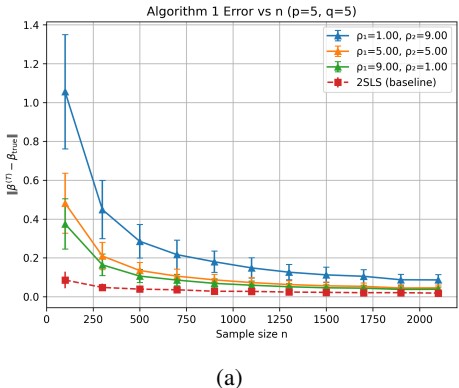
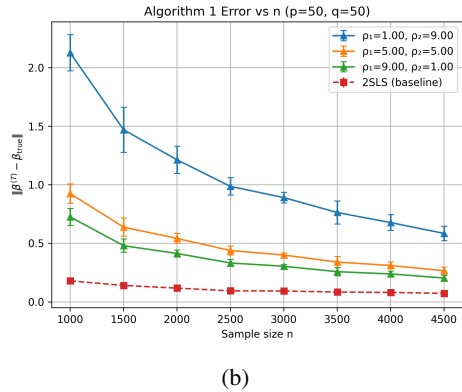

(a)  (b)

Figure 3: Comparison of Algorithm 1's performance versus $n$. We set $T = 20$, (a) $p = q = 5$, (b) $p = q = 50$. Note that the $T$ condition equation 4 is not satisfied in (b). We set the total budget $\rho = 10$ and compare three regimes: (i) $\rho_1 = 1, \rho_2 = 9$, (ii) $\rho_1 = 5, \rho_2 = 5$, (iii) $\rho_1 = 9, \rho_2 = 1$. The curves are averaged over 100 runs, with vertical bars representing the standard errors.

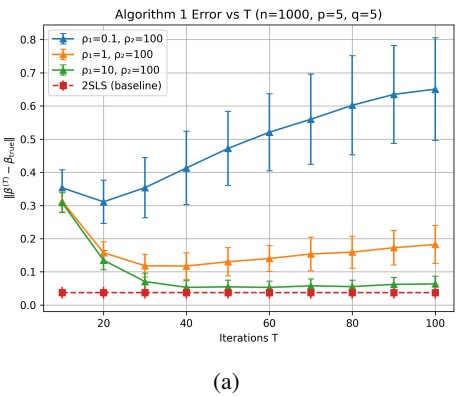
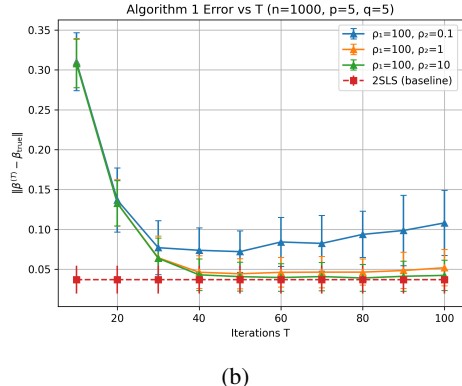

(a)  (b)

Figure 4: Comparison of Algorithm 1's performance versus number of iterations $T$. We fix $n = 1000$, $p = q = 5$, (a) keep $\rho_2$ large and vary $\rho_1$, (b) keep $\rho_1$ large and vary $\rho_2$. The curves are averaged over 100 runs, with vertical bars representing the standard errors.

and generate each entry of the dataset $(\mathbf{Z}, \mathbf{X}, \mathbf{Y}) = \{(\mathbf{z}_i, \mathbf{x}_i, y_i)\}_{i=1}^n$ according to the following model: $\mathbf{x}_i = \mathbf{\Theta}^\top \mathbf{z}_i + \mathbf{\Phi}^\top \mathbf{u}_i + \boldsymbol{\epsilon}_{x,i}$, and $\mathbf{y}_i = \boldsymbol{\beta}^\top \mathbf{x}_i + \boldsymbol{\phi}^\top \mathbf{u}_i + \epsilon_{y,i}$, where the ground-truth parameters are $\boldsymbol{\beta} \in \mathbb{R}^p, \mathbf{\Theta} \in \mathbb{R}^{q \times p}, \mathbf{\Phi} \in \mathbb{R}^{r \times p}, \boldsymbol{\phi} \in \mathbb{R}^r$. These parameters are drawn as follows: $\boldsymbol{\beta} \sim \mathcal{N}(\mathbf{0}, \mathbf{I}_p), \mathbf{\Theta} \sim 5\mathbf{I}_{q \times p} + \mathbf{E}$ with $\mathbf{E}_{ij} \sim \mathcal{N}(0, 1)$. $\mathbf{\Phi}_{ij} \sim \mathcal{N}(0, 1)$, and $\boldsymbol{\phi} \sim \mathcal{N}(\mathbf{0}, \mathbf{I}_r)$. For each simulation, we then sample $\mathbf{z}_i \sim \mathcal{N}(\mathbf{0}, \mathbf{I}_q), \mathbf{u}_i \sim \mathcal{N}(\mathbf{0}, \mathbf{I}_r), \boldsymbol{\epsilon}_{x,i} \sim \mathcal{N}(\mathbf{0}, \mathbf{I}_p)$, and $\epsilon_{y,i} \sim \mathcal{N}(0, 1)$.

Figure 3 compares the performance of Algorithm 1 across different sample sizes $n$ under varying privacy allocations. We fix the total privacy budget at $\rho = \rho_1 + \rho_2 = 10$, set the number of iterations to $T = 20$, and examine three regimes: (i) $\rho_1 = 1, \rho_2 = 9$, (ii) $\rho_1 = 5, \rho_2 = 5$, and (iii) $\rho_1 = 9, \rho_2 = 1$. In Figure 3(a), with $p = q = r = 5$, all points lie in the plateau region of Figure 2, so the error decreases at the rate $\frac{1}{\sqrt{n}}$. In contrast, Figure 3(b) sets $p = q = r = 50$. Here, $T = 20$ violates condition equation 5, leading to significantly larger errors compared to Figure 3(a). The impact of $T$ is further investigated in Figure 4, from which we observe that, with limited sample size $n$, if we enforce high privacy guarantee on $\{\mathbf{\Theta}^{(t)}\}_{t=1}^T$ (i.e. with small $\rho_1$), the error grows significantly after certain $T$ is reached. This cutoff aligns with the condition on $T$ specified in equation 5. In contrast, when privacy is required only for $\{\boldsymbol{\beta}^{(t)}\}_{t=1}^T$ (i.e., with small $\rho_2$), the error behavior closely matches the theoretical predictions illustrated in Figure 2.

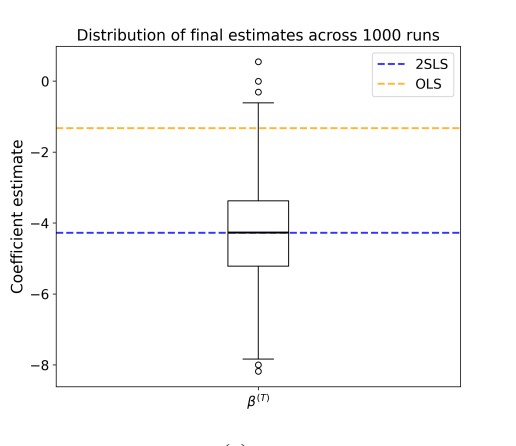 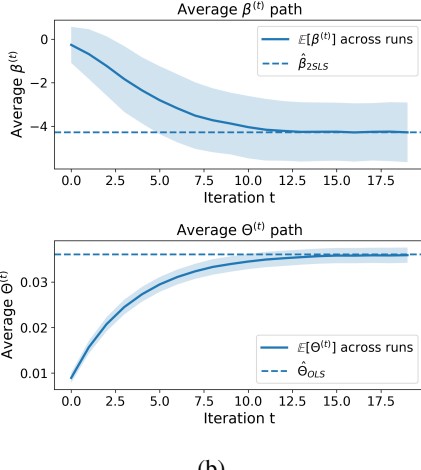

(a)                              (b)

Figure 5: Results on the Angrist dataset with $T = 20, \rho_1 = 1, \rho_2 = 1$. (a) Boxplot of estimated $\boldsymbol{\beta}^{(T)}$, over 1000 runs. (b) Learning paths of parameters $\boldsymbol{\beta}^{(t)}, \boldsymbol{\Theta}^{(t)}$, over 1000 runs. The shaded area represents the standard error.

## 4.2 REAL-DATA EXPERIMENTS

We further evaluate our algorithm on the Angrist dataset (Angrist & Evans, 1998), which has been widely applied in the IVaR literature. This study examines the causal effect of children bearing on female labor supply, leveraging the gender composition of the first two children as an instrument[4]. The endogenous regressor $\mathbf{x}$ is the number of children bearing, the outcome $\mathbf{y}$ is the mother's labor supply measured in number of working weeks per year, and the instrument $\mathbf{z}$ is a binary variable indicating whether the first two children are of the same gender. The original dataset contains $394,835$ samples. For illustration purpose, we randomly draw a subset of $20,000$ samples and keep $n = 8065$ effective observations with number of children $\geq 2$. We center all variables $\mathbf{z}, \mathbf{x}, \mathbf{y}$ and run Algorithm 1 with $T = 20$ iterations. Figure 5 presents the results over 1000 independent runs with privacy budgets $\rho_1 = 1, \rho_2 = 1$. As shown in Figure 5a, the estimated $\boldsymbol{\beta}^{(T)}$ concentrates around $-4.3$, indicating that having an additional child reduces the mother's labor supply by approximately $4.3$ weeks per year. This estimate is consistent with the 2SLS benchmark.

From Figure 5b, we observe that Algorithm 1 converges in expectation after about 15 iterations. The dispersion of the estimates is determined by the privacy budgets: increasing $\rho_1$ and $\rho_2$ yield estimates that are more tightly concentrated around the 2SLS benchmark, while smaller budgets result in greater variability. Additional experiments are provided in Appendix G.4.

## 5 CONCLUSION

We have introduced `DP-2S-GD`, a differentially private two-stage gradient descent method for IVaR problem. The algorithm achieves $(\rho_1 + \rho_2)$-zCDP by injecting carefully calibrated Gaussian noise. We have established finite-sample convergence guarantees that capture the trade-offs among optimization dynamics, privacy constraints, and statistical error. Our theoretical analysis shows that setting the number of iterations $T$ to be sub-linear yet super-logarithmic in $n$ minimizes the estimation error, a result that is corroborated by our experiments. We have further illustrated the practical utility of our method through an application to the Angrist dataset. On the other hand, we note that, regardless of the privacy constraint, the convergence of the two-stage gradient descent estimator to $\hat{\boldsymbol{\beta}}$ is slower by a $\sqrt{p}$ compared to the convergence of $\hat{\boldsymbol{\beta}}$ to the true parameter $\boldsymbol{\beta}$ (see Remark 3.8). Improving this rate (via algorithmic modifications) and establishing lower-bounds for privacy-accuracy tradeoffs for the IVaR problem are interesting future directions.

---

[4]Research shows that parents whose first two children are of the same sex are significantly more likely to have an additional child (Westoff & Parke, 1972). At the same time, the sex composition of the first two children can be treated as randomly assigned and is not directly related to the mother's labor supply.

REPRODUCIBILITY STATEMENT

All theoretical results are proved in the Appendix, and code for reproducing all experiments is provided in the supplementary material. LLM was used to polish writing and for finding related work.

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

## A  ADDITIONAL DEFINITIONS

**Definition A.1** (Rényi Divergence). Let $P$ and $Q$ be probability distributions on a measurable space $(\mathcal{X}, \mathcal{F})$, with $P$ absolutely continuous with respect to $Q$. For $\alpha > 1$, the Rényi divergence of order $\alpha$ between $P$ and $Q$ is defined as

$$D_\alpha(P \,\|\, Q) = \frac{1}{\alpha - 1} \log \int_{\mathcal{X}} \left( \frac{dP}{dQ}(x) \right)^\alpha \, dQ(x).$$

This family of divergences interpolates between several well-known measures: (i) As $\alpha \to 1$, $D_\alpha(P\|Q) \to D_{\mathrm{KL}}(P\|Q)$, the Kullback–Leibler divergence, and (ii) As $\alpha \to \infty$, $D_\alpha(P\|Q) \to \log\sup_{x \in \mathcal{X}} \frac{dP}{dQ}(x)$, the log of the essential supremum of the likelihood ratio.

**Definition A.2** (2S-GD). We introduce the baseline two-stage gradient descent algorithm without privacy constraints, denoted as `2S-GD`, in Algorithm 2.

---

**Algorithm 2** 2S-GD

---

1: **Input:** Data $\mathbf{Z} \in \mathbb{R}^{n \times q}$, $\mathbf{X} \in \mathbb{R}^{n \times p}$, $\mathbf{Y} \in \mathbb{R}^n$
2: **Parameters:** Step sizes $\eta, \alpha > 0$, number of iterations $T$
3: Initialize $\boldsymbol{\beta}^{(0)} = \mathbf{0}_p$, $\boldsymbol{\Theta}^{(0)} = \mathbf{0}_{q \times p}$
4: **for** $t = 0, 1, \ldots, T-1$ **do**
5:     $\boldsymbol{\Theta}^{(t+1)} = \boldsymbol{\Theta}^{(t)} - \frac{\eta}{n} \sum_{i=1}^n \mathbf{z}_i (\mathbf{z}_i^\top \boldsymbol{\Theta}^{(t)} - \mathbf{x}_i^\top)$
6:     $\boldsymbol{\beta}^{(t+1)} = \boldsymbol{\beta}^{(t)} - \frac{\alpha}{n} \sum_{i=1}^n \boldsymbol{\Theta}^{(t)\top} \mathbf{z}_i (\mathbf{z}_i^\top \boldsymbol{\Theta}^{(t)} \boldsymbol{\beta}^{(t)} - y_i)$
7: **end for**
8: **return** $\{\boldsymbol{\Theta}^{(t)}\}_{t=1}^T$, $\{\boldsymbol{\beta}^{(t)}\}_{t=1}^T$

---

## B  PROOF OF PROPOSITION 3.1

*Proof.* At iteration $t$ we are releasing two Gaussian-mechanisms on sums of clipped per-sample gradients (each clipped to norm not larger than $\gamma_1$ and $\gamma_2$), one with noise scale $\lambda_1$ (for $\boldsymbol{\Theta}$) and one with noise scale $\lambda_2$ (for $\boldsymbol{\beta}$). By the standard zCDP analysis:

- $\boldsymbol{\Theta}$-update: Sensitivity of the summed (clipped) gradients is $\Delta_1 = \frac{2\gamma_1}{n}$, and we add noise $\eta\boldsymbol{\Xi}$ with $\mathrm{vec}(\boldsymbol{\Xi}) \sim \mathcal{N}\left(0, \lambda_1^2 \mathbf{I}_q \otimes \mathbf{I}_p\right)$. By property of Gaussian mechanism, this step satisfies $\rho_1 = \frac{2\gamma_1^2}{n^2\lambda_1^2}$-zCDP

- $\boldsymbol{\beta}$-update: Similarly, $\Delta_2 = \frac{2\gamma_2}{n}$, this step is $\rho_2 = \frac{2\gamma_2^2}{n^2\lambda_2^2}$

By linear composition each iteration costs

$$\rho_{\text{per it}} = \rho_1 + \rho_2 = \frac{2}{n^2} \left( \frac{\gamma_1^2}{\lambda_1^2} + \frac{\gamma_2^2}{\lambda_2^2} \right).$$

Over $T$ iterations the overall mechanism satisfies $\rho = \frac{2T}{n^2} \left( \frac{\gamma_1^2}{\lambda_1^2} + \frac{\gamma_2^2}{\lambda_2^2} \right)$-zCDP.  □

## C  PROOF OF THEOREM 3.1

Before the full proof, we would like to provide a high-level summary of our proof strategy to facilitate readers' understanding. Conceptually, our proof tackles three key challenges:

(1) Two-stage coupling: We control a pair of noisy recursions $\{\boldsymbol{\Theta}^{(t)}\}_{t \leq T}$ and $\{\boldsymbol{\beta}^{(t)}\}_{t \leq T}$ where the second-stage gradient at time $t$ depends on the noisy first stage iterate $\boldsymbol{\Theta}^{(t)}$. This coupling does not appear in standard DP-SGD analyses.

(2) Noise propagation: Since we add Gaussian noise at every step and stage, the privacy accountant composes over all $2T$ iterations. To obtain a meaningful error bound for the final iterate $\boldsymbol{\beta}^{(T)}$, we need to separate the contributions of optimization, privacy, and sampling error. This decomposition gives rise to the three terms in Theorem 3.1 and explains the tradeoff pattern shown in Figure 2 and our experiments.

(3) Non-asymptotic rate: The analysis retains explicit dependence on all major parameters: sample size $n$, instrument dimension $q$, regressor dimension $p$, privacy budgets $(\rho_1, \rho_2)$, and confidence parameter $\tau$.

The whole proof of Theorem 3.1 is decomposed into several supporting lemmas in Appendix D to handle all these challenges.

We first re-state the result with additional details.

**Theorem 3.1.** For any fixed $\boldsymbol{\Theta} \in \mathbb{R}^{q \times p}$ and $\boldsymbol{\beta} \in \mathbb{R}^p$, consider the Algorithm 1 with step sizes satisfying

$$0 < \eta < \frac{2}{(1+\delta(\tau))^2}, \quad 0 < \alpha < \frac{4}{2\bar{\gamma}(\tau) + \underline{\gamma}(\tau)},$$

under Assumption 2, with parameters

$$\lambda_1 = \frac{2\gamma_1}{n}\sqrt{\frac{T}{\rho_1}}, \quad \lambda_2 = \frac{2\gamma_2}{n}\sqrt{\frac{T}{\rho_2}}, \quad \gamma_1 = \gamma_2 = c_0\left(\sqrt{q} + \sqrt{\tau + \log(nT)}\right)^2,$$

and number of iterations

$$T \lesssim \frac{\rho_1 n^{2-\epsilon}}{p(\sqrt{q} + \sqrt{\tau})^6},$$

where $\epsilon > 0$ is a small constant. If

$$n \geq c_1 \max\left\{pq(\tau + \log(pq))^2, \frac{\left(\sqrt{q} + \sqrt{\tau}\right)^3}{\sqrt{\min\{\rho_1, \rho_2\}}}\right\},$$

for any fixed $\tau$, with probability $1 - c_2 e^{-\tau}$, we have

$$\|\boldsymbol{\beta}^{(T)} - \hat{\boldsymbol{\beta}}\| \lesssim \kappa(\tau)^{\frac{T}{2}} + \frac{\sqrt{p}(\sqrt{q} + \sqrt{\tau})^3}{n\sqrt{\min\{\rho_1, \rho_2\}}}\sqrt{T} + \frac{\sqrt{pq}(\tau + \log(pq))}{\sqrt{n}},$$

where

$$\delta(\tau) := \frac{C_0 \sigma_z^2(\sqrt{q} + \sqrt{\tau})}{\sqrt{n}},$$

$$\underline{\gamma}(\tau) := (1 - \delta(\tau))^2 \left(\sigma_{\min}(\boldsymbol{\Theta}) - \psi(\tau)\right)^2, \quad \bar{\gamma}(\tau) := (1 + \delta(\tau))^2 \left(\|\boldsymbol{\Theta}\| + \psi(\tau)\right)^2,$$

$$\psi(\tau) := \frac{c_0 \sigma_z \sigma_2 \sqrt{pq}\,(\tau + \log(2pq))}{\sqrt{n}\,(1 - \delta(\tau))^2},$$

$$\kappa_{\boldsymbol{\beta}}(\tau) := \max\left\{|1 - \frac{\alpha\underline{\gamma}(\tau)}{2}|, |1 - \frac{\alpha(2\bar{\gamma}(\tau) + \underline{\gamma}(\tau))}{2}|\right\},$$

$$\kappa_{\boldsymbol{\Theta}}(\tau) := \max\left\{\left|1 - \eta\,(1 - \delta(\tau))^2\right|, \left|1 - \eta\,(1 + \delta(\tau))^2\right|\right\},$$

$$\kappa(\tau) := \max\{\kappa_{\boldsymbol{\beta}}(\tau), \kappa_{\boldsymbol{\Theta}}(\tau)\}.$$

(10)

*Proof.* Denote $\mathbf{e}_{\boldsymbol{\Theta}}^{(t)} := \boldsymbol{\Theta}^{(t)} - \hat{\boldsymbol{\Theta}}$ and $\mathbf{e}_{\boldsymbol{\beta}}^{(t)} := \boldsymbol{\beta}^{(t)} - \hat{\boldsymbol{\beta}}$. We have

$$
\begin{aligned}
\mathbf{e}_{\boldsymbol{\Theta}}^{(t+1)} &= \mathbf{e}_{\boldsymbol{\Theta}}^{(t)} - \frac{\eta}{n}\mathbf{Z}^{\top}\left(\mathbf{Z}\boldsymbol{\Theta}^{(t)} - \mathbf{X}\right) + \eta\boldsymbol{\Xi}^{(t)} \\
&= \left(\mathbf{I} - \frac{\eta}{n}\mathbf{Z}^{\top}\mathbf{Z}\right)\mathbf{e}_{\boldsymbol{\Theta}}^{(t)} + \frac{\eta}{n}\mathbf{Z}^{\top}\left(\mathbf{X} - \mathbf{Z}\hat{\boldsymbol{\Theta}}\right) + \eta\boldsymbol{\Xi}^{(t)} \\
&= \left(\mathbf{I} - \frac{\eta}{n}\mathbf{Z}^{\top}\mathbf{Z}\right)^{t+1}\mathbf{e}_{\boldsymbol{\Theta}}^{(0)} + \sum_{i=0}^{t}\eta\left(\mathbf{I} - \frac{\eta}{n}\mathbf{Z}^{\top}\mathbf{Z}\right)^{t-i}\left(\frac{1}{n}\mathbf{Z}^{\top}(\mathbf{X} - \mathbf{Z}\hat{\boldsymbol{\Theta}}) + \boldsymbol{\Xi}^{(i)}\right) \quad (11) \\
&= \left(\mathbf{I} - \frac{\eta}{n}\mathbf{Z}^{\top}\mathbf{Z}\right)^{t+1}\mathbf{e}_{\boldsymbol{\Theta}}^{(0)} + \underbrace{\sum_{i=0}^{t}\eta\left(\mathbf{I} - \frac{\eta}{n}\mathbf{Z}^{\top}\mathbf{Z}\right)^{t-i}\boldsymbol{\Xi}^{(i)}}_{\mathbf{N}^{(t)}},
\end{aligned}
$$

$$
\begin{aligned}
\mathbf{e}_{\boldsymbol{\beta}}^{(t+1)} &= \mathbf{e}_{\boldsymbol{\beta}}^{(t)} - \frac{\alpha}{n}\boldsymbol{\Theta}^{(t)}\mathbf{Z}^{\top}\left(\mathbf{Z}\boldsymbol{\Theta}^{(t)}\boldsymbol{\beta}^{(t)} - \mathbf{Y}\right) + \alpha\boldsymbol{\nu}^{(t)} \\
&= \left(\mathbf{I} - \frac{\alpha}{n}\boldsymbol{\Theta}^{(t)\top}\mathbf{Z}^{\top}\mathbf{Z}\boldsymbol{\Theta}^{(t)}\right)\mathbf{e}_{\boldsymbol{\beta}}^{(t)} + \frac{\alpha}{n}\left[\boldsymbol{\Theta}^{(t)\top}\mathbf{Z}^{\top}\mathbf{Y} - \boldsymbol{\Theta}^{(t)\top}\mathbf{Z}^{\top}\mathbf{Z}\boldsymbol{\Theta}^{(t)}\hat{\boldsymbol{\beta}}\right] + \alpha\boldsymbol{\nu}^{(t)} \\
&= \left(\mathbf{I} - \frac{\alpha}{n}\boldsymbol{\Theta}^{(t)\top}\mathbf{Z}^{\top}\mathbf{Z}\boldsymbol{\Theta}^{(t)}\right)\mathbf{e}_{\boldsymbol{\beta}}^{(t)} + \frac{\alpha}{n}\boldsymbol{\Theta}^{(t)\top}\mathbf{Z}^{\top}\left(\mathbf{Y} - \mathbf{Z}\boldsymbol{\Theta}^{(t)}\hat{\boldsymbol{\beta}}\right) + \alpha\boldsymbol{\nu}^{(t)} \\
&= \left(\mathbf{I} - \frac{\alpha}{n}\boldsymbol{\Theta}^{(t)\top}\mathbf{Z}^{\top}\mathbf{Z}\boldsymbol{\Theta}^{(t)}\right)\mathbf{e}_{\boldsymbol{\beta}}^{(t)} - \frac{\alpha}{n}\boldsymbol{\Theta}^{(t)\top}\mathbf{Z}^{\top}\left(\mathbf{Z}\left(\boldsymbol{\Theta}^{(t)} - \hat{\boldsymbol{\Theta}}\right)\hat{\boldsymbol{\beta}}\right) - \frac{\alpha}{n}\left(\boldsymbol{\Theta}^{(t)\top}\mathbf{Z}^{\top}\left(\mathbf{Z}\hat{\boldsymbol{\Theta}}\hat{\boldsymbol{\beta}} - \mathbf{Y}\right)\right) + \alpha\boldsymbol{\nu}^{(t)} \\
&:= \left[\mathbf{I} - \alpha\mathbf{H}^{(t)}\right]\mathbf{e}_{\boldsymbol{\beta}}^{(t)} - \frac{\alpha}{n}\boldsymbol{\Theta}^{(t)\top}\mathbf{Z}^{\top}\mathbf{Z}\mathbf{e}_{\boldsymbol{\Theta}}^{(t)}\hat{\boldsymbol{\beta}} - \frac{\alpha}{n}\boldsymbol{\Theta}^{(t)\top}\mathbf{Z}^{\top}\mathbf{r} + \alpha\boldsymbol{\nu}^{(t)},
\end{aligned}
$$

$$(12)$$

where $\mathbf{H}^{(t)} := \frac{1}{n}\boldsymbol{\Theta}^{(t)\top}\mathbf{Z}^{\top}\mathbf{Z}\boldsymbol{\Theta}^{(t)}$ and $\mathbf{r} := \mathbf{Z}\hat{\boldsymbol{\Theta}}\hat{\boldsymbol{\beta}} - \mathbf{Y}$. We first show that $\mathbf{H}^{(t)}$ is close to the target $\hat{\mathbf{H}} := \frac{1}{n}\hat{\boldsymbol{\Theta}}^{\top}\mathbf{Z}^{\top}\mathbf{Z}\hat{\boldsymbol{\Theta}}$. We define the event

$$
E_{T_0,T} = \left\{\|\mathbf{e}_{\boldsymbol{\Theta}}^{(k)}\| = \|\boldsymbol{\Theta}^{(k)} - \hat{\boldsymbol{\Theta}}\| \leq \varepsilon, \forall T_0 \leq k < T\right\}.
$$

Conditioning on the event $E_{T_0,T}$, we then have

$$
\begin{aligned}
\|\mathbf{H}^{(t)} - \hat{\mathbf{H}}\| &= \frac{1}{n}\|\boldsymbol{\Theta}^{(t)\top}\mathbf{Z}^{\top}\mathbf{Z}\boldsymbol{\Theta}^{(t)} - \hat{\boldsymbol{\Theta}}^{\top}\mathbf{Z}^{\top}\mathbf{Z}\hat{\boldsymbol{\Theta}}\| \\
&= \frac{1}{n}\|\boldsymbol{\Theta}^{(t)\top}\mathbf{Z}^{\top}\mathbf{Z}(\boldsymbol{\Theta}^{(t)} - \hat{\boldsymbol{\Theta}}) + (\boldsymbol{\Theta}^{(t)} - \hat{\boldsymbol{\Theta}})^{\top}\mathbf{Z}^{\top}\mathbf{Z}\hat{\boldsymbol{\Theta}}\| \\
&\leq \frac{1}{n}(\|\boldsymbol{\Theta}^{(t)}\| + \|\hat{\boldsymbol{\Theta}}\|)\|\mathbf{Z}^{\top}\mathbf{Z}\|\varepsilon \\
&\leq (2\|\hat{\boldsymbol{\Theta}}\| + \varepsilon)\frac{\mathbf{Z}^{\top}\mathbf{Z}}{n}\|\varepsilon, \quad \forall T_0 \leq t \leq T.
\end{aligned}
$$

From Lemma D.2, we have with probability at least $1 - 2e^{-\tau}$,

$$
\|\frac{\mathbf{Z}^{\top}\mathbf{Z}}{n}\| \leq (1 + \delta(\tau))^2,
$$

so that

$$
\|\mathbf{H}^{(t)} - \hat{\mathbf{H}}\| \leq (2\|\hat{\boldsymbol{\Theta}}\| + \varepsilon)(1 + \delta(\tau))^2\varepsilon, \quad \forall T_0 \leq t \leq T.
$$

Suppose $\underline{\gamma}(\tau), \bar{\gamma}(\tau)$ are some high probability bounds such that $\lambda_{\min}(\hat{\mathbf{H}}) \geq \underline{\gamma}(\tau) > 0$, $\lambda_{\max}(\hat{\mathbf{H}}) \leq \bar{\gamma}(\tau)$. From Lemma D.6, we can take

$$
\underline{\gamma}(\tau) := (1 - \delta(\tau))^2\left(\sigma_{\min}(\boldsymbol{\Theta}) - \frac{c_0\sigma_z\sigma_2\sqrt{pq}\left(\tau + \log(2pq)\right)}{\sqrt{n}\left(1 - \delta(\tau)\right)^2}\right)^2,
$$

$$
\bar{\gamma}(\tau) := (1 + \delta(\tau))^2\left(\|\boldsymbol{\Theta}\| + \frac{c_0\sigma_z\sigma_2\sqrt{pq}\left(\tau + \log(2pq)\right)}{\sqrt{n}\left(1 - \delta(\tau)\right)^2}\right)^2.
$$

If $\varepsilon$ satisfies the following condition:

$$\varepsilon \leq \frac{\underline{\gamma}(\tau)}{2(2\|\hat{\boldsymbol{\Theta}}\| + \varepsilon)(1 + \delta(\tau))^2},$$

i.e. we choose

$$\varepsilon \leq \sqrt{\|\hat{\boldsymbol{\Theta}}\|^2 + \frac{\underline{\gamma}(\tau)}{2(1 + \delta(\tau))^2}} - \|\hat{\boldsymbol{\Theta}}\|, \tag{13}$$

by Weyl's inequality, we then have

$$\lambda_{\min}\left(\mathbf{H}^{(t)}\right) \geq \lambda_{\min}(\hat{\mathbf{H}}) - \|\mathbf{H}^{(t)} - \hat{\mathbf{H}}\| \geq \frac{\underline{\gamma}(\tau)}{2}$$

$$\lambda_{\max}\left(\mathbf{H}^{(t)}\right) \leq \lambda_{\max}(\hat{\mathbf{H}}) + \|\mathbf{H}^{(t)} - \hat{\mathbf{H}}\| \leq \bar{\gamma}(\tau) + \frac{\underline{\gamma}(\tau)}{2}$$

This in turn implies that on $E_{T_0,T}$, when $0 < \alpha < \frac{4}{2\bar{\gamma}(\tau) + \underline{\gamma}(\tau)}$, we have

$$\|\mathbf{I} - \alpha\mathbf{H}^{(t)}\| \leq \max\left\{|1 - \alpha\lambda_{\min}(\mathbf{H}^{(t)})|, |1 - \alpha\lambda_{\max}(\mathbf{H}^{(t)})|\right\}$$

$$\leq \max\left\{|1 - \frac{\alpha\underline{\gamma}(\tau)}{2}|, |1 - \frac{\alpha(2\bar{\gamma}(\tau) + \underline{\gamma}(\tau))}{2}|\right\} := \kappa_{\boldsymbol{\beta}}(\tau) < 1, \tag{14}$$

hence the error recursion equation 12 satisfies

$$\|\mathbf{e}_{\boldsymbol{\beta}}^{(t+1)}\| \leq \kappa_{\boldsymbol{\beta}}(\tau)\|\mathbf{e}_{\boldsymbol{\beta}}^{(t)}\| + \frac{\alpha}{n}\|\boldsymbol{\Theta}^{(t)\top}\mathbf{Z}^\top\mathbf{Z}\mathbf{e}_{\boldsymbol{\Theta}}^{(t)}\hat{\boldsymbol{\beta}}\| + \frac{\alpha}{n}\|\boldsymbol{\Theta}^{(t)\top}\mathbf{Z}^\top\mathbf{r}\| + \alpha\|\boldsymbol{\nu}^{(t)}\|,$$

and

$$\|\mathbf{e}_{\boldsymbol{\beta}}^{(T)}\| \leq \kappa_{\boldsymbol{\beta}}(\tau)^{T-T_0}\|\mathbf{e}_{\boldsymbol{\beta}}^{(T_0)}\| + \frac{\alpha}{n}\sum_{k=T_0}^{T-1}\kappa_{\boldsymbol{\beta}}(\tau)^{T-1-k}\left(\|\boldsymbol{\Theta}^{(k)\top}\mathbf{Z}^\top\mathbf{Z}\mathbf{e}_{\boldsymbol{\Theta}}^{(k)}\hat{\boldsymbol{\beta}}\| + \|\boldsymbol{\Theta}^{(k)\top}\mathbf{Z}^\top\mathbf{r}\|\right) + \frac{\alpha}{1 - \kappa_{\boldsymbol{\beta}}(\tau)}\|\boldsymbol{\nu}\|$$

$$\leq \kappa_{\boldsymbol{\beta}}(\tau)^{T-T_0}\|\mathbf{e}_{\boldsymbol{\beta}}^{(T_0)}\| + \frac{\alpha\|\mathbf{Z}^\top\mathbf{Z}\|\|\hat{\boldsymbol{\beta}}\|}{n}\sum_{k=T_0}^{T-1}\kappa_{\boldsymbol{\beta}}(\tau)^{T-1-k}\|\boldsymbol{\Theta}^{(k)}\|\|\mathbf{e}_{\boldsymbol{\Theta}}^{(k)}\| + \frac{\alpha\|\mathbf{Z}^\top\mathbf{r}\|}{n}\sum_{k=T_0}^{T-1}\kappa_{\boldsymbol{\beta}}(\tau)^{T-1-k}\|\boldsymbol{\Theta}^{(k)}\|$$

$$+ \frac{\alpha}{1 - \kappa_{\boldsymbol{\beta}}(\tau)}\|\boldsymbol{\nu}\|$$

$$\leq \kappa_{\boldsymbol{\beta}}(\tau)^{T-T_0}\|\mathbf{e}_{\boldsymbol{\beta}}^{(T_0)}\| + \alpha(1 + \delta(\tau))^2\|\hat{\boldsymbol{\beta}}\|\sum_{k=T_0}^{T-1}\kappa_{\boldsymbol{\beta}}(\tau)^{T-1-k}\|\boldsymbol{\Theta}^{(k)}\|\|\mathbf{e}_{\boldsymbol{\Theta}}^{(k)}\| + \frac{\alpha\|\mathbf{Z}^\top\mathbf{r}\|}{n}\sum_{k=T_0}^{T-1}\kappa_{\boldsymbol{\beta}}(\tau)^{T-1-k}\|\boldsymbol{\Theta}^{(k)}\|$$

$$+ \frac{\alpha}{1 - \kappa_{\boldsymbol{\beta}}(\tau)}\|\boldsymbol{\nu}\|. \tag{15}$$

Under event $E_{T_0,T}$, we have the uniform bound:

$$\|\boldsymbol{\Theta}^{(k)}\| \leq \|\hat{\boldsymbol{\Theta}}\| + \varepsilon, \quad \forall T_0 \leq k < T,$$

$$\|\mathbf{e}_{\boldsymbol{\Theta}}^{(k)}\| \leq \varepsilon, \quad \forall T_0 \leq k < T.$$

Besides, from Lemma D.5 and Lemma D.7, we have when $n = \Omega(pq(\tau + \log(pq))^2)$, $\|\hat{\boldsymbol{\Theta}}\|$ and $\|\hat{\boldsymbol{\beta}}\|$ are bounded by some constants with high probability:

$$\|\hat{\boldsymbol{\beta}}\| \lesssim 1, \quad \|\hat{\boldsymbol{\Theta}}\| \lesssim 1.$$

From Lemma D.8, we have

$$\|\mathbf{Z}^\top\mathbf{r}\| \lesssim \sqrt{npq}\left(\tau + \log(pq)\right).$$

Since $\boldsymbol{\nu} \sim \mathcal{N}(0, \lambda_2^2\mathbf{I}_p)$, we have with probability $1 - e^{-\tau}$,

$$\|\boldsymbol{\nu}\| \lesssim \lambda_2\left(\sqrt{p} + \sqrt{\tau}\right).$$

Then from equation 15,

$$
\begin{aligned}
\|\mathbf{e}_{\boldsymbol{\beta}}^{(T)}\| &\leq \kappa_{\boldsymbol{\beta}}(\tau)^{T-T_0}\|\mathbf{e}_{\boldsymbol{\beta}}^{(T_0)}\| + \alpha(1+\delta(\tau))^2\|\hat{\boldsymbol{\beta}}\|\varepsilon(\|\hat{\boldsymbol{\Theta}}\|+\varepsilon)\sum_{k=T_0}^{T-1}\kappa_{\boldsymbol{\beta}}(\tau)^{T-1-k} + \frac{\alpha\|\mathbf{Z}^\top\mathbf{r}\|(\|\hat{\boldsymbol{\Theta}}\|+\varepsilon)}{n}\sum_{k=T_0}^{T-1}\kappa_{\boldsymbol{\beta}}(\tau)^{T-1-k} \\
&\quad + \frac{\alpha}{1-\kappa_{\boldsymbol{\beta}}(\tau)}\|\boldsymbol{\nu}\| \\
&\leq \kappa_{\boldsymbol{\beta}}(\tau)^{T-T_0}\|\mathbf{e}_{\boldsymbol{\beta}}^{(T_0)}\| + \frac{\alpha(1+\delta(\tau))^2\|\hat{\boldsymbol{\beta}}\|\varepsilon(\|\hat{\boldsymbol{\Theta}}\|+\varepsilon)}{1-\kappa_{\boldsymbol{\beta}}(\tau)} + \frac{\alpha\|\mathbf{Z}^\top\mathbf{r}\|(\|\hat{\boldsymbol{\Theta}}\|+\varepsilon)}{n\,(1-\kappa_{\boldsymbol{\beta}}(\tau))} + \frac{\alpha}{1-\kappa_{\boldsymbol{\beta}}(\tau)}\|\boldsymbol{\nu}\| \\
&\lesssim \kappa_{\boldsymbol{\beta}}(\tau)^{T-T_0}\|\mathbf{e}_{\boldsymbol{\beta}}^{(T_0)}\| + \varepsilon(1+\varepsilon) + \frac{\sqrt{pq}(\tau+\log(pq))}{\sqrt{n}}\,(1+\varepsilon) + \lambda_2\left(\sqrt{p}+\sqrt{\tau}\right).
\end{aligned}
\tag{16}
$$

It remains to bound $\|\mathbf{e}_{\boldsymbol{\beta}}^{(T_0)}\|$. Denote $\mathbf{L}^{(t)} := \mathbf{I} - \alpha\mathbf{H}^{(t)}, t = 0, 1, \ldots, T_0 - 1$. Note that from Lemma D.9, $\prod_{t=0}^{T_0-1}\|\mathbf{L}^{(t)}\|$ can be bounded by a constant for any $T_0 \leq T$. From equation 12, we have

$$
\mathbf{e}_{\boldsymbol{\beta}}^{(T_0)} = \prod_{t=0}^{T_0-1}\mathbf{L}^{(t)}\mathbf{e}_{\boldsymbol{\beta}}^{(0)} - \frac{\alpha}{n}\sum_{k=0}^{T_0-1}\prod_{t=k+1}^{T_0-1}\mathbf{L}^{(t)}\left[\boldsymbol{\Theta}^{(k)\top}\mathbf{Z}^\top\mathbf{Z}\mathbf{e}_{\boldsymbol{\Theta}}^{(k)}\hat{\boldsymbol{\beta}} + \boldsymbol{\Theta}^{(k)\top}\mathbf{Z}^\top\mathbf{r}\right] + \alpha\sum_{k=0}^{T_0-1}\prod_{t=k+1}^{T_0-1}\mathbf{L}^{(t)}\boldsymbol{\nu}^{(k)}.
$$

Then

$$
\begin{aligned}
\|\mathbf{e}_{\boldsymbol{\beta}}^{(T_0)}\| &\leq \left(\prod_{t=0}^{T_0-1}\|\mathbf{L}^{(t)}\|\right)\|\mathbf{e}_{\boldsymbol{\beta}}^{(0)}\| + \frac{\alpha(\|\hat{\boldsymbol{\Theta}}\|+\varepsilon)\|\mathbf{Z}^\top\mathbf{Z}\|\|\hat{\boldsymbol{\beta}}\|}{n}\sum_{k=0}^{T_0-1}\prod_{t=k+1}^{T_0-1}\|\mathbf{L}^{(t)}\|\|\mathbf{e}_{\boldsymbol{\Theta}}^{(k)}\| \\
&\quad + \frac{\alpha(\|\hat{\boldsymbol{\Theta}}\|+\varepsilon)\|\mathbf{Z}^\top\mathbf{r}\|}{n}\sum_{k=0}^{T_0-1}\prod_{t=k+1}^{T_0-1}\|\mathbf{L}^{(t)}\| + \frac{\alpha}{1-\kappa_{\boldsymbol{\beta}}(\tau)}\|\boldsymbol{\nu}\| \\
&\lesssim \|\hat{\boldsymbol{\beta}}\| + \frac{(\|\hat{\boldsymbol{\Theta}}\|+\varepsilon)\|\mathbf{Z}^\top\mathbf{Z}\|\|\hat{\boldsymbol{\beta}}\|}{n}\sum_{k=0}^{T_0-1}\|\mathbf{e}_{\boldsymbol{\Theta}}^{(k)}\| + \frac{(\|\hat{\boldsymbol{\Theta}}\|+\varepsilon)\|\mathbf{Z}^\top\mathbf{r}\|}{n}T_0 + \lambda_2\left(\sqrt{p}+\sqrt{\tau}\right) \\
&\lesssim 1 + (1+\varepsilon)T_0\max_{0\leq k\leq T_0-1}\|\mathbf{e}_{\boldsymbol{\Theta}}^{(k)}\| + \frac{\sqrt{pq}(\tau+\log(pq))}{\sqrt{n}}\,(1+\varepsilon)\,T_0 + \lambda_2\left(\sqrt{p}+\sqrt{\tau}\right).
\end{aligned}
\tag{17}
$$

Now, it remains to determine the values of $\varepsilon, T_0, T$, and the bound for $\max_{0\leq k\leq T_0-1}\|\mathbf{e}_{\boldsymbol{\Theta}}^{(k)}\|$.

From Lemma D.4, where we take $\lambda_1 := \frac{2\gamma_1}{n}\sqrt{\frac{T}{\rho_1}}$, with probability at least $1 - 3e^{-\tau}$, we have $E_{T_0,T} = \{\|\mathbf{e}_{\boldsymbol{\Theta}}^{(k)}\| \leq \varepsilon, \forall T_0 \leq k < T\}$ holds[5], where

$$
\begin{aligned}
\varepsilon &:= \kappa_{\boldsymbol{\Theta}}(\tau)^{T_0}\|\hat{\boldsymbol{\Theta}}\| + \frac{\eta\lambda_1}{\sqrt{1-\kappa_{\boldsymbol{\Theta}}(\tau)^2}}\left(\sqrt{pq}+\sqrt{2p\left(\log(p)+\tau\right)}\right) \\
&= \kappa_{\boldsymbol{\Theta}}(\tau)^{T_0}\|\hat{\boldsymbol{\Theta}}\| + \frac{2\eta\gamma_1}{n\sqrt{1-\kappa_{\boldsymbol{\Theta}}(\tau)^2}}\sqrt{\frac{T}{\rho_1}}\left(\sqrt{pq}+\sqrt{2p\left(\log(p)+\tau\right)}\right) \\
&\lesssim \kappa_{\boldsymbol{\Theta}}(\tau)^{T_0} + \mu(\tau),
\end{aligned}
\tag{18}
$$

where

$$
\delta(\tau) := \frac{C_0\sigma_z^2(\sqrt{q}+\sqrt{\tau})}{\sqrt{n}},
$$

$$
\kappa_{\boldsymbol{\Theta}}(\tau) := \max\left\{\left|1-\eta(1-\delta(\tau))^2\right|, \left|1-\eta(1+\delta(\tau))^2\right|\right\},
$$

$$
\mu(\tau) := \lambda_1\left(\sqrt{pq}+\sqrt{p\left(\log(p)+\tau\right)}\right).
$$

---

[5]A rigorous analysis requires setting $\tau := \tau + \log(T)$ to account for the union bound. However, under condition equation 4, $\log(T)$ grows slower than any positive power of $n$, thus we omit this term. Similar argument applies to later analysis.

Similarly, we have with probability at least $1 - 3e^{-\tau}$,

$$
\begin{aligned}
\max_{0 \le k \le T_0 - 1} \|\mathbf{e}_{\boldsymbol{\Theta}}^{(k)}\| &\le \|\hat{\boldsymbol{\Theta}}\| + \frac{\eta \lambda_1}{\sqrt{1 - \kappa_{\boldsymbol{\Theta}}(\tau)^2}} \left( \sqrt{pq} + \sqrt{2p(\log(p) + \tau)} \right) \\
&= \varepsilon + \left(1 - \kappa_{\boldsymbol{\Theta}}(\tau)^{T_0}\right) \|\hat{\boldsymbol{\Theta}}\| \\
&\lesssim 1 + \mu(\tau).
\end{aligned}
\tag{19}
$$

Next, we need to pick $T, T_0$ such that condition equation 13 is satisfied:

$$
\varepsilon \le \sqrt{\|\hat{\boldsymbol{\Theta}}\|^2 + \frac{\gamma(\tau)}{2(1 + \delta(\tau))^2}} - \|\hat{\boldsymbol{\Theta}}\| := \bar{\varepsilon}.
\tag{20}
$$

This can be done by setting

$$
\kappa_{\boldsymbol{\Theta}}(\tau)^{T_0} \|\hat{\boldsymbol{\Theta}}\| \le \frac{\bar{\varepsilon}}{2},
$$

$$
\frac{2\eta \gamma_1}{n \sqrt{1 - \kappa_{\boldsymbol{\Theta}}(\tau)^2}} \sqrt{\frac{T}{\rho_1}} \left( \sqrt{pq} + \sqrt{2p(\log(p) + \tau)} \right) \le \frac{\bar{\varepsilon}}{2},
\tag{21}
$$

where from Lemma D.1, when $n \ge \left(\sqrt{q} + \sqrt{\tau}\right)^3 \max\{\frac{1}{\sqrt{\rho_1}}, \frac{1}{\sqrt{\rho_2}}\}$, we set $\gamma_1 = c_1(\sqrt{q} + \sqrt{\tau + \log(nT)})^2$. We take

$$
T_0 \ge \left\lceil \log_{\kappa_{\boldsymbol{\Theta}}(\tau)} \left( \frac{\bar{\varepsilon}}{2\|\hat{\boldsymbol{\Theta}}\|} \right) \right\rceil = \left\lceil \log_{\kappa_{\boldsymbol{\Theta}}(\tau)} \left( \frac{\sqrt{1 + \frac{\gamma(\tau)}{2(1+\delta(\tau))^2 \|\hat{\boldsymbol{\Theta}}\|^2}} - 1}{2} \right) \right\rceil := t_0(n),
\tag{22}
$$

$$
T \lesssim \frac{\rho_1 n^{2 - \epsilon}}{R(\tau)^2},
\tag{23}
$$

where $\epsilon > 0$ is a small constant to guarantee equation 21 converges to 0 as $n \to \infty$, and

$$
\begin{aligned}
R(\tau) &:= (\sqrt{q} + \sqrt{\tau})^2 (\sqrt{pq} + \sqrt{p(\log(p) + \tau)}) \\
&\lesssim \sqrt{p}(\sqrt{q} + \sqrt{\tau})^3.
\end{aligned}
\tag{24}
$$

Plugging $T$ and $\gamma_1$ into $\mu(\tau)$, we have

$$
\begin{aligned}
\mu(\tau) &= \lambda_1 \left( \sqrt{pq} + \sqrt{p(\log(p) + \tau)} \right) \\
&= \frac{2\gamma_1}{n} \sqrt{\frac{T}{\rho_1}} \left( \sqrt{pq} + \sqrt{p(\log(p) + \tau)} \right) \\
&\lesssim \frac{R(\tau)}{\sqrt{\rho_1}} \frac{\sqrt{T}}{n}
\end{aligned}
$$

So when T satisfies condition equation 21 and $n$ satisfies condition equation 6, we have $\mu(\tau) \lesssim 1$, and the bounds equation 18 equation 19 can be bounded by constants:

$$
\varepsilon \lesssim 1, \quad \max_{0 \le k \le T_0 - 1} \|\mathbf{e}_{\boldsymbol{\Theta}}^{(k)}\| \lesssim 1.
\tag{25}
$$

In equation 22, we have $t_0(n) \to \log_{1-\eta} \left( \frac{\sqrt{1 + \frac{\sigma_{\min}(\boldsymbol{\Theta})^2}{2\|\boldsymbol{\Theta}\|^2}} - 1}{2} \right)$. So $t_0(n)$ is upper bounded by a constant integer $C_2$. With $T_0 = C_2$, plug in equation 25 into equation 17, we have

$$
\begin{aligned}
\|\mathbf{e}_{\boldsymbol{\beta}}^{(C_2)}\| &\lesssim 1 + (1 + \varepsilon) T_0 \max_{0 \le k \le T_0 - 1} \|\mathbf{e}_{\boldsymbol{\Theta}}^{(k)}\| + \frac{\sqrt{pq}(\tau + \log(pq))}{\sqrt{n}} (1 + \varepsilon) T_0 + \lambda_2 \left( \sqrt{p} + \sqrt{\tau} \right) \\
&\lesssim 1 + C_2 \left( 1 + \frac{\sqrt{pq}(\tau + \log(pq))}{\sqrt{n}} \right) + \lambda_2 \left( \sqrt{p} + \sqrt{\tau} \right) \\
&\lesssim 1 + \lambda_2 \left( \sqrt{p} + \sqrt{\tau} \right).
\end{aligned}
\tag{26}
$$

We further take $\tilde{T}_0 := \max\{\frac{T}{2}.C_2\}$. Note that from equation 16, the bound of $\|\mathbf{e}_{\boldsymbol{\beta}}^{(T)}\|$ will always decrease after $T > T_0 := C_2$. Hence, the bound equation 26 still holds for $\tilde{T}_0$:

$$\|\mathbf{e}_{\boldsymbol{\beta}}^{(\tilde{T}_0)}\| \lesssim 1 + \lambda_2 \left(\sqrt{p} + \sqrt{\tau}\right). \tag{27}$$

Plug in equation 27 into equation 16, we have the final bound:

$$\|\mathbf{e}_{\boldsymbol{\beta}}^{(T)}\| \lesssim \kappa_{\boldsymbol{\beta}}(\tau)^{T-\tilde{T}_0}\|\mathbf{e}_{\boldsymbol{\beta}}^{(\tilde{T}_0)}\| + \varepsilon(1+\varepsilon) + \frac{\sqrt{pq}(\tau + \log(pq))}{\sqrt{n}}\left(1+\varepsilon\right) + \lambda_2\left(\sqrt{p} + \sqrt{\tau}\right)$$

$$\lesssim \kappa_{\boldsymbol{\beta}}(\tau)^{\frac{T}{2}}\left(1 + \lambda_2\left(\sqrt{p} + \sqrt{\tau}\right)\right) + \left(\kappa_{\boldsymbol{\Theta}}(\tau)^{\frac{T}{2}} + \mu(\tau)\right) + \frac{\sqrt{pq}(\tau + \log(pq))}{\sqrt{n}} + \lambda_2\left(\sqrt{p} + \sqrt{\tau}\right)$$

$$\lesssim \kappa_{\boldsymbol{\beta}}(\tau)^{\frac{T}{2}} + \kappa_{\boldsymbol{\Theta}}(\tau)^{\frac{T}{2}} + \mu(\tau) + \lambda_2\left(\sqrt{p} + \sqrt{\tau}\right) + \frac{\sqrt{pq}(\tau + \log(pq))}{\sqrt{n}}, \tag{28}$$

where $\mu(\tau) \lesssim \frac{R(\tau)}{\sqrt{\rho_1}}\frac{\sqrt{T}}{n}$, $\lambda_2 = \frac{2\gamma_2}{n}\sqrt{\frac{T}{\rho_2}}$. From Lemma D.1, we take $\gamma_2 = c_2\left(\sqrt{q} + \sqrt{\tau + \log(nT)}\right)^2$. Continue on equation 28, we have

$$\|\mathbf{e}_{\boldsymbol{\beta}}^{(T)}\| \lesssim \underbrace{\kappa_{\boldsymbol{\beta}}(\tau)^{\frac{T}{2}}}_{(i)} + \underbrace{\kappa_{\boldsymbol{\Theta}}(\tau)^{\frac{T}{2}} + \frac{R(\tau)}{\sqrt{\rho_1}}\frac{\sqrt{T}}{n}}_{(ii)} + \underbrace{\frac{R(\tau)}{\sqrt{\rho_2}}\frac{\sqrt{T}}{n}}_{(iii)} + \underbrace{\frac{\sqrt{pq}(\tau + \log(pq))}{\sqrt{n}}}_{(iv)}, \tag{29}$$

which concludes the proof. The error bound equation 29 consists of four terms: (i) the effect of shrinkage factor $\kappa_{\boldsymbol{\beta}}(\tau)$, (ii) the estimation error from $e_{\boldsymbol{\Theta}}^{(t)} := \boldsymbol{\Theta}^{(t)} - \hat{\boldsymbol{\Theta}}$, (iii) the error from additive noise $\boldsymbol{\nu}^{(t)}$, and (iv) the random residual error from $\mathbf{r} := \mathbf{Z}\hat{\boldsymbol{\Theta}}\hat{\boldsymbol{\beta}} - \mathbf{Y}$. $\qquad\square$

## D SUPPORTING LEMMAS

In this section, we collect the supporting lemmas that were used in the proof of the main theorem. Throughout the proof, we suppose that Assumption 1 and Assumption 2 hold. Unless otherwise specified, we assume the learning rates $\alpha, \eta$ satisfy condition equation 3, with parameters chosen according to equation 4, and sample size $n$ satisfies condition equation 6

**Lemma D.1** (No clipping condition). Under Assumption 2, if

$$\gamma_1 \gtrsim \left(\sqrt{q} + \sqrt{\tau + \log(nT)}\right)^2,$$

$$\gamma_2 \gtrsim \left(\sqrt{q} + \sqrt{\tau + \log(nT)}\right)^2,$$

learning rates $\alpha, \eta$ satisfy condition equation 3, and $n$ satisfies following condition

$$n = \Omega\left(\left(\sqrt{q} + \sqrt{\tau}\right)^3 \frac{\sqrt{T}}{\sqrt{\min(\rho_1, \rho_2)}}\right)$$

then the Algorithm 1 clips no gradients with probability at least $1 - \tilde{c}e^{-\tau}$.

The proof of Lemma D.1 is in Appendix E.1.

**Lemma D.2** (High probability bound of sub-Gaussian random matrices). Suppose $\mathbf{Z}$ is an $n \times q$ matrix whose rows $\mathbf{Z}_i$ are independent mean-zero sub-Gaussian isotropic random vectors with sub-Gaussian norm $\|\mathbf{Z}_i\|_{\psi_2} \leq \sigma_2$ for all $i = 1, \ldots, n$. Then, for any $\tau > 0$, we have with probability at least $1 - 2e^{-\tau}$,

$$\sqrt{n}\left(1 - \delta(\tau)\right) \leq \sigma_{\min}(\mathbf{Z}) \leq \sigma_{\max}(\mathbf{Z}) \leq \sqrt{n}\left(1 + \delta(\tau)\right),$$

where $\delta(\tau) := \frac{C_0\sigma_z^2(\sqrt{q}+\sqrt{\tau})}{\sqrt{n}}$. When $n \geq C_0^2\sigma_z^4\left(\sqrt{q} + \sqrt{\tau}\right)^2$, we further have

$$n(1 - \delta(\tau))^2 \leq \lambda_{\min}\left(\mathbf{Z}^\top\mathbf{Z}\right) \leq \lambda_{\max}\left(\mathbf{Z}^\top\mathbf{Z}\right) \leq n(1 + \delta(\tau))^2,$$

where $C_0$ is a universal constant, $\sigma_{\min}(\cdot)$, $\sigma_{\max}(\cdot)$ denote the minimum and maximum singular values of a matrix, $\lambda_{\min}(\cdot)$, $\lambda_{\max}(\cdot)$ denote the minimum and maximum eigenvalues of a matrix, respectively.

The proof of Lemma D.2 is in Appendix E.2.

**Lemma D.3** (High probability bound for the product of sub-Gaussian random matrices). Let $\mathbf{Z}$ be an $n \times q$ matrix whose rows $\mathbf{Z}_i$ are independent mean-zero sub-Gaussian random vectors with sub-Gaussian norm $\|\mathbf{Z}_i\|_{\psi_2} \leq \sigma_z$ for all $i = 1, \ldots, n$. Let $\boldsymbol{\mathcal{E}}_2$ be an $n \times p$ matrix whose rows $\boldsymbol{\mathcal{E}}_{2,i}$ are independent mean-zero sub-Gaussian random vectors with sub-Gaussian norm $\|\boldsymbol{\mathcal{E}}_{2,i}\|_{\psi_2} \leq \sigma_2$ for all $i = 1, \ldots, n$. Then, for any $\tau > 0$, we have with probability at least $1 - e^{-\tau}$,

$$\|\mathbf{Z}^\top \boldsymbol{\mathcal{E}}_2\| \leq c_0 \sigma_z \sigma_2 \sqrt{npq} \left(\tau + \log(2pq)\right).$$

The proof of Lemma D.3 is in Appendix E.3.

**Lemma D.4** (High probability bound of additive noise). Let $\mathbf{e}_{\boldsymbol{\Theta}}^{(t)} = \left(\mathbf{I} - \frac{\eta}{n}\mathbf{Z}^\top \mathbf{Z}\right)^t \mathbf{e}_{\boldsymbol{\Theta}}^{(0)} + \mathbf{N}^{(t-1)}$, where $\mathbf{N}^{(t)} := \sum_{i=0}^{t} \eta \left(\mathbf{I} - \frac{\eta}{n}\mathbf{Z}^\top \mathbf{Z}\right)^{t-i} \boldsymbol{\Xi}^{(i)}$, and $\boldsymbol{\Xi}^{(i)}$ are generated from Algorithm 1. Suppose the learning rate $\eta$ satisfies the following condition:

$$0 < \eta < \frac{2}{\left(1 + \delta(\tau)\right)^2},$$

where $\delta(\tau) := \frac{C_0 \sigma_z^2 (\sqrt{q} + \sqrt{\tau})}{\sqrt{n}}$. When $n \geq C_0^2 \sigma_z^4 \left(\sqrt{q} + \sqrt{\tau}\right)^2$, with probability at least $1 - 3e^{-\tau}$, we have

$$\|\mathbf{N}^{(t)}\| \leq \frac{\eta \lambda_1}{\sqrt{1 - \kappa_{\boldsymbol{\Theta}}^2(\tau)}} \left(\sqrt{pq} + \sqrt{2p\left(\log(p) + \tau\right)}\right),$$

and

$$\|\mathbf{e}_{\boldsymbol{\Theta}}^{(t)}\| \leq \kappa_{\boldsymbol{\Theta}}^t(\tau)\|\mathbf{e}_{\boldsymbol{\Theta}}^{(0)}\| + \frac{\eta \lambda_1}{\sqrt{1 - \kappa_{\boldsymbol{\Theta}}^2(\tau)}} \left(\sqrt{pq} + \sqrt{2p\left(\log(p) + \tau\right)}\right),$$

where $\kappa_{\boldsymbol{\Theta}}(\tau) := \max\left\{\left|1 - \eta(1 - \frac{C_0 \sigma_z^2 (\sqrt{q} + \sqrt{\tau})}{\sqrt{n}})^2\right|, \left|1 - \eta(1 + \frac{C_0 \sigma_z^2 (\sqrt{q} + \sqrt{\tau})}{\sqrt{n}})^2\right|\right\} < 1$.

The proof of Lemma D.4 is in Appendix E.4.

**Lemma D.5.** Let $\boldsymbol{\Psi} := \hat{\boldsymbol{\Theta}} - \boldsymbol{\Theta} = (\mathbf{Z}^\top \mathbf{Z})^{-1}\mathbf{Z}^\top \boldsymbol{\mathcal{E}}_2$. When $n \geq C_0^2 \sigma_z^4 (\sqrt{q} + \sqrt{\tau})^2$, we have with probability at least $1 - 3e^{-\tau}$,

$$\|\boldsymbol{\Psi}\| \leq \frac{c_0 \sigma_z \sigma_2 \sqrt{pq}\left(\tau + \log(2pq)\right)}{\sqrt{n}\left(1 - \delta(\tau)\right)^2},$$

where $\delta(\tau) := \frac{C_0 \sigma_z^2 (\sqrt{q} + \sqrt{\tau})}{\sqrt{n}}$, $C_0, c_0$ are absolute constants.

The proof of Lemma D.5 is in Appendix E.5.

**Lemma D.6.** Suppose Assumption 2 holds. Let $\hat{\mathbf{H}} := \frac{1}{n}\hat{\boldsymbol{\Theta}}^\top \mathbf{Z}^\top \mathbf{Z}\hat{\boldsymbol{\Theta}}$. When $n \geq C_1 pq(\tau + \log(pq))^2$, the following inequalities hold with probability at least $1 - 3e^{-\tau}$:

$$\lambda_{\min}(\hat{\mathbf{H}}) \geq (1 - \delta(\tau))^2 \left(\sigma_{\min}(\boldsymbol{\Theta}) - \frac{c_0 \sigma_z \sigma_2 \sqrt{pq}\left(\tau + \log(2pq)\right)}{\sqrt{n}\left(1 - \delta(\tau)\right)^2}\right)^2$$

$$\lambda_{\max}(\hat{\mathbf{H}}) \leq (1 + \delta(\tau))^2 \left(\|\boldsymbol{\Theta}\| + \frac{c_0 \sigma_z \sigma_2 \sqrt{pq}\left(\tau + \log(2pq)\right)}{\sqrt{n}\left(1 - \delta(\tau)\right)^2}\right)^2$$

The proof of Lemma D.6 is in Appendix E.6.

**Lemma D.7.** Suppose Assumption 2 holds. When $n \geq C_1 pq(\tau + \log(pq))^2$, we have the following inequality holds with probability at least $1 - 4e^{-\tau}$:

$$\|\hat{\boldsymbol{\beta}} - \boldsymbol{\beta}\| \leq \mathcal{O}\left(\frac{\sqrt{q}\left(\tau + \log(q)\right)}{\sqrt{n}}\right).$$

The proof of Lemma D.7 is in Appendix E.7.

**Lemma D.8.** Let $\mathbf{r} := \mathbf{Z}\hat{\boldsymbol{\Theta}}\hat{\boldsymbol{\beta}} - \mathbf{Y}$. For any fixed $\tau$, when $n \geq C_1 pq(\tau + \log(pq))^2$, with probability at least $1 - 3e^{-\tau}$, we have

$$\|\mathbf{Z}^\top \mathbf{r}\| \leq \mathcal{O}\left(\sqrt{npq}\,(\tau + \log(pq))\right).$$

The proof of Lemma D.8 is in Appendix E.8.

**Lemma D.9.** Let $\mathbf{L}^{(t)} := \mathbf{I} - \frac{\alpha}{n}\boldsymbol{\Theta}^{(t)\top}\mathbf{Z}^\top\mathbf{Z}\boldsymbol{\Theta}^{(t)}$. We have with probability $1 - \tilde{c}e^{-\tau}$, for any $0 < T_0 \leq T$

$$\prod_{t=0}^{T_0-1} \|\mathbf{L}^{(t)}\| \lesssim 1.$$

The proof of Lemma D.9 is in Appendix E.9.

# E    PROOF OF SUPPORTING LEMMAS

## E.1    PROOF OF LEMMA D.1

*Proof.* Consider non-clipping version of Algorihm 1. Denote $\mathbf{e}_{\boldsymbol{\Theta}}^{(t)} := \boldsymbol{\Theta}^{(t)} - \hat{\boldsymbol{\Theta}}$ and $\mathbf{e}_{\boldsymbol{\beta}}^{(t)} := \boldsymbol{\beta}^{(t)} - \hat{\boldsymbol{\beta}}$. For $t = 0, \ldots, T-1$, we have

$$
\begin{aligned}
\mathbf{e}_{\boldsymbol{\Theta}}^{(t+1)} &= \mathbf{e}_{\boldsymbol{\Theta}}^{(t)} - \frac{\eta}{n}\mathbf{Z}^\top\left(\mathbf{Z}\boldsymbol{\Theta}^{(t)} - \mathbf{X}\right) + \eta\boldsymbol{\Xi}^{(t)} \\
&= \left(\mathbf{I} - \frac{\eta}{n}\mathbf{Z}^\top\mathbf{Z}\right)\mathbf{e}_{\boldsymbol{\Theta}}^{(t)} + \frac{\eta}{n}\mathbf{Z}^\top\left(\mathbf{X} - \mathbf{Z}\hat{\boldsymbol{\Theta}}\right) + \eta\boldsymbol{\Xi}^{(t)},
\end{aligned}
\tag{30}
$$

and

$$
\begin{aligned}
\mathbf{e}_{\boldsymbol{\beta}}^{(t+1)} &= \mathbf{e}_{\boldsymbol{\beta}}^{(t)} - \frac{\alpha}{n}\boldsymbol{\Theta}^{(t)}\mathbf{Z}^\top\left(\mathbf{Z}\boldsymbol{\Theta}^{(t)}\boldsymbol{\beta}^{(t)} - \mathbf{Y}\right) + \alpha\boldsymbol{\nu}^{(t)} \\
&= \left(\mathbf{I} - \frac{\alpha}{n}\boldsymbol{\Theta}^{(t)\top}\mathbf{Z}^\top\mathbf{Z}\boldsymbol{\Theta}^{(t)}\right)\mathbf{e}_{\boldsymbol{\beta}}^{(t)} + \frac{\alpha}{n}\left[\boldsymbol{\Theta}^{(t)\top}\mathbf{Z}^\top\mathbf{Y} - \boldsymbol{\Theta}^{(t)\top}\mathbf{Z}^\top\mathbf{Z}\boldsymbol{\Theta}^{(t)}\hat{\boldsymbol{\beta}}\right] + \alpha\boldsymbol{\nu}^{(t)} \\
&= \left(\mathbf{I} - \frac{\alpha}{n}\boldsymbol{\Theta}^{(t)\top}\mathbf{Z}^\top\mathbf{Z}\boldsymbol{\Theta}^{(t)}\right)\mathbf{e}_{\boldsymbol{\beta}}^{(t)} + \frac{\alpha}{n}\boldsymbol{\Theta}^{(t)\top}\mathbf{Z}^\top\left(\mathbf{Y} - \mathbf{Z}\boldsymbol{\Theta}^{(t)}\hat{\boldsymbol{\beta}}\right) + \alpha\boldsymbol{\nu}^{(t)} \\
&= \left(\mathbf{I} - \frac{\alpha}{n}\boldsymbol{\Theta}^{(t)\top}\mathbf{Z}^\top\mathbf{Z}\boldsymbol{\Theta}^{(t)}\right)\mathbf{e}_{\boldsymbol{\beta}}^{(t)} - \frac{\alpha}{n}\boldsymbol{\Theta}^{(t)\top}\mathbf{Z}^\top\left(\mathbf{Z}\left(\boldsymbol{\Theta}^{(t)} - \hat{\boldsymbol{\Theta}}\right)\hat{\boldsymbol{\beta}}\right) - \frac{\alpha}{n}\left(\boldsymbol{\Theta}^{(t)\top}\mathbf{Z}^\top\left(\mathbf{Z}\hat{\boldsymbol{\Theta}}\hat{\boldsymbol{\beta}} - \mathbf{Y}\right)\right) + \alpha\boldsymbol{\nu}^{(t)} \\
&:= \mathbf{L}^{(t)}\mathbf{e}_{\boldsymbol{\beta}}^{(t)} - \frac{\alpha}{n}\boldsymbol{\Theta}^{(t)\top}\mathbf{Z}^\top\mathbf{Z}\mathbf{e}_{\boldsymbol{\Theta}}^{(t)}\hat{\boldsymbol{\beta}} - \frac{\alpha}{n}\boldsymbol{\Theta}^{(t)\top}\mathbf{Z}^\top\mathbf{r} + \alpha\boldsymbol{\nu}^{(t)},
\end{aligned}
\tag{31}
$$

where $\mathbf{L}^{(i)} := \left(\mathbf{I} - \frac{\alpha}{n}\boldsymbol{\Theta}^{(i)\top}\mathbf{Z}^\top\mathbf{Z}\boldsymbol{\Theta}^{(i)}\right)$, $\mathbf{r} := \mathbf{Z}\hat{\boldsymbol{\Theta}}\hat{\boldsymbol{\beta}} - \mathbf{Y}$. By iteratively applying recursion formulas equation 30equation 31 until $t = 0$, with $\boldsymbol{\Theta}^{(0)} = \mathbf{0}_{q \times p}$ and $\boldsymbol{\beta}^{(0)} = \mathbf{0}_p$, we have

$$\boldsymbol{\Theta}^{(t)} = \hat{\boldsymbol{\Theta}} - \left(\mathbf{I} - \frac{\eta}{n}\mathbf{Z}^\top\mathbf{Z}\right)^t\hat{\boldsymbol{\Theta}} + \sum_{i=0}^{t-1}\eta\left(\mathbf{I} - \frac{\eta}{n}\mathbf{Z}^\top\mathbf{Z}\right)^{t-1-i}\boldsymbol{\Xi}^{(i)},$$

$$\boldsymbol{\beta}^{(t)} = \hat{\boldsymbol{\beta}} - \prod_{i=0}^{t-1}\mathbf{L}^{(i)}\hat{\boldsymbol{\beta}} - \frac{\alpha}{n}\sum_{i=0}^{t-1}\prod_{j=i+1}^{t-1}\mathbf{L}^{(j)}\left[\boldsymbol{\Theta}^{(i)\top}\mathbf{Z}^\top\mathbf{Z}\mathbf{e}_{\boldsymbol{\Theta}}^{(i)}\hat{\boldsymbol{\beta}} + \boldsymbol{\Theta}^{(i)\top}\mathbf{Z}^\top\mathbf{r}\right] + \sum_{i=0}^{t-1}\alpha\prod_{j=i+1}^{t-1}\mathbf{L}^{(j)}\boldsymbol{\nu}^{(i)}.$$

The gradients at step $t$ are given by

$$g_i^{\boldsymbol{\Theta}}(t) := \mathbf{z}_i\left(\mathbf{z}_i^\top\boldsymbol{\Theta}^{(t)} - \mathbf{x}_i^\top\right),$$

$$g_i^{\boldsymbol{\beta}}(t) := \boldsymbol{\Theta}^{(t)\top}\mathbf{z}_i\left(\mathbf{z}_i^\top\boldsymbol{\Theta}^{(t)}\boldsymbol{\beta}^{(t)} - y_i\right).$$

**Bound on $g_i^{\boldsymbol{\Theta}}(t)$:**

We have

$$\left\|g_i^{\boldsymbol{\Theta}}(t)\right\| = \left\|\mathbf{z}_i\left(\mathbf{z}_i^\top\boldsymbol{\Theta}^{(t)} - \mathbf{x}_i^\top + \mathbf{z}_i^\top\boldsymbol{\Theta} - \mathbf{z}_i^\top\boldsymbol{\Theta}\right)\right\|$$

$$\begin{aligned}
&= \left\| \mathbf{z}_i \mathbf{z}_i^\top \left( \mathbf{\Theta}^{(t)} - \mathbf{\Theta} \right) - \mathbf{z}_i \left( \mathbf{x}_i^\top - \mathbf{z}_i^\top \mathbf{\Theta} \right) \right\| \\
&\leq \left\| \mathbf{z}_i \mathbf{z}_i^\top \left( \mathbf{\Theta}^{(t)} - \mathbf{\Theta} \right) \right\| + \left\| \mathbf{z}_i \left( \mathbf{x}_i^\top - \mathbf{z}_i^\top \mathbf{\Theta} \right) \right\| \\
&\leq \left\| \mathbf{z}_i \mathbf{z}_i^\top \left( \hat{\mathbf{\Theta}} - \mathbf{\Theta} - \left( \mathbf{I} - \frac{\eta}{n} \mathbf{Z}^\top \mathbf{Z} \right)^t \hat{\mathbf{\Theta}} + \sum_{j=0}^{t-1} \eta \left( \mathbf{I} - \frac{\eta}{n} \mathbf{Z}^\top \mathbf{Z} \right)^{t-1-j} \mathbf{\Xi}^{(j)} \right) \right\| + \left\| \mathbf{z}_i \left( \mathbf{x}_i^\top - \mathbf{z}_i^\top \mathbf{\Theta} \right) \right\| \\
&\leq \underbrace{\left\| \mathbf{z}_i \mathbf{z}_i^\top \left( \hat{\mathbf{\Theta}} - \mathbf{\Theta} - \left( \mathbf{I} - \frac{\eta}{n} \mathbf{Z}^\top \mathbf{Z} \right)^t \left( \hat{\mathbf{\Theta}} - \mathbf{\Theta} \right) \right) \right\|}_{(i)} + \underbrace{\left\| \mathbf{z}_i \mathbf{z}_i^\top \left( \mathbf{I} - \frac{\eta}{n} \mathbf{Z}^\top \mathbf{Z} \right)^t \mathbf{\Theta} \right\|}_{(ii)} \\
&\quad + \underbrace{\left\| \sum_{j=0}^{t-1} \eta \mathbf{z}_i \mathbf{z}_i^\top \left( \mathbf{I} - \frac{\eta}{n} \mathbf{Z}^\top \mathbf{Z} \right)^{t-1-j} \mathbf{\Xi}^{(j)} \right\|}_{(iii)} + \underbrace{\left\| \mathbf{z}_i \left( \mathbf{x}_i^\top - \mathbf{z}_i^\top \mathbf{\Theta} \right) \right\|}_{(iv)}.
\end{aligned}$$

We further have

$$\begin{aligned}
(i) &= \left\| \mathbf{z}_i \mathbf{z}_i^\top \left( \mathbf{I} - \left( \mathbf{I} - \frac{\eta}{n} \mathbf{Z}^\top \mathbf{Z} \right)^t \right) \left( \hat{\mathbf{\Theta}} - \mathbf{\Theta} \right) \right\| \\
&\leq \| \mathbf{z}_i \|^2 \left( 1 + \left\| \left( \mathbf{I} - \frac{\eta}{n} \mathbf{Z}^\top \mathbf{Z} \right)^t \right\| \right) \left\| \hat{\mathbf{\Theta}} - \mathbf{\Theta} \right\|,
\end{aligned}$$

$$\begin{aligned}
(ii) &= \left\| \mathbf{z}_i \mathbf{z}_i^\top \left( \mathbf{I} - \frac{\eta}{n} \mathbf{Z}^\top \mathbf{Z} \right)^t \mathbf{\Theta} \right\| \\
&\leq \| \mathbf{z}_i \|^2 \left\| \left( \mathbf{I} - \frac{\eta}{n} \mathbf{Z}^\top \mathbf{Z} \right)^t \right\| \| \mathbf{\Theta} \|,
\end{aligned}$$

$$\begin{aligned}
(iii) &= \left\| \sum_{j=0}^{t-1} \eta \mathbf{z}_i \mathbf{z}_i^\top \left( \mathbf{I} - \frac{\eta}{n} \mathbf{Z}^\top \mathbf{Z} \right)^{t-1-j} \mathbf{\Xi}^{(j)} \right\| \\
&\leq \eta \| \mathbf{z}_i \|^2 \left\| \sum_{j=0}^{t-1} \left( \mathbf{I} - \frac{\eta}{n} \mathbf{Z}^\top \mathbf{Z} \right)^{t-1-j} \mathbf{\Xi}^{(j)} \right\| \\
&\leq \eta \| \mathbf{z}_i \|^2 \sum_{j=0}^{t-1} \left\| \left( \mathbf{I} - \frac{\eta}{n} \mathbf{Z}^\top \mathbf{Z} \right)^{t-1-j} \right\| \left\| \mathbf{\Xi}^{(j)} \right\|,
\end{aligned}$$

$$\begin{aligned}
(iv) &= \| \mathbf{z}_i \left( \mathbf{x}_i^\top - \mathbf{z}_i^\top \mathbf{\Theta} \right) \| = \| \mathbf{z}_i \boldsymbol{\epsilon}_{2,i} \| \\
&\leq \| \mathbf{z}_i \| \| \boldsymbol{\epsilon}_{2,i} \|.
\end{aligned}$$

Under sub-Gaussian assumption on $\mathbf{z}_i$ and $\boldsymbol{\epsilon}_2$, we have with probability at least $1 - e^{-\tau}$,

$$\begin{aligned}
\| \mathbf{z}_i \| &\lesssim \sigma_z (\sqrt{q} + \sqrt{\tau}), \\
\| \boldsymbol{\epsilon}_{2,i} \| &\lesssim \sigma_2 \left( \sqrt{p} + \sqrt{\tau} \right).
\end{aligned}$$

From Lemma D.4, we have when $0 < \eta < \frac{2}{(1+\delta(\tau))^2}$ and $n \geq C_0^2 \sigma_z^4 (\sqrt{q} + \sqrt{\tau})^2$, with probability at least $1 - 2e^{-\tau}$,

$$\left\| \mathbf{I} - \frac{\eta}{n} \mathbf{Z}^\top \mathbf{Z} \right\| \leq \kappa_{\mathbf{\Theta}}(\tau) < 1.$$

From Lemma D.5, when $n \geq C_0^2 \sigma_z^4 (\sqrt{q} + \sqrt{\tau})^2$, we have with probability at least $1 - 3e^{-\tau}$,

$$\left\| \hat{\mathbf{\Theta}} - \mathbf{\Theta} \right\| \lesssim 1.$$

Additionally, by standard concentration results in random matrix theory, with probability $1 - e^{-\tau}$, we have

$$\left\| \mathbf{\Xi}^{(j)} \right\| \leq \lambda_1 \left( \sqrt{p} + \sqrt{q} + \sqrt{2(\log 2 + \tau)} \right).$$

To sum up, we have

$$
\begin{aligned}
(i) &\lesssim \sigma_z^2 (\sqrt{q} + \sqrt{\tau})^2, \\
(ii) &\lesssim \sigma_z^2 (\sqrt{q} + \sqrt{\tau})^2, \\
(iii) &\lesssim \sigma_z^2 (\sqrt{q} + \sqrt{\tau})^2 \lambda_1 \left( \sqrt{p} + \sqrt{q} + \sqrt{\tau} \right), \\
(iv) &\lesssim \sigma_z \sigma_2 (\sqrt{q} + \sqrt{\tau}) \left( \sqrt{p} + \sqrt{\tau} \right).
\end{aligned}
$$

With $\lambda_1 = \frac{2\gamma_1}{n} \sqrt{\frac{T}{\rho_1}}$, we take $\tau' = \tau + \log(nT)$ and plug everything back in the final bound, we have with probability at least $1 - \frac{c}{nT} e^{-\tau}$,

$$\left\| g_i^{\mathbf{\Theta}}(t) \right\| \lesssim \sigma_z^2 \sigma_2 (\sqrt{q} + \sqrt{\tau + \log(nT)})^2 \left( 1 + \lambda_1 \left( \sqrt{p} + \sqrt{q} + \sqrt{\tau + \log(nT)} \right) \right)$$

$$\lesssim \sigma_z^2 \sigma_2 (\sqrt{q} + \sqrt{\tau + \log(nT)})^2 \left( 1 + \frac{\gamma_1}{n} \sqrt{\frac{T}{\rho_1}} \left( \sqrt{p} + \sqrt{q} + \sqrt{\tau + \log(nT)} \right) \right).$$

We want to choose appropriate $\gamma_1$ such that $\left\| g_i^{\mathbf{\Theta}}(t) \right\| \leq \gamma_1$ with high probability, for all $i = 1, \ldots, n, t = 0, \ldots, T - 1$. Therefore, the condition for $\gamma_1$ is

$$\gamma_1 \geq \frac{\sigma_z^2 \sigma_2 (\sqrt{q} + \sqrt{\tau + \log(nT)})^2}{1 - \frac{\sigma_z^2 \sigma_2 \left( \sqrt{q} + \sqrt{\tau + \log(nT)} \right)^2}{n} \sqrt{\frac{T}{\rho_1}} \left( \sqrt{p} + \sqrt{q} + \sqrt{\tau + \log(nT)} \right)} \tag{32}$$

$$\gtrsim \left( \sqrt{q} + \sqrt{\tau + \log(nT)} \right)^2,$$

which is subject to the condition

$$n = \Omega \left( \left( \sqrt{q} + \sqrt{\tau} \right)^2 \sqrt{\frac{T}{\rho_1}} \left( \sqrt{p} + \sqrt{q} + \sqrt{\tau} \right) \right)$$

$$= \Omega \left( \left( \sqrt{q} + \sqrt{\tau} \right)^3 \sqrt{\frac{T}{\rho_1}} \right),$$

where we ignore the $\sqrt{\log(nT)}$ term since it grows slower than any positive power of $n$. Finally, taking the union bound over $i = 1, \ldots, n$ and $t = 0, \ldots, T - 1$ completes the proof.

**Bound on $g_i^{\boldsymbol{\beta}}(t)$:**

From equation 32, if we take $\gamma_1 \gtrsim \left( \sqrt{q} + \sqrt{\tau + \log(nT)} \right)^2$, with probability at least $1 - ce^{-\tau}$, $\| g_i^{\mathbf{\Theta}}(t) \| \leq \gamma_1, \forall i = 1, \ldots, n$ and $t = 0, \ldots, T - 1$. Now we analyze the gradient $g_i^{\boldsymbol{\beta}}(t)$. Under model

$$
\begin{aligned}
y_i &= \boldsymbol{\beta}^\top \boldsymbol{x}_i + \epsilon_{1,i} \\
\boldsymbol{x}_i &= \mathbf{\Theta}^\top \boldsymbol{z}_i + \boldsymbol{\epsilon}_{2,i}
\end{aligned}
$$

we have

$$
\begin{aligned}
g_i^{\boldsymbol{\beta}}(t) &= \mathbf{\Theta}^{(t)\top} \mathbf{z}_i \left( \mathbf{z}_i^\top \mathbf{\Theta}^{(t)} \boldsymbol{\beta}^{(t)} - \mathbf{z}_i^\top \mathbf{\Theta}^{(t)} \boldsymbol{\beta} + \mathbf{z}_i^\top \mathbf{\Theta}^{(t)} \boldsymbol{\beta} - y_i \right) \\
&= \mathbf{\Theta}^{(t)\top} \mathbf{z}_i \left( \mathbf{z}_i^\top \mathbf{\Theta}^{(t)} \boldsymbol{\beta}^{(t)} - \mathbf{z}_i^\top \mathbf{\Theta}^{(t)} \boldsymbol{\beta} + \mathbf{z}_i^\top \mathbf{\Theta}^{(t)} \boldsymbol{\beta} - \boldsymbol{\beta}^\top (\mathbf{\Theta}^\top \mathbf{z}_i + \boldsymbol{\epsilon}_{2,i}) - \epsilon_{1,i} \right) \\
&= \mathbf{\Theta}^{(t)\top} \mathbf{z}_i \mathbf{z}_i^\top \mathbf{\Theta}^{(t)} \left( \boldsymbol{\beta}^{(t)} - \boldsymbol{\beta} \right) + \mathbf{\Theta}^{(t)\top} \mathbf{z}_i \left( \mathbf{z}_i^\top \mathbf{\Theta}^{(t)} \boldsymbol{\beta} - \mathbf{z}_i^\top \mathbf{\Theta} \boldsymbol{\beta} \right) - \mathbf{\Theta}^{(t)\top} \mathbf{z}_i \left( \boldsymbol{\beta}^\top \boldsymbol{\epsilon}_{2i} + \epsilon_{1i} \right) \\
&= \underbrace{\mathbf{\Theta}^{(t)\top} \mathbf{z}_i \mathbf{z}_i^\top \mathbf{\Theta}^{(t)} \left( \boldsymbol{\beta}^{(t)} - \boldsymbol{\beta} \right)}_{(i)} + \underbrace{\mathbf{\Theta}^{(t)\top} \mathbf{z}_i \mathbf{z}_i^\top \left( \mathbf{\Theta}^{(t)} - \mathbf{\Theta} \right) \boldsymbol{\beta}}_{(ii)} - \underbrace{\mathbf{\Theta}^{(t)\top} \mathbf{z}_i \left( \boldsymbol{\beta}^\top \boldsymbol{\epsilon}_{2i} + \epsilon_{1i} \right)}_{(iii)}
\end{aligned}
$$

$$\tag{33}$$

Note that

$$\boldsymbol{\beta}^{(t)} - \hat{\boldsymbol{\beta}} = -\prod_{i=0}^{t-1}\mathbf{L}^{(i)}\hat{\boldsymbol{\beta}} - \frac{\alpha}{n}\sum_{i=0}^{t-1}\prod_{j=i+1}^{t-1}\mathbf{L}^{(j)}\left[\boldsymbol{\Theta}^{(i)\top}\mathbf{Z}^{\top}\mathbf{Z}\mathbf{e}_{\boldsymbol{\Theta}}^{(i)}\hat{\boldsymbol{\beta}} + \boldsymbol{\Theta}^{(i)\top}\mathbf{Z}^{\top}\mathbf{r}\right] + \sum_{i=0}^{t-1}\alpha\prod_{j=i+1}^{t-1}\mathbf{L}^{(j)}\boldsymbol{\nu}^{(i)}$$

$$:= \prod_{i=0}^{t-1}\mathbf{L}^{(i)}\left(\boldsymbol{\beta}^{(0)} - \hat{\boldsymbol{\beta}}\right) - \frac{\alpha}{n}\sum_{i=0}^{t-1}\prod_{j=i+1}^{t-1}\mathbf{L}^{(j)}\left[\boldsymbol{\Theta}^{(i)\top}\mathbf{Z}^{\top}\mathbf{Z}\mathbf{e}_{\boldsymbol{\Theta}}^{(i)}\hat{\boldsymbol{\beta}} + \boldsymbol{\Theta}^{(i)\top}\mathbf{Z}^{\top}\mathbf{r}\right] + \alpha\tilde{\boldsymbol{\nu}}^{(t)},$$

where $\tilde{\boldsymbol{\nu}}^{(t)} := \sum_{i=0}^{t-1}\prod_{j=i+1}^{t-1}\mathbf{L}^{(j)}\boldsymbol{\nu}^{(i)}$. Similar to equation 27, we take $T_0 := \max\{\frac{T}{2}, C_2\}$. When $t \leq T_0$, we have

$$\|\boldsymbol{\beta}^{(t)} - \hat{\boldsymbol{\beta}}\| \lesssim 1 + \tilde{\boldsymbol{\nu}}^{(t)}. \tag{34}$$

When $T_0 < t \leq T$, the error begins to shrink with $t$, so the bound equation 34 holds uniformly for all $t = 1, \ldots, T$. It remains to determine the bound for $\|\tilde{\boldsymbol{\nu}}^{(t)}\|$. Note that since $\boldsymbol{\nu}^{(i)} \sim \mathcal{N}(0, \lambda_2^2\mathbf{I}_p^2)$, we have with probability $1 - e^{-\tau}$,

$$\|\boldsymbol{\nu}^{(i)}\| \lesssim \lambda_2\left(\sqrt{p} + \sqrt{\tau + \log(T)}\right), \forall i = 0, \ldots, T - 1.$$

**Case 1:** $t \leq T_0$. In this case, we have

$$\|\tilde{\boldsymbol{\nu}}^{(t)}\| = \|\sum_{i=0}^{t-1}\prod_{j=i+1}^{t-1}\mathbf{L}^{(j)}\boldsymbol{\nu}^{(i)}\|$$

$$\leq \sum_{i=0}^{T_0-1}\prod_{j=i+1}^{T_0-1}\|\mathbf{L}^{(j)}\|\|\boldsymbol{\nu}^{(i)}\|$$

$$\lesssim \lambda_2\left(\sqrt{p} + \sqrt{\tau + \log(T)}\right)\sum_{i=0}^{T_0-1}\prod_{j=i+1}^{T_0-1}\|\mathbf{L}^{(j)}\|$$

$$\lesssim \lambda_2 T_0\left(\sqrt{p} + \sqrt{\tau + \log(T)}\right)$$

where the last line follows from the fact that $\prod_{j=i+1}^{T_0-1}\|\mathbf{L}^{(j)}\|$ can be bounded by constant, following from Lemma D.9.

**Case 2:** $t > T_0$. We have

$$\tilde{\boldsymbol{\nu}}^{(t)} = \sum_{i=0}^{t-1}\prod_{j=i+1}^{t-1}\mathbf{L}^{(j)}\boldsymbol{\nu}^{(i)} = \sum_{i=0}^{T_0-1}\prod_{j=i+1}^{t-1}\mathbf{L}^{(j)}\boldsymbol{\nu}^{(i)} + \sum_{i=T_0}^{t-1}\prod_{j=i+1}^{t-1}\mathbf{L}^{(j)}\boldsymbol{\nu}^{(i)}$$

For any $j \geq T_0$, we have $\|\mathbf{L}^{(j)}\| \leq \kappa_{\boldsymbol{\beta}}(\tau) < 1$. Hence, we have

$$\|\tilde{\boldsymbol{\nu}}^{(t)}\| \leq \left\|\sum_{i=0}^{T_0-1}\prod_{j=i+1}^{t-1}\mathbf{L}^{(j)}\boldsymbol{\nu}^{(i)}\right\| + \left\|\sum_{i=T_0}^{t-1}\prod_{j=i+1}^{t-1}\mathbf{L}^{(j)}\boldsymbol{\nu}^{(i)}\right\|$$

$$\leq \left\|\sum_{i=0}^{T_0-1}\prod_{j=i+1}^{T_0-1}\mathbf{L}^{(j)}\prod_{j'=T_0}^{t-1}\mathbf{L}^{(j')}\boldsymbol{\nu}^{(i)}\right\| + \left\|\sum_{i=T_0}^{t-1}\prod_{j=i+1}^{t-1}\mathbf{L}^{(j)}\boldsymbol{\nu}^{(i)}\right\|$$

$$\leq \left\|\sum_{i=0}^{T_0-1}\prod_{j=i+1}^{T_0-1}\mathbf{L}^{(j)}\boldsymbol{\nu}^{(i)}\right\| + \left\|\sum_{i=T_0}^{t-1}\prod_{j=i+1}^{t-1}\mathbf{L}^{(j)}\boldsymbol{\nu}^{(i)}\right\|$$

$$\lesssim \lambda_2 T_0\left(\sqrt{p} + \sqrt{\tau + \log(T)}\right) + \sum_{i=T_0}^{t-1}\kappa_{\boldsymbol{\beta}}(\tau)^{t-1-i}\lambda_2\left(\sqrt{p} + \sqrt{\tau + \log(T)}\right)$$

$$\lesssim \lambda_2 T_0\left(\sqrt{p} + \sqrt{\tau + \log(T)}\right)$$

So we have the following uniform bound:

$$\|\boldsymbol{\beta}^{(t)} - \boldsymbol{\beta}\| \lesssim 1 + \lambda_2 T_0 \left( \sqrt{p} + \sqrt{\tau + \log(T)} \right)$$

$$\lesssim 1 + \frac{\gamma_2 \sqrt{T}}{n\sqrt{\rho_2}} \left( \sqrt{p} + \sqrt{\tau + \log(T)} \right), \forall t = 1, \dots, T,$$

where we ignore the error from $\|\hat{\boldsymbol{\beta}} - \boldsymbol{\beta}\|$ as it diminishes with $n$, according to Lemma D.7. Besides, according to equation 25, we have

$$\|\boldsymbol{\Theta}^{(t)} - \boldsymbol{\Theta}\| \lesssim 1.$$

Then we have with probability $1 - c'e^{-\tau}$, for any $t = 1, \dots, T$, $i = 1, \dots, n$,

$$(i) = \left\| \boldsymbol{\Theta}^{(t)\top} \mathbf{z}_i \mathbf{z}_i^\top \boldsymbol{\Theta}^{(t)} \left( \boldsymbol{\beta}^{(t)} - \boldsymbol{\beta} \right) \right\|$$

$$\lesssim \sigma_z^2 \left( \sqrt{q} + \sqrt{\tau + \log(nT)} \right)^2 \left( 1 + \frac{\gamma_2 \sqrt{T}}{n\sqrt{\rho_2}} \left( \sqrt{p} + \sqrt{\tau + \log(T)} \right) \right),$$

$$(ii) = \left\| \boldsymbol{\Theta}^{(t)\top} \mathbf{z}_i \mathbf{z}_i^\top \left( \boldsymbol{\Theta}^{(t)} - \boldsymbol{\Theta} \right) \boldsymbol{\beta} \right\|$$

$$\lesssim \sigma_z^2 (\sqrt{q} + \sqrt{\tau + \log(nT)})^2,$$

$$(iii) = \left\| \boldsymbol{\Theta}^{(t)\top} \mathbf{z}_i \left( \boldsymbol{\beta}^\top \boldsymbol{\epsilon}_{2i} + \epsilon_{1i} \right) \right\|$$

$$\lesssim \sigma_z \tilde{\sigma} \sqrt{\tau + \log(nT)} (\sqrt{q} + \sqrt{\tau}),$$

where the last inequality follows from the term $\left( \boldsymbol{\beta}^\top \boldsymbol{\epsilon}_{2i} + \epsilon_{1i} \right)$ is zero-mean sub-Gaussian with parameter $\tilde{\sigma} := \sqrt{\sigma_2^2 \|\boldsymbol{\beta}\|^2 + \sigma_1^2}$. Plug in (i)-(iii) and equation 34 into equation 33, we have the dominating term

$$\|g_i^{\boldsymbol{\beta}}(t)\| \lesssim \sigma_z^2 \left( \sqrt{q} + \sqrt{\tau + \log(nT)} \right)^2 \left( 1 + \frac{\gamma_2 \sqrt{T}}{n\sqrt{\rho_2}} \left( \sqrt{p} + \sqrt{\tau + \log(T)} \right) \right).$$

In order to guarantee the no-clipping condition, we can take $\gamma_2$ such that

$$\sigma_z^2 \left( \sqrt{q} + \sqrt{\tau + \log(nT)} \right)^2 \left( 1 + \frac{\gamma_2 \sqrt{T}}{n\sqrt{\rho_2}} \left( \sqrt{p} + \sqrt{\tau + \log(T)} \right) \right) \le \gamma_2.$$

Solving for $\gamma_2$, we have

$$\gamma_2 \ge \frac{\sigma_z^2 (\sqrt{q} + \sqrt{\tau + \log(nT)})^2}{1 - \frac{\sigma_z^2 \left( \sqrt{q} + \sqrt{\tau + \log(nT)} \right)^2 \sqrt{T}}{n\sqrt{\rho_2}} \left( \sqrt{p} + \sqrt{\tau + \log(T)} \right)}, \tag{35}$$

which is subject to the condition

$$n = \Omega \left( (\sqrt{q} + \sqrt{\tau})^2 \frac{\sqrt{T}}{\sqrt{\rho_2}} \left( \sqrt{p} + \sqrt{\tau} \right) \right)$$

$$= \Omega \left( (\sqrt{q} + \sqrt{\tau})^3 \frac{\sqrt{T}}{\sqrt{\rho_2}} \right),$$

where we ignore the $\sqrt{\log(nT)}$ term since it grows slower than any positive power of $n$. $\square$

### E.2 PROOF OF LEMMA D.2

*Proof.* The first inequality chain follows directly from the standard concentration inequality for sub-Gaussian random matrices (see (Vershynin, 2018), Theorem 4.6.1). The second inequality chain follows from the fact that $\sigma_i(Z) = \sqrt{\lambda_i(Z^\top Z)}$ for $i = 1, \dots, q$. $\square$

### E.3 PROOF OF LEMMA D.3

*Proof.* We have the $(j, k)$-th entry of $\mathbf{Z}^\top \boldsymbol{\mathcal{E}}_2$ is given by

$$\left(\mathbf{Z}^\top \boldsymbol{\mathcal{E}}_2\right)_{jk} = \sum_{i=1}^n \mathbf{Z}_{i,j} \boldsymbol{\mathcal{E}}_{2,i,k},$$

the sub-exponential norm of this term can be bounded by

$$\| \left(\mathbf{Z}^\top \boldsymbol{\mathcal{E}}_2\right)_{jk} \|_{\psi_1} = \| \sum_{i=1}^n \mathbf{Z}_{i,j} \boldsymbol{\mathcal{E}}_{2,i,k} \|_{\psi_1} \leq \sigma_z \sigma_2 \sqrt{n}.$$

Thus we have the tail bound for each $(j, k)$:

$$\mathbb{P}\left(| \left(\mathbf{Z}^\top \boldsymbol{\mathcal{E}}_2\right)_{jk} | \geq \tau\right) \leq 2e^{-\frac{\tau}{c_0 \sigma_z \sigma_2 \sqrt{n}}}.$$

Taking the union bound over $j = 1, \ldots, p$ and $k = 1, \ldots, q$, we have

$$\mathbb{P}\left(\|\mathbf{Z}^\top \boldsymbol{\mathcal{E}}_2\| \geq \tau\right) \leq \mathbb{P}\left(\|\mathbf{Z}^\top \boldsymbol{\mathcal{E}}_2\|_{\max} \geq \frac{\tau}{\sqrt{pq}}\right) \leq 2pq e^{-\frac{\tau}{c_0 \sigma_z \sigma_2 \sqrt{n} \sqrt{pq}}}.$$

Equivalently, with probability at least $1 - e^{-\tau}$, we have

$$\|\mathbf{Z}^\top \boldsymbol{\mathcal{E}}_2\| \leq c_0 \sigma_z \sigma_2 \sqrt{npq} \left(\tau + \log(2pq)\right).$$

$\square$

### E.4 PROOF OF LEMMA D.4

*Proof.* From Lemma D.2, when $n \geq C_0^2 \sigma_z^4 \left(\sqrt{q} + \sqrt{\tau}\right)^2$, with probability at least $1 - 2e^{-\tau}$, we have

$$\lambda_{\min}\left(\frac{\mathbf{Z}^\top \mathbf{Z}}{n}\right) \geq (1 - \delta(\tau))^2,$$

$$\lambda_{\max}\left(\frac{\mathbf{Z}^\top \mathbf{Z}}{n}\right) \leq (1 + \delta(\tau))^2,$$

where $\delta(\tau) := \frac{C_0 \sigma_z^2 (\sqrt{q} + \sqrt{\tau})}{\sqrt{n}}$. When $0 < \eta < \frac{2}{(1+\delta(\tau))^2}$, we can bound the spectral radius of $\mathbf{I} - \frac{\eta}{n} \mathbf{Z}^\top \mathbf{Z}$ with probability at least $1 - 2e^{-\tau}$:

$$\rho\left(\mathbf{I} - \frac{\eta}{n} \mathbf{Z}^\top \mathbf{Z}\right) \leq \kappa_{\boldsymbol{\Theta}}(\tau) := \max\left\{\left|1 - \eta(1 - \delta(\tau))^2\right|, \left|1 - \eta(1 + \delta(\tau))^2\right|\right\} < 1,$$

where $\rho(\cdot)$ denotes the spectral radius of a matrix. If we define the event $E_{\kappa_{\boldsymbol{\Theta}}(\tau)} = \{\mathbf{Z} : \rho\left(\mathbf{I} - \frac{\eta}{n} \mathbf{Z}^\top \mathbf{Z}\right) \leq \kappa_{\boldsymbol{\Theta}}(\tau)\}$, then conditional on event $E_{\kappa_{\boldsymbol{\Theta}}(\tau)}$, we have the following holds for each column $k = 1, 2, \ldots, p$:

$$\mathbf{N}_k^{(t)} = \sum_{i=0}^t \eta\left(\mathbf{I} - \frac{\eta}{n} \mathbf{Z}^\top \mathbf{Z}\right)^{t-i} \boldsymbol{\Xi}_k^{(i)} \sim \mathcal{N}\left(0, \underbrace{\eta^2 \lambda_1^2 \left[\mathbf{I} - \left(\mathbf{I} - \frac{\eta}{n} \mathbf{Z}^\top \mathbf{Z}\right)^2\right]^{-1} \left[\mathbf{I} - \left(\mathbf{I} - \frac{\eta}{n} \mathbf{Z}^\top \mathbf{Z}\right)^{2(t+1)}\right]}_{\tilde{\boldsymbol{\Sigma}}_k}\right),$$

where

$$\|\tilde{\boldsymbol{\Sigma}}_k\| \leq \eta^2 \lambda_1^2 \left[\sum_{i=0}^t \left\|\left(\mathbf{I} - \frac{\eta}{n} \mathbf{Z}^\top \mathbf{Z}\right)^{2(t-i)}\right\|\right]$$

$$\leq \eta^2 \lambda_1^2 \sum_{i=0}^t \kappa_{\boldsymbol{\Theta}}^{2i}(\tau)$$

$$\leq \frac{\eta^2 \lambda_1^2}{1 - \kappa_{\boldsymbol{\Theta}}^2(\tau)}.$$

A standard result following Lemma 1 of (Laurent & Massart, 2000) gives the following bound holds with probability at least $1 - \frac{1}{p} e^{-\tau}$:

$$\|\mathbf{N}_k^{(t)}\| \leq \sqrt{tr(\tilde{\Sigma}_k)} + \sqrt{2\|\tilde{\Sigma}_k\| (\log(p) + \tau)}$$

$$\leq \sqrt{q\|\tilde{\Sigma}_k\|} + \sqrt{2\|\tilde{\Sigma}_k\| (\log(p) + \tau)}$$

$$\leq \frac{\eta \lambda_1}{\sqrt{1 - \kappa_{\boldsymbol{\Theta}}^2(\tau)}} \left( \sqrt{q} + \sqrt{2 (\log(p) + \tau)} \right)$$

Taking the union bound over each column $k = 1, \ldots, p$, we have the following holds with probability at least $1 - e^{-\tau}$, conditional on $E_{\kappa_{\boldsymbol{\Theta}}(\tau)}$:

$$\|\mathbf{N}^{(t)}\| \leq \frac{\eta \lambda_1}{\sqrt{1 - \kappa_{\boldsymbol{\Theta}}^2(\tau)}} \left( \sqrt{pq} + \sqrt{2p (\log(p) + \tau)} \right),$$

and

$$\|\mathbf{e}_{\boldsymbol{\Theta}}^{(t)}\| \leq \kappa_{\boldsymbol{\Theta}}^t(\tau) \|\mathbf{e}_{\boldsymbol{\Theta}}^{(0)}\| + \frac{\eta \lambda_1}{\sqrt{1 - \kappa_{\boldsymbol{\Theta}}^2(\tau)}} \left( \sqrt{pq} + \sqrt{2p (\log(p) + \tau)} \right)$$

Finally, uncondition on $E_{\kappa_{\boldsymbol{\Theta}}(\tau)}$ and take the union bound over the event $E_{\kappa_{\boldsymbol{\Theta}}(\tau)}$ gives the desired result. $\qquad\square$

### E.5 PROOF OF LEMMA D.5

*Proof.* We have

$$\|\boldsymbol{\Psi}\| = \|(\mathbf{Z}^\top \mathbf{Z})^{-1} \mathbf{Z}^\top \mathbf{E_2}\|$$
$$\leq \frac{\|\mathbf{Z}^\top \boldsymbol{\mathcal{E}}_2\|}{\sigma_{\min}^2(\mathbf{Z})} \tag{36}$$

From Lemma D.2, we have when $n \geq C_0^2 \sigma_z^4 (\sqrt{q} + \sqrt{\tau})^2$, with probability at least $1 - 2e^{-\tau}$, we have

$$\sigma_{\min}^2(\mathbf{Z}) = \lambda_{\min} \left( \mathbf{Z}^\top \mathbf{Z} \right) \geq n(1 - \delta(\tau))^2, \tag{37}$$

For the numerator, from Lemma D.3, we have with probability at least $1 - e^{-\tau}$,

$$\|\mathbf{Z}^\top \boldsymbol{\mathcal{E}}_2\| \leq c_0 \sigma_z \sigma_2 \sqrt{npq} (\tau + \log(2pq)). \tag{38}$$

Finally, plug in equation 37 and equation 38 into equation 36, we have with probability at least $1 - 3e^{-\tau}$,

$$\|\boldsymbol{\Psi}\| \leq \frac{c_0 \sigma_z \sigma_2 \sqrt{npq} (\tau + \log(2pq))}{n (1 - \delta(\tau))^2} = \frac{c_0 \sigma_z \sigma_2 \sqrt{pq} (\tau + \log(2pq))}{\sqrt{n} (1 - \delta(\tau))^2}.$$

$\qquad\square$

### E.6 PROOF OF LEMMA D.6

*Proof.* We decompose $\hat{\boldsymbol{\Theta}} := \boldsymbol{\Theta} + \boldsymbol{\Psi}$, where $\boldsymbol{\Psi} := (\mathbf{Z}^\top \mathbf{Z})^{-1} \mathbf{Z}^\top \mathbf{E_2}$. We have

$$\lambda_{\min}(\hat{\mathbf{H}}) = \lambda_{\min} \left( \frac{1}{n} (\boldsymbol{\Theta} + \boldsymbol{\Psi})^\top \mathbf{Z}^\top \mathbf{Z} (\boldsymbol{\Theta} + \boldsymbol{\Psi}) \right)$$

$$\geq \lambda_{\min}(\frac{\mathbf{Z}^\top \mathbf{Z}}{n}) \lambda_{\min} \left( (\boldsymbol{\Theta} + \boldsymbol{\Psi})^\top (\boldsymbol{\Theta} + \boldsymbol{\Psi}) \right) \tag{39}$$

$$= \lambda_{\min}(\frac{\mathbf{Z}^\top \mathbf{Z}}{n}) \sigma_{\min}^2(\boldsymbol{\Theta} + \boldsymbol{\Psi})$$

It remains to give a high probability bound for $\sigma^2_{\min}(\mathbf{Z})$ and $\sigma^2_{\min}(\mathbf{\Theta} + \mathbf{\Psi})$. For the first term, from Lemma D.2, we have when $n \geq C_0^2 \sigma_z^4 (\sqrt{q} + \sqrt{\tau})^2$, with probability at least $1 - 2e^{-\tau}$, we have

$$\sigma^2_{\min}(\mathbf{Z}) = \lambda_{\min}\left(\frac{\mathbf{Z}^\top \mathbf{Z}}{n}\right) \geq (1 - \delta(\tau))^2, \tag{40}$$

where $\delta(\tau) := \frac{C_0 \sigma_z^2 (\sqrt{q} + \sqrt{\tau})}{\sqrt{n}}$. For the second term, we apply Werl's inquality:

$$\sigma_{\min}(\mathbf{\Theta} + \mathbf{\Psi}) \geq \sigma_{\min}(\mathbf{\Theta}) - \|\mathbf{\Psi}\|. \tag{41}$$

From Lemma D.5, we have with probability at least $1 - 3e^{-\tau}$,

$$\|\mathbf{\Psi}\| \leq \frac{c_0 \sigma_z \sigma_2 \sqrt{pq}\,(\tau + \log(2pq))}{\sqrt{n}\,(1 - \delta(\tau))^2}. \tag{42}$$

Note that the RHS of equation 41 should be greater than 0, which requires $n = \Omega\left(pq(\tau + \log(pq))^2\right)$. Plug in equation 40 equation 41 equation 42 into equation 39, we have:

$$\begin{aligned}
\lambda_{\min}(\hat{\mathbf{H}}) &\geq (1 - \delta(\tau))^2 \left(\sigma_{\min}(\mathbf{\Theta}) - \|\mathbf{\Psi}\|\right)^2 \\
&\geq (1 - \delta(\tau))^2 \left(\sigma_{\min}(\mathbf{\Theta}) - \frac{c_0 \sigma_z \sigma_2 \sqrt{pq}\,(\tau + \log(2pq))}{\sqrt{n}\,(1 - \delta(\tau))^2}\right)^2.
\end{aligned} \tag{43}$$

Similarly, we have

$$\begin{aligned}
\lambda_{\max}(\hat{\mathbf{H}}) &= \lambda_{\max}\left(\frac{1}{n}(\mathbf{\Theta} + \mathbf{\Psi})^\top \mathbf{Z}^\top \mathbf{Z}(\mathbf{\Theta} + \mathbf{\Psi})\right) \\
&\leq \lambda_{\max}\left(\frac{\mathbf{Z}^\top \mathbf{Z}}{n}\right) \lambda_{\max}\left((\mathbf{\Theta} + \mathbf{\Psi})^\top (\mathbf{\Theta} + \mathbf{\Psi})\right) \\
&\leq (1 + \delta(\tau))^2 \left(\|\mathbf{\Theta}\| + \|\mathbf{\Psi}\|\right)^2 \\
&\leq (1 + \delta(\tau))^2 \left(\|\mathbf{\Theta}\| + \frac{c_0 \sigma_z \sigma_2 \sqrt{pq}\,(\tau + \log(2pq))}{\sqrt{n}\,(1 - \delta(\tau))^2}\right)^2,
\end{aligned} \tag{44}$$

which completes the proof. $\qquad\square$

### E.7 PROOF OF LEMMA D.7

*Proof.* We have

$$\begin{aligned}
\hat{\boldsymbol{\beta}} - \boldsymbol{\beta} &= \left(\hat{\mathbf{\Theta}}^\top \mathbf{Z}^\top \mathbf{Z} \hat{\mathbf{\Theta}}\right)^{-1} \hat{\mathbf{\Theta}}^\top \mathbf{Z}^\top \mathbf{Y} - \boldsymbol{\beta} \\
&= \left(\mathbf{X}^\top \mathbf{Z}(\mathbf{Z}^\top \mathbf{Z})^{-1}\mathbf{Z}^\top \mathbf{X}\right)^{-1} \mathbf{X}^\top \mathbf{Z}(\mathbf{Z}^\top \mathbf{Z})^{-1}\mathbf{Z}^\top \mathbf{Y} - \boldsymbol{\beta} \\
&= \left(\mathbf{X}^\top \mathbf{Z}(\mathbf{Z}^\top \mathbf{Z})^{-1}\mathbf{Z}^\top \mathbf{X}\right)^{-1} \mathbf{X}^\top \mathbf{Z}(\mathbf{Z}^\top \mathbf{Z})^{-1}\mathbf{Z}^\top \boldsymbol{\mathcal{E}}_1 \\
&= \frac{1}{n}(\hat{\mathbf{H}})^{-1} \mathbf{X}^\top \mathbf{Z}(\mathbf{Z}^\top \mathbf{Z})^{-1}\mathbf{Z}^\top \boldsymbol{\mathcal{E}}_1.
\end{aligned}$$

So that

$$\|\hat{\boldsymbol{\beta}} - \boldsymbol{\beta}\| \leq \frac{1}{n}\|(\hat{\mathbf{H}})^{-1}\|\|\mathbf{X}^\top \mathbf{Z}\|\|(\mathbf{Z}^\top \mathbf{Z})^{-1}\|\|\mathbf{Z}^\top \boldsymbol{\mathcal{E}}_1\| \tag{45}$$

From Lemma D.2 and Lemma D.6, when $n \geq C_1 pq(\tau + \log(pq))^2$, with probability at least $1 - 3e^{-\tau}$, we have the following bounds:

$$\|(\mathbf{Z}^\top \mathbf{Z})^{-1}\| \lesssim \frac{1}{n}, \quad \|(\hat{\mathbf{H}})^{-1}\| \lesssim 1. \tag{46}$$

Similar to equation 38, we have with probability at least $1 - e^{-\tau}$,

$$\|\mathbf{Z}^\top \boldsymbol{\mathcal{E}}_1\| \leq c_0 \sigma_z \sigma_1 \sqrt{nq}\,(\tau + \log(2q)) = \mathcal{O}\left(\sqrt{nq}\,(\tau + \log(q))\right). \tag{47}$$

It remains to derive a bound for $\|\mathbf{X}^\top \mathbf{Z}\|$. We have

$$\mathbf{X}^\top \mathbf{Z} = (\mathbf{Z}\mathbf{\Theta})^\top \mathbf{Z} + \mathcal{E}_2^\top \mathbf{Z}$$
$$= \mathbf{\Theta}^\top \mathbf{Z}^\top \mathbf{Z} + \mathcal{E}_2^\top \mathbf{Z},$$

where from Lemma D.2, we have with probability at least $1 - 2e^{-\tau}$,

$$\|\mathbf{Z}^\top \mathbf{Z}\| \le n(1 + \delta(\tau))^2,$$

and from equation 38, with probability at least $1 - e^{-\tau}$,

$$\|\mathbf{Z}^\top \mathcal{E}_2\| \le c_0 \sigma_z \sigma_2 \sqrt{npq} \left(\tau + \log(2pq)\right) = \mathcal{O}\left(\sqrt{npq}\left(\tau + \log(pq)\right)\right),$$

so we have with probability at least $1 - 3e^{-\tau}$,

$$\|\mathbf{X}^\top \mathbf{Z}\| \le n(1 + \delta(\tau))^2 \|\mathbf{\Theta}\| + c_0 \sigma_z \sigma_2 \sqrt{npq}\left(\tau + \log(2pq)\right) = \mathcal{O}\left(n + \sqrt{npq}\left(\tau + \log(pq)\right)\right). \tag{48}$$

From equation 45, with equation 46equation 47equation 48, when $n \ge C_0^2 \sigma_z^4 (\sqrt{q} + \sqrt{\tau})^2$, with probability at least $1 - 4e^{-\tau}$,

$$\|\hat{\boldsymbol{\beta}} - \boldsymbol{\beta}\| \lesssim \frac{\sqrt{nq}\left(\tau + \log(q)\right)\left(n + \sqrt{npq}\left(\tau + \log(pq)\right)\right)}{n^2}$$
$$= \frac{\sqrt{q}\left(\tau + \log(q)\right)}{\sqrt{n}} + \frac{q\sqrt{p}(\tau + \log(q))(\tau + \log(pq))}{n}.$$

When $n = \Omega\left(pq(\tau + \log(pq))^2\right)$, the above expression can be further simplified to

$$\|\hat{\boldsymbol{\beta}} - \boldsymbol{\beta}\| \lesssim \frac{\sqrt{q}(\tau + \log(q))}{\sqrt{n}}, \tag{49}$$

which concludes the proof. $\qquad\square$

### E.8 PROOF OF LEMMA D.8

*Proof.* We can decompose $\mathbf{r}$ as:

$$\mathbf{r} = \mathbf{Z}\hat{\mathbf{\Theta}}\hat{\boldsymbol{\beta}} - \mathbf{Y} = \mathbf{Z}\hat{\mathbf{\Theta}}\hat{\boldsymbol{\beta}} - (\mathbf{Z}\mathbf{\Theta} + \mathcal{E}_2)\boldsymbol{\beta} - \mathcal{E}_1$$
$$= \mathbf{Z}\hat{\mathbf{\Theta}}\hat{\boldsymbol{\beta}} - \mathbf{Z}\mathbf{\Theta}\boldsymbol{\beta} - \mathcal{E}_2\boldsymbol{\beta} - \mathcal{E}_1$$
$$= \mathbf{Z}\hat{\mathbf{\Theta}}\hat{\boldsymbol{\beta}} - \mathbf{Z}\mathbf{\Theta}\hat{\boldsymbol{\beta}} + \mathbf{Z}\mathbf{\Theta}\hat{\boldsymbol{\beta}} - \mathbf{Z}\mathbf{\Theta}\boldsymbol{\beta} - \mathcal{E}_2\boldsymbol{\beta} - \mathcal{E}_1$$
$$= \mathbf{Z}\left(\hat{\mathbf{\Theta}} - \mathbf{\Theta}\right)\hat{\boldsymbol{\beta}} + \mathbf{Z}\mathbf{\Theta}\left(\hat{\boldsymbol{\beta}} - \boldsymbol{\beta}\right) - \mathcal{E}_2\boldsymbol{\beta} - \mathcal{E}_1$$

and

$$\mathbf{Z}^\top \mathbf{r} = \mathbf{Z}^\top \mathbf{Z}\left(\hat{\mathbf{\Theta}} - \mathbf{\Theta}\right)\hat{\boldsymbol{\beta}} + \mathbf{Z}^\top \mathbf{Z}\mathbf{\Theta}\left(\hat{\boldsymbol{\beta}} - \boldsymbol{\beta}\right) - \mathbf{Z}^\top\left(\mathcal{E}_2\boldsymbol{\beta} + \mathcal{E}_1\right) \tag{50}$$

It suffices to bound $\|\mathbf{Z}^\top \mathbf{Z}\|$, $\|\hat{\mathbf{\Theta}} - \mathbf{\Theta}\|$, $\|\hat{\boldsymbol{\beta}} - \boldsymbol{\beta}\|$, and $\|\mathbf{Z}^\top\left(\mathcal{E}_2\boldsymbol{\beta} + \mathcal{E}_1\right)\|$. From Lemma D.2, we have with probability at least $1 - 2e^{-\tau}$,

$$\|\mathbf{Z}^\top \mathbf{Z}\| \le n\left(1 + \delta(\tau)\right)^2 \lesssim n.$$

From Lemma D.5, we can take

$$\left\|\hat{\mathbf{\Theta}} - \mathbf{\Theta}\right\| \le \frac{c_0 \sigma_z \sigma_2 \sqrt{pq}\left(\tau + \log(2pq)\right)}{\sqrt{n}\left(1 - \delta(\tau)\right)^2} = \mathcal{O}\left(\frac{\sqrt{pq}\left(\tau + \log(pq)\right)}{\sqrt{n}}\right).$$

From Lemma D.7,

$$\left\|\hat{\boldsymbol{\beta}} - \boldsymbol{\beta}\right\| \lesssim \frac{\sqrt{q}\left(\tau + \log(q)\right)}{\sqrt{n}}.$$

For the error $\mathbf{E}_{total} := \boldsymbol{\mathcal{E}}_2 \boldsymbol{\beta} + \boldsymbol{\mathcal{E}}_1$, note that $\mathbf{E}_{total,i} = \sum_{j=1}^p \boldsymbol{\mathcal{E}}_{2,ij} \boldsymbol{\beta}_j + \boldsymbol{\mathcal{E}}_{1,i}$ is zero-mean sub-Gaussian with parameter $\tilde{\sigma} := \sqrt{\sigma_2^2 \|\boldsymbol{\beta}\|^2 + \sigma_1^2}$, and hence the sub-exponential norm of $\mathbf{Z}^\top \mathbf{E}_{total}$ can be bounded by

$$\|(\mathbf{Z}^\top \mathbf{E}_{total})_k\|_{\psi_1} = \|\sum_{i=1}^n \mathbf{Z}_{i,k} \mathbf{E}_{total,i}\|_{\psi_1} \leq \sigma_z \tilde{\sigma} \sqrt{n}$$

Thus we have the tail bound:

$$\mathbb{P}\left(|(\mathbf{Z}^\top \mathbf{E}_{total})_k| \geq \tau\right) \leq 2e^{-\frac{\tau}{c_0 \sigma_z \tilde{\sigma} \sqrt{n}}}.$$

Taking the union bound over $k = 1, \ldots, q$, we have

$$\mathbb{P}\left(\|\mathbf{Z}^\top \mathbf{E}_{total}\| \geq \tau\right) \leq \mathbf{P}\left(\|\mathbf{Z}^\top \mathbf{E}_{total}\|_\infty \geq \frac{\tau}{\sqrt{q}}\right) \leq 2qe^{-\frac{\tau}{c_0 \sigma_z \tilde{\sigma} \sqrt{nq}}}.$$

Equivalently, with probability at least $1 - e^{-\tau}$,

$$\|\mathbf{Z}^\top \mathbf{E}_{total}\| \leq c_0 \sigma_z \tilde{\sigma} \sqrt{nq}\left(\tau + \log(2q)\right) = \mathcal{O}\left(\sqrt{nq}\left(\tau + \log(q)\right)\right).$$

Plugging these bounds into equation 50, we have with probability at least $1 - 3e^{-\tau}$,

$$\begin{aligned}
\|\mathbf{Z}^\top \mathbf{r}\| &\leq \|\mathbf{Z}^\top \mathbf{Z}\| \delta_{\hat{\Theta}}(\|\boldsymbol{\beta}\| + \delta_{\hat{\beta}}) + \|\mathbf{Z}^\top \mathbf{Z}\| \|\boldsymbol{\Theta}\| \delta_{\hat{\beta}} + \|\mathbf{Z}^\top \mathbf{E}_{total}\| \\
&= \|\mathbf{Z}^\top \mathbf{Z}\|(\|\boldsymbol{\Theta}\| + \delta_{\hat{\Theta}})\delta_{\hat{\beta}} + \|\mathbf{Z}^\top \mathbf{Z}\| \|\boldsymbol{\beta}\| \delta_{\hat{\Theta}} + \|\mathbf{Z}^\top \mathbf{E}_{total}\| \\
&\lesssim n\left(1 + \frac{\sqrt{pq}\left(\tau + \log(pq)\right)}{\sqrt{n}}\right)\frac{\sqrt{q}\left(\tau + \log(q)\right)}{\sqrt{n}} + n\frac{\sqrt{pq}\left(\tau + \log(pq)\right)}{\sqrt{n}} + \sqrt{nq}\left(\tau + \log(q)\right) \\
&\lesssim \sqrt{npq}\left(\tau + \log(pq)\right).
\end{aligned}$$

$\square$

### E.9 Proof of Lemma D.9

*Proof.* We have

$$\begin{aligned}
\|\mathbf{L}^{(t)}\| &= \|\mathbf{I} - \frac{\alpha}{n} \boldsymbol{\Theta}^{(t)\top} \mathbf{Z}^\top \mathbf{Z} \boldsymbol{\Theta}^{(t)}\| \\
&= \|\mathbf{I} - \frac{\alpha}{n}\left(\boldsymbol{\Theta}^{(t)} - \hat{\boldsymbol{\Theta}} + \hat{\boldsymbol{\Theta}}\right) \mathbf{Z}^\top \mathbf{Z} \left(\boldsymbol{\Theta}^{(t)} - \hat{\boldsymbol{\Theta}} + \hat{\boldsymbol{\Theta}}\right)^\top\| \\
&= \|\mathbf{I} - \frac{\alpha}{n}\left(\mathbf{e}_{\boldsymbol{\Theta}}^{(t)} + \hat{\boldsymbol{\Theta}}\right) \mathbf{Z}^\top \mathbf{Z} \left(\mathbf{e}_{\boldsymbol{\Theta}}^{(t)} + \hat{\boldsymbol{\Theta}}\right)^\top\| \\
&\leq 1 + \frac{\alpha}{n}\|\mathbf{Z}^\top \mathbf{Z}\|\left(\|\mathbf{e}_{\boldsymbol{\Theta}}^{(t)}\| + \|\hat{\boldsymbol{\Theta}}\|\right)^2 \\
&\lesssim 1 + \left(\|\mathbf{e}_{\boldsymbol{\Theta}}^{(t)}\| + 1\right)^2,
\end{aligned}$$

where $\mathbf{e}_{\boldsymbol{\Theta}}^{(t)} := \boldsymbol{\Theta}^{(t)} - \hat{\boldsymbol{\Theta}}$. Note that from Lemma D.4, with parameters choice equation 4 and sample size condition equation 6, we have $\|\mathbf{e}_{\boldsymbol{\Theta}}^{(t)}\| \lesssim 1, \forall t = 0, 1, \ldots, \lceil C_2 \rceil - 1$. So that there exists a constant $c_L$, such that $\|\mathbf{L}^{(t)}\| \leq c_L, \forall t = 0, 1, \ldots, \lceil C_2 \rceil - 1$, where $C_2$ is the upper bound of $t_0(n)$ in equation 22. Besides, when $0 < \alpha < \frac{4}{2\bar{\gamma}(\tau) + \underline{\gamma}(\tau)}$, from equation 14, we have $\|\mathbf{L}^{(t)}\| < 1, \forall t = \lceil C_2 \rceil, \ldots, T_0 - 1$. Therefore, we have

$$\prod_{t=0}^{T_0-1} \|\mathbf{L}^{(t)}\| \leq c_L^{\lceil C_2 \rceil} \lesssim 1,$$

which concludes the proof. $\square$

# F  ADDITIONAL DISCUSSIONS

## F.1  PRIVACY FOR $\beta$ ONLY

In Algorithm 1, the privacy parameter $\rho$ is with respect to $\mathbf{\Theta}^{(1)}, \ldots, \mathbf{\Theta}^{(T)}, \boldsymbol{\beta}^{(1)}, \ldots, \boldsymbol{\beta}^{(T)}$. However, in some applications, we may only care about the privacy of the major estimator $\boldsymbol{\beta}^{(1)}, \ldots, \boldsymbol{\beta}^{(T)}$. We note that in Algorithm 1, one can modify the output to only include $\boldsymbol{\beta}^{(1)}, \ldots, \boldsymbol{\beta}^{(T)}$ while still maintaining the privacy guarantees. We have the following lemma:

**Lemma F.1.** For $\rho_1 \in (0, \infty]$ and $\lambda_1 \in [0, \infty)$ Algorithm 1 is $\rho$-zCDP for output $\boldsymbol{\beta}^{(1)}, \ldots, \boldsymbol{\beta}^{(T)}$, where $\rho := \rho_2 = \frac{2T\gamma_2^2}{n^2\lambda_2^2}$.

Suppose that $\rho_1 = \infty$, i.e. we remove $\mathbf{\Xi}$, the additive noise of the first stage. One can show that we can get a slightly tighter bound for equation 7. However, for any fixed $\rho_2$, we observe that there is no improvement on the rate of convergence than Theorem 3.1.

Consider the following algorithm:

---
**Algorithm 3** DP-2S-GD-$\beta$
---
1: **Input:** Data $\mathbf{Z} \in \mathbb{R}^{n \times q}$, $\mathbf{X} \in \mathbb{R}^{n \times p}$, $\mathbf{Y} \in \mathbb{R}^n$
2: **Parameters:** Clipping threshold $\gamma_2 > 0$, noise scale $\lambda_2 > 0$, step sizes $\alpha, \eta > 0$, number of iterations $T$, initial estimates $\boldsymbol{\beta}^{(0)} = \mathbf{0}_p$, $\mathbf{\Theta}^{(0)} = \mathbf{0}_{q \times p}$
3: **for** $t = 0, 1, \ldots, T-1$ **do**
4:     Draw $\boldsymbol{\nu}^{(t)} \sim \mathcal{N}(0, \lambda_2^2 \mathbf{I}_p)$.
5:     $\mathbf{\Theta}^{(t+1)} = \mathbf{\Theta}^{(t)} - \frac{\eta}{n} \sum_{i=1}^n \mathbf{z}_i (\mathbf{z}_i^\top \mathbf{\Theta}^{(t)} - \mathbf{x}_i^\top)$
6:     $\boldsymbol{\beta}^{(t+1)} = \boldsymbol{\beta}^{(t)} - \frac{\alpha}{n} \sum_{i=1}^n \mathrm{CLIP}_{\gamma_2} \left\{ \mathbf{\Theta}^{(t)\top} \mathbf{z}_i \left( \mathbf{z}_i^\top \mathbf{\Theta}^{(t)} \boldsymbol{\beta}^{(t)} - y_i \right) \right\} + \alpha \boldsymbol{\nu}^{(t)}$
7: **end for**
8: **return** $\boldsymbol{\beta}^{(1)}, \ldots, \boldsymbol{\beta}^{(T)}$

---

We have the following theorem:

**Theorem F.1.** *For any fixed $\mathbf{\Theta} \in \mathbb{R}^{q \times p}$ and $\boldsymbol{\beta} \in \mathbb{R}^p$, consider the Algorithm 3 with step sizes satisfying*

$$0 < \eta < \frac{2}{(1 + \delta(\tau))^2}, \quad 0 < \alpha < \frac{4}{2\bar{\gamma}(\tau) + \underline{\gamma}(\tau)},$$

*under Assumption 2, with parameters*

$$\lambda_1 = \frac{2\gamma_1}{n} \sqrt{\frac{T}{\rho_1}}, \quad \lambda_2 = \frac{2\gamma_2}{n} \sqrt{\frac{T}{\rho_2}}, \quad \gamma_1 = \gamma_2 = c_0 \left( \sqrt{q} + \sqrt{\tau + \log(nT)} \right)^2,$$

*if*

$$n \geq c_1 \max \left\{ pq(\tau + \log(pq))^2, \frac{\left( \sqrt{q} + \sqrt{\tau} \right)^3}{\sqrt{\rho_2}} \right\},$$

*for any fixed $\tau$, with probability $1 - c_2 e^{-\tau}$, we have*

$$\|\boldsymbol{\beta}^{(T)} - \hat{\boldsymbol{\beta}}\| \lesssim \kappa(\tau)^{\frac{T}{2}} + \frac{\sqrt{p}(\sqrt{q} + \sqrt{\tau})^3}{n\sqrt{\rho_2}} \sqrt{T} + \frac{\sqrt{pq}(\tau + \log(pq))}{\sqrt{n}},$$

*where the definitions of $\kappa(\tau)$, $\bar{\gamma}(\tau)$, $\underline{\gamma}(\tau)$ and $\delta(\tau)$ are the same as in Theorem 3.1.*

*Proof.* The proof follows from similar approach as in the proof of Theorem 3.1. However, in equation 11, we can simplify as follows:

$$\mathbf{e}_{\mathbf{\Theta}}^{(t+1)} = \left( \mathbf{I} - \frac{\eta}{n} \mathbf{Z}^\top \mathbf{Z} \right)^{t+1} \mathbf{e}_{\mathbf{\Theta}}^{(0)}.$$

So in equation 18, we take

$$\varepsilon = \kappa_{\boldsymbol{\Theta}}(\tau)^{T_0}\|\hat{\boldsymbol{\Theta}}\| \lesssim \kappa_{\boldsymbol{\Theta}}(\tau)^{T_0},$$

and in equation 19,

$$\max_{0 \le k \le T_0 - 1}\|\mathbf{e}_{\boldsymbol{\Theta}}^{(k)}\| \le \|\hat{\boldsymbol{\Theta}}\| \lesssim 1.$$

Thus, to satisfy condition equation 20, we only need

$$\kappa_{\boldsymbol{\Theta}}(\tau)^{T_0}\|\hat{\boldsymbol{\Theta}}\| \le \bar{\varepsilon},$$

where $\bar{\varepsilon} := \sqrt{\|\hat{\boldsymbol{\Theta}}\|^2 + \frac{\gamma(\tau)}{2(1+\delta(\tau))^2}} - \|\hat{\boldsymbol{\Theta}}\|$. Comparing this with equation 21, we can see that there is no constraint on $T$. We only need to take

$$T_0 \ge t_0(n),$$

where $t_0(n)$ is defined in equation 22. We still take partition point $\tilde{T}_0 := \max\{\frac{T}{2}, C_2\}$, similar to equation 27, we have

$$\|\mathbf{e}_{\boldsymbol{\beta}}^{(\tilde{T}_0)}\| \lesssim 1 + \lambda_2\left(\sqrt{p} + \sqrt{\tau}\right).$$

Further, from equation 16, we have

$$
\begin{aligned}
\|\mathbf{e}_{\boldsymbol{\beta}}^{(T)}\| &\lesssim \kappa_{\boldsymbol{\beta}}(\tau)^{T - \tilde{T}_0}\|\mathbf{e}_{\boldsymbol{\beta}}^{(\tilde{T}_0)}\| + \varepsilon(1 + \varepsilon) + \frac{\sqrt{pq}(\tau + \log(pq))}{\sqrt{n}}(1 + \varepsilon) + \lambda_2\left(\sqrt{p} + \sqrt{\tau}\right) \\
&\lesssim \kappa_{\boldsymbol{\beta}}(\tau)^{\frac{T}{2}}\left(1 + \lambda_2\left(\sqrt{p} + \sqrt{\tau}\right)\right) + \kappa_{\boldsymbol{\Theta}}(\tau)^{\frac{T}{2}} + \frac{\sqrt{pq}(\tau + \log(pq))}{\sqrt{n}} + \lambda_2\left(\sqrt{p} + \sqrt{\tau}\right) \\
&\lesssim \kappa_{\boldsymbol{\beta}}(\tau)^{\frac{T}{2}} + \kappa_{\boldsymbol{\Theta}}(\tau)^{\frac{T}{2}} + \frac{\sqrt{pq}(\tau + \log(pq))}{\sqrt{n}} + \lambda_2\left(\sqrt{p} + \sqrt{\tau}\right),
\end{aligned}
\tag{51}
$$

where $\lambda_2 = \frac{2\gamma_2}{n}\sqrt{\frac{T}{\rho_2}}$, and $\gamma_2 = c_2\left(\sqrt{q} + \sqrt{\tau + \log(nT)}\right)^2$. Plug in $\lambda_2$ into equation 51, we have

$$\|\mathbf{e}_{\boldsymbol{\beta}}^{(T)}\| \lesssim \underbrace{\kappa_{\boldsymbol{\beta}}(\tau)^{\frac{T}{2}}}_{(i)} + \underbrace{\kappa_{\boldsymbol{\Theta}}(\tau)^{\frac{T}{2}}}_{(ii)} + \underbrace{\frac{R(\tau)}{\sqrt{\rho_2}}\frac{\sqrt{T}}{n}}_{(iii)} + \underbrace{\frac{\sqrt{pq}(\tau + \log(pq))}{\sqrt{n}}}_{(iv)} \tag{52}$$

Comparing equation 52 with equation 29, we observe that the error term in (ii) is reduced due to the absence of noise in $\boldsymbol{\Theta}^{(t)}$ update. When $T = \mathcal{O}(n)$, this improvement is insignificant as the order of the bound equation 52 is dominated by (iv). However, in Theorem F.1, since there is no restriction on $T$, equation 52 holds for all $T$. □

We conduct experiments to compare the performance of Algorithm 1 and Algorithm 3 under the same setup as in Section 4. We fix $n = 1000$ and $p = q = r = 5$. For Algorithm 1, we set $\rho_1 = 1$ and vary $\rho_2 \in \{0.1, 1, 10\}$, while running both algorithms for a range of iterations. The results are shown in Figure 6. We observe that when $T = \mathcal{O}(n)$, the two algorithms exhibit comparable performance. However, when $T$ grows larger, Algorithm 1 diverges, whereas Algorithm 3 continues to maintain a stable error trajectory.

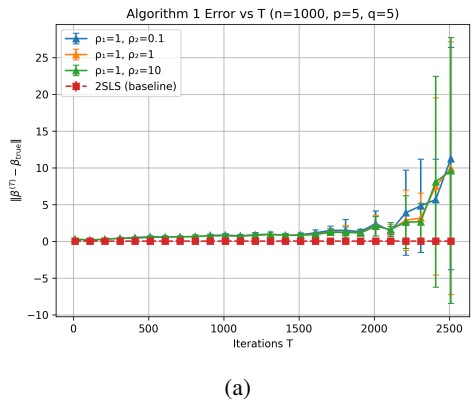 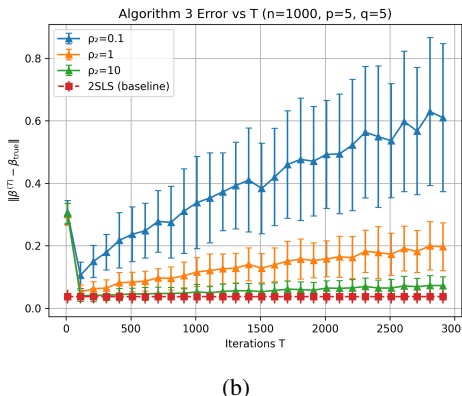

$$(a) \qquad\qquad\qquad\qquad (b)$$

Figure 6: Comparison of Algorithm 1 and Algorithm 3. We fix $n = 1000, p = q = r = 5$, and vary $\rho_2 \in \{0.1, 1, 10\}$. (a) Error curve for Algorithm 1, where we set $\rho_1 = 1$. (b) Error curve for Algorithm 3. All the curves are averaged over 100 runs, with vertical bars representing the standard errors.

### F.2 COMPARISON TO FRIENDLYCORE APPROACH

In this section, we provide a brief comparison between our two-stage gradient perturbation approach and a computationally friendly approach. Tsfadia et al. (2022) proposes a general DP meta-framework for aggregation tasks (e.g., averaging, clustering, covariance estimation) on data in a metric space. The key idea is: Given a dataset $D = (\mathbf{x}_1, \ldots, \mathbf{x}_n)$ in a metric space and a "friend-ship" predicate $f(\mathbf{x}, \mathbf{y})$ that encodes when two points are close / well-behaved, FriendlyCore is a DP procedure that extracts a subset $C \subseteq D$ (the "core") with two properties:

- Structural niceness: $C$ is $f$-friendly (e.g., all points in $C$ lie in a ball of radius $r$, or satisfy a separation condition useful for clustering). Outliers that violate the predicate are removed.
- Stability and DP: For neighboring datasets $D, D'$, the cores $C, C'$ differ only on a small number of points, and FriendlyCore itself is $(\varepsilon, \delta)$-DP or $\rho$-zCDP. This lets you plug $C$ into any friendly-instance DP algorithm without re-doing a worst-case sensitivity analysis.

Section 5 of Tsfadia et al. (2022) shows how to use this framework for private averaging, clustering, and covariance estimation. For averaging, Algorithm 5.1 FC_Avg does:

- Split the privacy budget as $\rho_1 = 0.1\rho, \rho_2 = 0.9\rho$.
- Run FriendlyCore on $D$ with a predicate that enforces an effective diameter $r$ (all but a few outliers lie in a ball of radius $r$ ).
- On the core $C$, run FriendlyAvg, which is essentially a Gaussian-mechanism mean estimator tuned for zCDP .

Algorithm 5.1 from Tsfadia et al. (2022) can be used as a building block to make a DP version of this 2SLS analysis: The 2SLS estimator depends on sample means of sufficient statistics: $\frac{1}{n}\mathbf{Z}^\top\mathbf{Z}, \frac{1}{n}\mathbf{Z}^\top\mathbf{X}, \frac{1}{n}\mathbf{Z}^\top\mathbf{Y}$. Stack and vectorize these matrices into a vector in $\mathbb{R}^d$ with $d = q^2 + pq + q = \mathcal{O}(q^2)$. Each data point contributes a vector of this form; call these contributions $\tilde{\mathbf{x}}_i$. Under the sub-Gaussian design, each $\tilde{\mathbf{x}}_i$ has bounded effective diameter

$$r = \Theta(\sqrt{d} + \sqrt{\tau})$$

with high probability. Using that $r$, Algorithm 5.1 (FC Avg) gives a $\rho$-zCDP estimate of the mean of the sufficient-statistics vector with error

$$\mathcal{O}\left(\frac{r}{n}\sqrt{\frac{d}{\rho}}\right) = \mathcal{O}\left(\frac{\sqrt{d} + \sqrt{\tau}}{n}\sqrt{\frac{d}{\rho}}\right) = \mathcal{O}\left(\frac{d + \sqrt{d\tau}}{n\sqrt{\rho}}\right).$$

Lemma D.7's proof shows that $\hat{\beta}$ is a smooth function of those sample covariances. The final estimate $\hat{\beta}_{FC}$ is then a post-processing step that applies the usual 2SLS formula to these privatized moments. If we replace the non-private moment estimates in Lemma D.7 by the FC_Avg privatized moments, the Lipschitz dependence of $\hat{\beta}$ on the moments converts the FC_Avg error into an additional term in the error bound, scaling like at the order of $L\frac{q^2+q\sqrt{\tau}}{n\sqrt{\rho}}$, where $L$ is the Lipschitz constant (see Lemma F.2). Hence, the error bound for a FriendlyCore-based DP 2SLS estimator would be the combination of the non-private rate from Lemma D.7 plus this additional term. In particular,

$$\|\hat{\boldsymbol{\beta}}_{FC} - \boldsymbol{\beta}\| \leq \|\hat{\boldsymbol{\beta}}_{FC} - \hat{\boldsymbol{\beta}}\| + \|\hat{\boldsymbol{\beta}} - \boldsymbol{\beta}\| = \mathcal{O}\left(L\frac{q^2+q\sqrt{\tau}}{n\sqrt{\rho}} + \frac{\sqrt{pq}(\tau+\log(pq))}{\sqrt{n}}\right).$$

**Remark F.1.** Comparing this bound with our bound in Theorem 3.1, we see that the orders are similar. At the same time, the FriendlyCore-based estimator reflects a different set of algorithmic and statistical trade-offs from our DP-2S-GD method:

- **Black-box generality vs. algorithm-aware design.** FriendlyCore offers a highly general subsample–aggregate framework that applies to a broad class of moment-based estimators, including 2SLS, by treating them as black-box functions of empirical moments. In contrast, DP-2S-GD directly privatizes the two-stage gradient procedure itself. This algorithm-aware approach enables us to analyze how privacy noise propagates through the optimization dynamics and to obtain an explicit privacy–iteration trade-off (e.g., the "too many iterations hurt" behavior in Figure 2), which does not emerge in one-shot subsample–aggregate pipelines.

- **Subsample-and-aggregate vs. full-data iterative updates.** FriendlyCore relies on repeated subsampling to identify a stable "core" before applying the estimator on that core, which provides robustness and generality but increases computational cost and effectively reduces the sample size available to each subsample. DP-2S-GD uses all samples at every iteration with per-sample gradient clipping and a simple zCDP accountant, making it attractive in large-scale, iterative, or streaming settings where repeated subsampling may be impractical.

**Lemma F.2.** Let $\mathbf{s} := \text{vec}(\frac{\mathbf{Z}^\top\mathbf{Z}}{n}, \frac{\mathbf{Z}^\top\mathbf{X}}{n}, \frac{\mathbf{Z}^\top\mathbf{Y}}{n})$, and $\boldsymbol{\beta}(\mathbf{s})$ denotes the mapping from $\mathbf{s}$ to $\hat{\boldsymbol{\beta}}_{2SLS}$. There exists a constant $L > 0$ such that for any two sets of moments $\mathbf{s}_1, \mathbf{s}_2$, we have

$$\|\boldsymbol{\beta}(\mathbf{s}_1) - \boldsymbol{\beta}(\mathbf{s}_2)\| \leq L\|\mathbf{s}_1 - \mathbf{s}_2\|.$$

*Proof.* Let $\boldsymbol{\Sigma}_{zz} = \frac{1}{n}\mathbf{Z}^\top\mathbf{Z}, \qquad \boldsymbol{\Sigma}_{zx} = \frac{1}{n}\mathbf{Z}^\top\mathbf{X}, \qquad \boldsymbol{\Sigma}_{zy} = \frac{1}{n}\mathbf{Z}^\top\mathbf{Y}$ denote the empirical second moments and define the "moment vector"

$$\mathbf{s} := (\boldsymbol{\Sigma}_{zz}, \boldsymbol{\Sigma}_{zx}, \boldsymbol{\Sigma}_{zy}).$$

From these moments we form the usual 2SLS normal equations

$$\mathbf{G}(\mathbf{s})\boldsymbol{\beta}(\mathbf{s}) = \mathbf{h}(\mathbf{s}),$$

where $\mathbf{G}(\mathbf{s}) := \mathbf{X}^\top\mathbf{P}_z\mathbf{X}, \quad \mathbf{h}(\mathbf{s}) := \mathbf{X}^\top\mathbf{P}_z\mathbf{Y}, \quad \mathbf{P}_z := \mathbf{Z}(\mathbf{Z}^\top\mathbf{Z})^{-1}\mathbf{Z}^\top$. Hence,

$$\mathbf{G}(\mathbf{s}) = \boldsymbol{\Sigma}_{xz}\boldsymbol{\Sigma}_{zz}^{-1}\boldsymbol{\Sigma}_{zx}, \qquad \mathbf{h}(\mathbf{s}) = \boldsymbol{\Sigma}_{xz}\boldsymbol{\Sigma}_{zz}^{-1}\boldsymbol{\Sigma}_{zy},$$

with $\boldsymbol{\Sigma}_{xz} = \boldsymbol{\Sigma}_{zx}^\top$. Thus the 2SLS estimator can be written as

$$\boldsymbol{\beta}(\mathbf{s}) = \mathbf{G}(\mathbf{s})^{-1}\mathbf{h}(\mathbf{s}).$$

Consider two sets of moments $s_1, s_2$ with corresponding

$$\mathbf{G}_i := \mathbf{G}(\mathbf{s}_i), \quad \mathbf{h}_i := \mathbf{h}(\mathbf{s}_i), \quad \boldsymbol{\beta}_i := \boldsymbol{\beta}(\mathbf{s}_i) = \mathbf{G}_i^{-1}\mathbf{h}_i \quad (i = 1, 2).$$

We have

$$\begin{aligned}\boldsymbol{\beta}_1 - \boldsymbol{\beta}_2 &= \mathbf{G}_1^{-1}\mathbf{h}_1 - \mathbf{G}_2^{-1}\mathbf{h}_2 \\ &= \mathbf{G}_1^{-1}(\mathbf{h}_1 - \mathbf{h}_2) + (\mathbf{G}_1^{-1} - \mathbf{G}_2^{-1})\mathbf{h}_2\end{aligned}$$

$$= \mathbf{G}_1^{-1}(\mathbf{h}_1 - \mathbf{h}_2) + \mathbf{G}_1^{-1}(\mathbf{G}_2 - \mathbf{G}_1)\mathbf{G}_2^{-1}\mathbf{h}_2,$$

where in the last equality we used the identity

$$\mathbf{G}_1^{-1} - \mathbf{G}_2^{-1} = \mathbf{G}_1^{-1}(\mathbf{G}_2 - \mathbf{G}_1)\mathbf{G}_2^{-1}.$$

Taking norms and using submultiplicativity,

$$\|\boldsymbol{\beta}_1 - \boldsymbol{\beta}_2\| \le \|\mathbf{G}_1^{-1}\| \, \|\mathbf{h}_1 - \mathbf{h}_2\| + \|\mathbf{G}_1^{-1}\| \, \|\mathbf{G}_2 - \mathbf{G}_1\| \, \|\mathbf{G}_2^{-1}\| \, \|\mathbf{h}_2\|. \tag{53}$$

Assume (as in Assumption 2) that the population Gram matrix is well conditioned, so that on a high-probability event

$$\lambda_{\min}(\mathbf{G}_i) \ge \lambda_0 > 0 \quad \Rightarrow \quad \|\mathbf{G}_i^{-1}\| \le \lambda_0^{-1}, \quad i = 1, 2,$$

and that the moments are uniformly bounded so that $\|\mathbf{h}_2\| \le C_h$. Here the specific formula of $\lambda_0$ could be referred to Lemma D.6. Moreover, $\mathbf{G}(\mathbf{s})$ and $h(\mathbf{s})$ are smooth (in fact, rational) functions of the entries of $(\boldsymbol{\Sigma}_{zz}, \boldsymbol{\Sigma}_{zx}, \boldsymbol{\Sigma}_{zy})$, and in a neighbourhood of the true moments there exist constants $C_G, C_h > 0$ such that

$$\|\mathbf{G}_2 - \mathbf{G}_1\| \le C_G \|\mathbf{s}_2 - \mathbf{s}_1\|, \qquad \|\mathbf{h}_2 - \mathbf{h}_1\| \le C_h \|\mathbf{s}_2 - \mathbf{s}_1\|.$$

Plugging these bounds into equation 53 yields

$$\|\boldsymbol{\beta}_1 - \boldsymbol{\beta}_2\| \le \frac{1}{\lambda_0} C_h \|\mathbf{s}_1 - \mathbf{s}_2\| + \frac{1}{\lambda_0^2} C_G C_h \|\mathbf{s}_1 - \mathbf{s}_2\| = L \|\mathbf{s}_1 - \mathbf{s}_2\|,$$

where

$$L := \frac{C_h}{\lambda_0} + \frac{C_G C_h}{\lambda_0^2}.$$

Hence the 2SLS map $\boldsymbol{\beta} : \mathbf{s} \mapsto \hat{\boldsymbol{\beta}}_{\text{2SLS}}(\mathbf{s})$ is Lipschitz in the sample moments:

$$\|\boldsymbol{\beta}(\mathbf{s}_1) - \boldsymbol{\beta}(\mathbf{s}_2)\| \le L \|\mathbf{s}_1 - \mathbf{s}_2\|.$$

$\square$

# G    ADDITIONAL EXPERIMENTS

## G.1    TUNING STEP SIZE

In this section, we empirically examine how the step sizes $\alpha$ and $\eta$ affect the convergence of Algorithm 1. Using the same setup as in Section 4.1, we fix $n = 2000$ and $p = q = r = 5$, and run Algorithm 1 for $T = 20$ iterations, with $\rho_1 = \rho_2 = 5$. In each plot, we vary one of $\eta, \alpha$ over its admissible range given by equation 3, while fixing the other step size at a sub-optimal level (close to its upper bound). The results, shown in Figure 7, indicate that as long as $\eta$ and $\alpha$ lie within the theoretically justified region, the convergence behavior is fairly insensitive to the exact step-size choice. As noted in Remark 3.2, our theoretical upper bound for $\alpha$ is slightly conservative due to the need to control the randomness introduced by the first-stage estimates $\boldsymbol{\Theta}^{(t)}$.

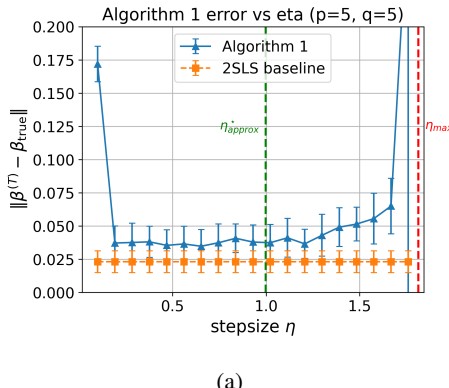 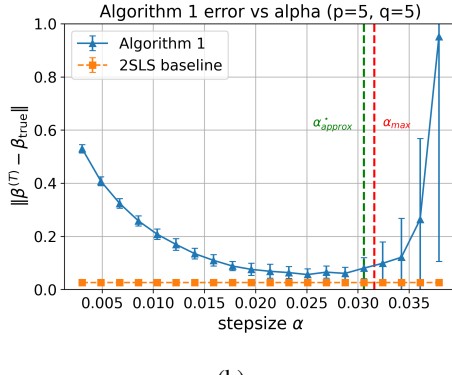

(a)                 (b)

Figure 7: Convergence behavior under different step sizes $\eta, \alpha$. The (theoretical) upper bounds $\eta_{\max}$ and $\alpha_{\max}$ are given by equation 3. The (approximate) optimal $\eta^\star_{\text{approx}}$ and $\alpha^\star_{\text{approx}}$ are calculated according to equation 8. (a) Varying $\eta$ while fixing $\alpha = \alpha_{\max}$. (b) Varying $\alpha$ while fixing $\eta = 0.9\eta_{\max}$. All the curves are averaged over 100 runs, with vertical bars representing the standard errors.

## G.2 Effect of clipping threshold

In this section, we empirically examine how the clipping thresholds $\gamma_1$ and $\gamma_2$ influence the utility of Algorithm 1. Using the same setup as in Section 4.1, we fix $n = 2000$ and $p = q = r = 5$, and run Algorithm 1 for $T = 20$ iterations under privacy budgets $\rho_1 = \rho_2 = 5$. For simplicity, we set $\gamma_1 = \gamma_2 = \gamma$ and vary $\gamma$ over the range $[1, 1000]$. The results are reported in Figure 8.

We observe that when $\gamma$ is set too small, the per-sample gradients are frequently clipped, causing the updates to be severely distorted and resulting in larger estimation error. As $\gamma$ increases, clipping becomes less frequent and the estimation error decreases. However, once $\gamma$ exceeds a certain level, the sensitivity of the gradients grows, which requires injecting larger noise to satisfy the target privacy budget. This increased noise leads to larger fluctuations in the final estimates. Consequently, the most effective choice of $\gamma$ is the smallest value that ensures gradient clipping does not occur with high probability.

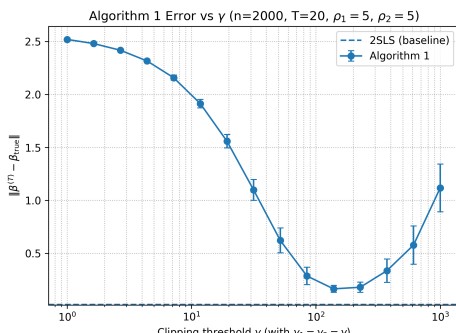

Figure 8: Effect of clipping threshold $\gamma$ on the utility of Algorithm 1. We fix $n = 2000, p = q = r = 5, T = 20, \rho_1 = \rho_2 = 5$, and set $\gamma_1 = \gamma_2 = \gamma$. The error curve is averaged over 100 runs, with vertical bars representing the standard errors.

## G.3 Convergence Rate Comparison

In this section, we empirically compare the convergence rate of 2S-GD (Algorithm 2) and the standard 2SLS estimator. The experiment setup is exactly the same as in Section 4.1. We set $p = q = r = 20$, and vary $n$ from 500 to 5000. For the 2S-GD estimator, we run $T = 100$

iterations so that it converges sufficiently. The results are shown in Figure 9. We observe that the convergence rate of 2S-GD is slower than that of 2SLS.

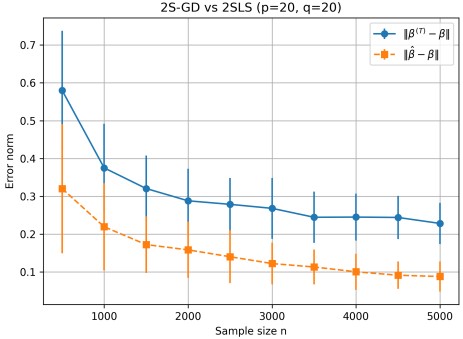

Figure 9: Comparison of the convergence rates of 2S-GD and 2SLS. The error curves $\|\boldsymbol{\beta}^{(T)} - \boldsymbol{\beta}\|$ (for 2S-GD) and $\|\hat{\boldsymbol{\beta}} - \boldsymbol{\beta}\|$ (for 2SLS) are averaged over 100 runs, with vertical bars representing the standard errors.

### G.4 ADDITIONAL EXPERIMENTS ON ANGRIST DATASET

We provide additional experimental results on the Angrist dataset with different privacy parameters $\rho_1, \rho_2$. We consider two settings of privacy parameters: (i) $\rho_1 = 0.1, \rho_2 = 0.1$; (ii) $\rho_1 = 10, \rho_2 = 10$. The results are shown in Figures 10 and 11. We observe that when $\rho_1, \rho_2$ are small, the estimates of $\boldsymbol{\beta}^{(T)}$ have larger variance. When $\rho_1, \rho_2$ are larger, the estimates of $\boldsymbol{\beta}^{(T)}$ are more concentrated around the expected value. In both settings, the estimates of $\boldsymbol{\beta}^{(t)}$ converge in expectation within $T = 20$ iterations.

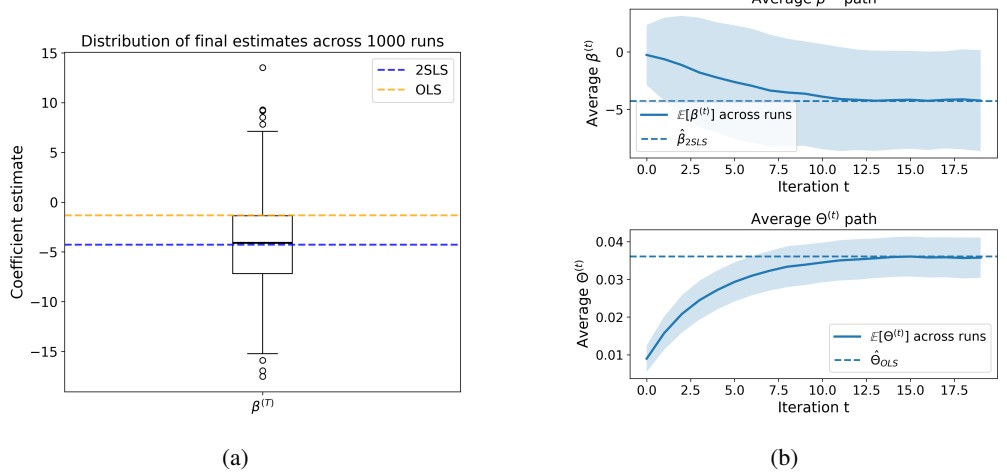

Figure 10: Results on the Angrist dataset with $T = 20, \rho_1 = 0.1, \rho_2 = 0.1$. (a) Boxplot of estimated $\boldsymbol{\beta}^{(T)}$, over 1000 runs. (b) Learning paths of parameters $\boldsymbol{\beta}^{(t)}, \boldsymbol{\Theta}^{(t)}$, over 1000 runs. The shaded area represents the standard error.

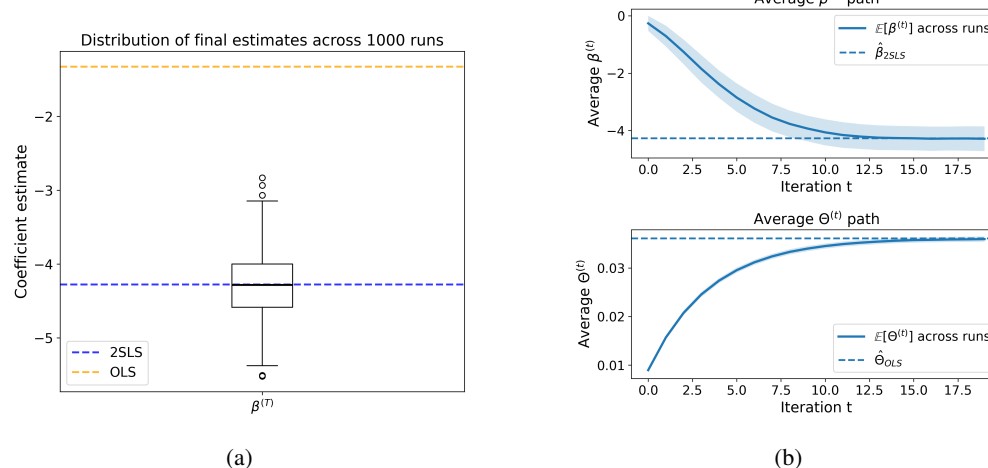

(a)                                                                                  (b)

Figure 11: Results on the Angrist dataset with $T = 20, \rho_1 = 10, \rho_2 = 10$. (a) Boxplot of estimated $\boldsymbol{\beta}^{(T)}$, over 1000 runs. (b) Learning paths of parameters $\boldsymbol{\beta}^{(t)}, \boldsymbol{\Theta}^{(t)}$, over 1000 runs. The shaded area represents the standard error.

### G.5   EXPERIMENTS ON CARD DATASET

The Card dataset (Card, 1993) is a widely used empirical dataset in labor economics for studying the causal effect of education on earnings. In this study, the endogenous regressor is individuals' years of schooling (**educ**), and the outcome variable is log earnings (**lwage**). There are several instruments available, most notably the college-proximity indicators (**nearc2** and **nearc4**), which capture whether an individual grew up near a two-year or four-year college. Additional instruments include parental education—father's and mother's years of schooling (**fatheduc** and **motheduc**)—which provide further exogenous variation in educational attainment.

There are 2191 samples in total. We consider the following covariates: $\mathbf{Z} =$[**nearc2**, **nearc4**, **fatheduc**, **motheduc**], $\mathbf{X} =$[**educ**], $\mathbf{Y} =$[**lwage**]. We standardize each column of $\mathbf{Z}$ to have zero mean and unit variance. We run Algorithm 1 with privacy parameters $\rho_1, \rho_2 \in \{0.1, 1, 10\}$, and number of iterations $T = 15$. We report the boxplot of final estimates and the learning path for $\boldsymbol{\beta}^{(t)}$. The results are shown in Figure 12. We observe that as $\rho_1, \rho_2$ increase, the estimates of $\boldsymbol{\beta}^{(T)}$ become more concentrated. In all settings, the estimates of $\boldsymbol{\beta}^{(t)}$ converge in expectation within $T = 10$ iterations.

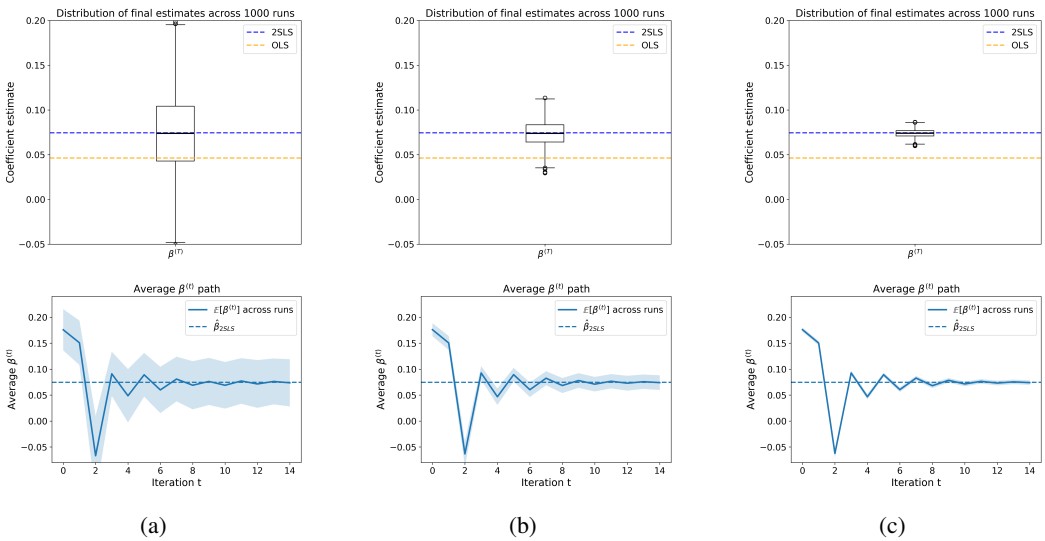

Figure 12: Experimental results on the Card dataset with $T = 15$. Each column shows the boxplot of final estimates (top) and learning path (bottom). (a)$\rho_1 = \rho_2 = 0.1$. (b)$\rho_1 = \rho_2 = 1$. (c)$\rho_1 = \rho_2 = 10$. The shaded area in the learning path represents the standard error.

