# OpenReview forum: "Differentially Private Two-Stage Gradient Descent for Instrumental Variable Regression"
_ICLR.cc/2026/Conference — ICLR 2026 Poster_

### Official Review · Reviewer_Cx28 · 2025-10-27

**Soundness:** 3
**Presentation:** 3
**Contribution:** 3
**Rating:** 4
**Confidence:** 4

**Summary:**

This paper addresses the problem of instrumental variable (IV) regression under differential privacy constraints. The authors propose  a noisy two-stage gradient-descent algorithm that injects calibrated Gaussian noise and uses per-sample gradient clipping to obtain $\rho$-zero-concentrated differential privacy (zCDP). They also provide a formal privacy accountant and derive finite-sample convergence bounds that explicitly characterize the trade-offs among privacy budgets $(\rho_1,\rho_2)$, sample size $n$, problem dimensions $p,q$, and the number of iterations $T$. The authors validate the theory with synthetic simulations and a real-data application.

**Strengths:**

1. This paper addresses a compelling problem that is likely to interest the statistics and econometric community, particularly those focused on IV regression and privacy.
2. This paper gives a clean privacy accountant and a detailed non-asymptotic utility bound.
3. This paper is well written and clearly structured.

**Weaknesses:**

1. The method and proofs are limited to the linear IV model. It may be more interesting to consider nonlinear IV or machine learning-based IV methods.
2. Even under the non-private setting, the error rate has an additional $\sqrt{p}$ factor compared to that of the z2SLS estimator. This worsened dimension dependence could be limiting in moderate-to-high-dimensional problems.
3. The necessity of privacy protection in IV method should be more clarified, it is more likely a combination of two methods in the present version.

**Questions:**

1. Do you see a path to extend your framework and privacy analysis to nonlinear IV methods? What are the main technical obstacles?
2. Can you provide more intuition about the source of the extra $\sqrt{p}$ factor compared to 2SLS, and whether an improved analysis or algorithmic modification could remove or reduce this gap?
3. In the utility analysis, it seems that the authors do not explicitly incorporate gradient clipping into the iterative update equations. The clipping thresholds only appear in the gradient sensitivity, which determines the noise scale. Could the authors explicitly include the gradient clipping step in the iteration and provide a corresponding convergence analysis? Furthermore, how should practitioners choose the clipping thresholds in practice, and how sensitive is the empirical performance to these choices?
4. How should the step sizes be selected in practice? Have the authors conducted any sensitivity analysis to evaluate how the choice of step sizes affects the empirical performance and convergence?
5. The utility analysis requires a condition on $T$ to keep noise scales manageable. In practice, does early stopping based on a private validation objective suffice? Could you provide an algorithmic prescription to choose $T$ that balances convergence and privacy noise?
6. In the real data analysis, both the endogenous dimension $p$ and the instrumental dimension $q$ are equal to 1. Are there any real datasets with higher-dimensional instruments or regressors where the proposed method could be demonstrated? It would be interesting to see how the algorithm performs in such multi-dimensional settings.

---

> ### Author Response · Authors · 2025-11-20
>
> We sincerely thank the reviewer for your valuable comments on our paper. We appreciate the opportunity to address your concerns.
>
> **W1/Q1. The method and proofs are limited to the linear IV model. It may be more interesting to consider nonlinear IV or machine learning-based IV methods. Do you see a path to extend your framework and privacy analysis to nonlinear IV methods? What are the main technical obstacles?**
>
> We agree that many real-world problems exhibit inherent non-linearity. However, we also emphasize that it is common in applied work to assume linearity between **x**  and **y**, largely due to the interpretability reasons. When non-linear effects are suspected, practitioners often apply basis expansions (e.g., polynomial features, splines, or other engineered transformations) to map the original features into a higher-dimensional space where a linear IV regression model remains applicable. Our analysis still holds under such cases.
>
> Extending our framework to fully non-linear IV methods (e.g. deep learning-based) is indeed an interesting direction for future work, but it would require new assumptions and techniques to handle the complexities introduced by non-linearity and the associated privacy analysis, which are beyond the scope of the current paper.
>
> **W2/Q2. Even under the non-private setting, the error rate has an additional $\sqrt{p}$ factor compared to that of the z2SLS estimator. This worsened dimension dependence could be limiting in moderate-to-high-dimensional problems. Can you provide more intuition about the source of the extra $\sqrt{p}$ factor compared to 2SLS, and whether an improved analysis or algorithmic modification could remove or reduce this gap?**
>
> Indeed, even in the absence of privacy noise and as $T\to\infty$, there remains a performance gap between the gradient-descent estimator $\boldsymbol{\beta}^{(T)}$ and the closed-form 2SLS estimator $\hat{\boldsymbol{\beta}}$. As an intuitive explanation, the key insight is that the closed-form 2SLS estimator solves both stages exactly using sample moments, whereas the gradient-descent procedure has to approximate the second-stage moment condition. In the population, the optimality condition for 2SLS ensures $\mathbb{E}[\mathbf{z}(\mathbf{y}-\mathbf{x}^\top\boldsymbol{\beta}^\star)]=0$. However, in the finite-sample gradient-descent iteration, the update direction will eventually depend on $\frac{1}{n}\mathbf{Z}^\top(\mathbf{Y}-\mathbf{Z}\hat{\boldsymbol{\Theta}}\hat{\boldsymbol{\beta}}),$ which involves the empirical residual $\mathbf{r}:=\mathbf{Y}-\mathbf{Z}\hat{\boldsymbol{\Theta}}\hat{\boldsymbol{\beta}}$.
>
>
> Specifically, in our proof of Theorem 1, we have the error decomposition given in equation (29). The additional factor in the convergence rate arises from the slower decay of the residual term $\mathbf{r}:=\mathbf{Z}\hat{\boldsymbol{\Theta}}\hat{\boldsymbol{\beta}}-\mathbf{Y}$, which first appears in the third term of the last line of equation (12). In particular, the quantity $\|\tfrac{1}{n}\mathbf{Z}^\top\mathbf{r}\|$ converges at order $\tfrac{\sqrt{pq}(\tau+\log(pq))}{\sqrt{n}}$ (see Lemma D.8), which is slower than the convergence rate of the closed form 2SLS estimator $\hat{\boldsymbol{\beta}}$ (see Lemma D.7). Our experiments in Appendix G.3 justifies the existence of this gap.
>
> Therefore, we believe that this performance gap persists regardless of refinements in the analysis, and is an inherent limitation of gradient-based approximations to 2SLS. Nonetheless, the gradient-descent formulation enables rigorous sensitivity control via gradient clipping, which in turn allows us to establish non-asymptotic  utility guarantees—something that is difficult to obtain for the closed-form estimator.

---

> ### Author Response · Authors · 2025-11-20
>
> **W3. The necessity of privacy protection in IV method should be more clarified, it is more likely a combination of two methods in the present version.**
>
> Instrumental variable (IV) regression is a core method in econometrics and the social sciences for estimating causal effects when endogeneity is present. In many applications, the underlying data contain sensitive personal attributes such as income, health status, or educational records. Applying IV regression in a differentially private manner is therefore essential to protect individual-level information while still enabling reliable causal inference.
>
> Our contribution lies in integrating IV regression into the differential privacy framework, addressing the joint challenges of endogeneity and privacy protection. This integration is not a simple combination of two existing techniques; rather, it fills an important methodological gap for privacy-preserving causal analysis in sensitive data environments. Adapting DP mechanisms to IV regression is technically challenging: the 2SLS estimator depends on nested matrix multiplications and inversions, making its sensitivity difficult to characterize. To overcome this, we design a two-stage gradient-descent-based algorithm that allows noise to be injected in a principled, iteration-level manner. This structure enables us to rigorously control sensitivity, calibrate noise, and establish non-asymptotic utility guarantees.
>
> By developing differentially private methods for IV regression, we make it possible for researchers to perform robust causal analyses without compromising the confidentiality of individuals in the dataset. We will expand the discussion in the introduction to more clearly articulate the practical motivation for privacy protection in IV regression settings and the significance of our contribution.

---

> ### Author Response · Authors · 2025-11-20
>
> **Q3. In the utility analysis, it seems that the authors do not explicitly incorporate gradient clipping into the iterative update equations. The clipping thresholds only appear in the gradient sensitivity, which determines the noise scale. Could the authors explicitly include the gradient clipping step in the iteration and provide a corresponding convergence analysis? Furthermore, how should practitioners choose the clipping thresholds in practice, and how sensitive is the empirical performance to these choices?**
>
> Our utility analysis follows the standard framework in the DP-SGD literature, where per-sample gradients are first clipped and then aggregated before adding Gaussian noise. The clipping thresholds $\gamma_1$ and $\gamma_2$ directly control the sensitivity of the gradients, which in turn determines the noise scale required to achieve a target privacy level. In our implementation, we set $\gamma_1=\gamma_2=c_0(\sqrt{q}+\sqrt{\tau+\log(nT)})^{2}$ for some constant $c_0>0$, ensuring that with high probability, the gradients remain within the clipping bounds across the iterations. This choice is motivated by analytical tractability and is consistent with prior work on DP linear regression (e.g., Brown et al., 2024). Reducing the clipping threshold induces more frequent clipping, effectively shrinking the update steps, while increasing the threshold enlarges the sensitivity and thus the noise magnitude, ultimately degrading utility. We conducted additional experiments to empirically study the effect of clipping thresholds in Appendix G.2.
>
> **Q4. How should the step sizes be selected in practice? Have the authors conducted any sensitivity analysis to evaluate how the choice of step sizes affects the empirical performance and convergence?**
>
> We thank the reviewer for raising this important practical consideration. We have added Remark 3.3 to discuss the optimal choices of step sizes that minimize contraction rate, and we have included corresponding empirical results in Appendix G.1 to further support this discussion.
>
> **Q5. The utility analysis requires a condition on $T$ to keep noise scales manageable. In practice, does early stopping based on a private validation objective suffice? Could you provide an algorithmic prescription to choose $T$ that balances convergence and privacy noise?**
>
> We are not entirely sure what the reviewer meant by “early stopping based on a private validation objective.” If the comment refers to the impact of the numer of iterations $T$ on overall performance, Figure 2 already shows how the loss landscape evolves with $T$, revealing the fundamental trade-off between optimization error and accumulated privacy noise. Remark 3.7 provides a detailed analysis of this tradeoff and indicates that, in practice, $T$ should be selected sub-linear but super-logarithmic to $n$ to achieve near-optimal utility. This tradeoff is further validated by our empirical results in Figure 4.
>
> **Q6. In the real data analysis, both the endogenous dimension $p$ and the instrumental dimension $q$ are equal to 1. Are there any real datasets with higher-dimensional instruments or regressors where the proposed method could be demonstrated? It would be interesting to see how the algorithm performs in such multi-dimensional settings.**
>
> We appreciate the suggestion to experiment with higher-dimensional datasets. We added another experiment study using the Card dataset (Card, 1993), which studies the effect of education(x) on earnings(y), using proximity to colleges and parents' education levels as instruments(z). In this dataset, we construct a higher-dimensional instrument set by including multiple indicators related to college proximity and parental education, resulting in $q=4$ instruments. The endogenous regressor (years of education) remains one-dimensional ($p=1$). We have included the reulst of this experiment in Appendix G.5.
>
> Our simulation study in Section 4.1 also evaluates the algorithm's behavior when both p and q are moderately large.
>
> **References**
>
> Gavin Brown, Krishnamurthy Dvijotham, Georgina Evans, Daogao Liu, Adam Smith, and
> Abhradeep Thakurta. Private gradient descent for linear regression: Tighter error bounds and
> instance-specific uncertainty estimation. arXiv preprint arXiv:2402.13531, 2024.
>
> David Card. Using geographic variation in college proximity to estimate the return to schooling.
> NBER Working Papers 4483, National Bureau of Economic Research, Inc, 1993.

---

> ### Comment · Reviewer_Cx28 · 2025-11-26
> **Response to the authors**
>
> Thank you for the feedback. After reviewing the comments, I think most of my questions and concerns are addressed. I suggest the author kindly revise their paper and add more discussion on their contributions and limitations of this work.  I have increased my score to 6.

---

### Official Review · Reviewer_Gj3i · 2025-10-29

**Soundness:** 4
**Presentation:** 3
**Contribution:** 3
**Rating:** 8
**Confidence:** 4

**Summary:**

This submission presents a differentially private algorithm for instrumental variable regression, a central tool in causal inference. The standard non-private algorithm for this task involves solving two least-squares problems. The algorithm presented here is a two-stage version of DP gradient descent. They give a theoretical analysis of the algorithm's convergence (under the assumption that the data arise from a well-specified model with subgaussian noise). For appropriate hyperparameter settings, as we add get more samples the error from privacy becomes lower than the error from sampling.

They also present experiments on synthetic and real data, and the method seems quite practical.

**Strengths:**

Instrumental variable regression is an important task and extremely worthy of study under privacy. While there is a lot of work on private ordinary least squares, I'm surprised there's none on this topic.

The method presented here is a clear contribution, both theoretically and practically. I was unable to verify the proofs, but the results are very plausible.

**Weaknesses:**

The main convergence proof is quite long and, to my taste, comes with insufficient exposition. It was hard for me to understand which parts of the analysis are new.

There are no approaches that explicitly tackle DP IVaR, but there are well-known techniques that directly yield results. In particular, the "subsample and aggregate" framework can usually make use of any non-private estimator with meaningful concentration guarantees. For the task here, combining the Algorithm 5.1 of "FriendlyCore" [1] with the submission's Lemma D.7 yields a zCDP estimator with $\ell_2$ error roughly $\sqrt{pq/n}$, I think. This matches the non-private error and, in some regimes, might be better than Theorem 3.1's $\sqrt{p}q^{3/2}/n$ term.

[1] https://arxiv.org/abs/2110.10132

**Questions:**

Can you formally combine Lemma D.7 with the FriendlyCore approach and compare the resulting theorem with your Theorem 3.1?

Can you briefly describe what (if any) new technical challenges your main convergence proof must overcome?

Can you fix the following minor issues with your background discussion?
- $(\varepsilon,\delta)$-DP algorithms satisfy basic composition, where the privacy terms accumulate linearly, but also *advanced* composition [see 2]. Here the asymptotics match those of zCDP, but the latter is usually more practical.
- You mention that gradient-perturbation methods can avoid spectrum-based blowups, but (i) private first-order methods still suffer on ill-conditioned data [3,4] and (ii) there are techniques for privately estimating $X^T X$ that avoid these blow-ups [5,6], although when used within a sufficient-statistics approach for regression they may incur higher sample complexity than necessary [see 7 for discussion].

[2] https://dpcourse.github.io/2025-spring/lecnotes-web/DP-S25-notes-lec-10-adv-composition.pdf

[3] https://arxiv.org/abs/2207.04686

[4] https://arxiv.org/abs/2301.13273

[5] https://arxiv.org/abs/1805.00216

[6] https://arxiv.org/abs/2301.12250

[7] https://arxiv.org/abs/2404.15409

---

> ### Author Response · Authors · 2025-11-21
>
> We sincerely thank the reviewer for your valuable comments on our paper. We appreciate the opportunity to address your concerns.
>
> **W1/Q2. The main convergence proof is quite long and, to my taste, comes with insufficient exposition. It was hard for me to understand which parts of the analysis are new. Can you briefly describe what (if any) new technical challenges your main convergence proof must overcome?**
>
> We agree that Appendix C and D are quite long. We have added a high-level overview at the beginning of Appendix C that highlights the structure of our proof and the specific difficulties that do not arise in standard DP-SGD analyses.
>
> In particular, our convergence proof must address several technical challenges:
>
>   - Two-stage coupling: We control a pair of noisy recursions $\boldsymbol{\Theta}^{(t)}, \boldsymbol{\beta}^{(t)}$, $t=1,\ldots,T$ where the second-stage gradient at time $t$ depends on the noisy first stage iterate $\boldsymbol{\Theta}^{(t)}$. This coupling does not appear in standard DP-SGD analyses.
>
>  - Noise propagation: Since we add Gaussian noise at every step and stage, the privacy accountant composes over all $2T$ iterations. To obtain a meaningful error bound for the final iterate $\boldsymbol{\beta}^{(T)}$, we need to separate the contributions of optimization, privacy, and sampling error. This decomposition gives rise to the three terms in Theorem 3.1 and explains the tradeoff pattern shown in Figure 2 and our experiments.
>
> - Non-asymptotic rate: The analysis retains explicit dependence on all major parameters: sample size $n$, instrument dimension $q$, regressor dimension $p$, privacy budgets $(\rho_1,\rho_2)$, and confidence parameter $\tau$.
>
> **W2/Q1. There are no approaches that explicitly tackle DP IVaR, but there are well-known techniques that directly yield results. In particular, the "subsample and aggregate" framework can usually make use of any non-private estimator with meaningful concentration guarantees. For the task here, combining the Algorithm 5.1 of "FriendlyCore" [1] with the submission's Lemma D. 7 yields a zCDP estimator with $\ell_2$ error roughly $\sqrt{p q / n}$, I think. This matches the non-private error and, in some regimes, might be better than Theorem 3.1 's $\sqrt{p} q^{3 / 2} / n$ term. Can you formally combine Lemma D. 7 with the FriendlyCore approach and compare the resulting theorem with your Theorem 3.1?**
>
> We appreciate the reviewer for pointing out the FriendlyCore method as a potential baseline.
> In our revision, we have included a comparison between FriendlyCore and our proposed DP-2S-GD algorithm in Appendix F.2. In summary, we found out the given the privacy budget $\rho$, the error rate of FriendlyCore adapted to 2SLS is of order $\mathcal{O}\left(L\frac{q^2+q\sqrt{\tau}}{n\sqrt{\rho}}+\frac{\sqrt{pq}(\tau+\log(pq))}{\sqrt{n}}\right)$, where $L$ is the Lipschitz constant correspond to mapping $f:\text{vec}(\frac{\mathbf{Z}^\top\mathbf{Z}}{n},\frac{\mathbf{Z}^\top\mathbf{X}}{n},\frac{\mathbf{Z}^\top\mathbf{Y}}{n})\mapsto\hat{\boldsymbol{\beta}}_{2SLS}$. We note that this order roughly matches our non-aymptotic error bound in Theorem 3.1, up to constant factors and the dependence on $L$.
>
>
> **Q3. Can you fix the following minor issues with your background discussion?**
>    - **$(\varepsilon, \delta)$-DP algorithms satisfy basic composition, where the privacy terms accumulate linearly, but also advanced composition [see 2]. Here the asymptotics match those of zCDP, but the latter is usually more practical.**
>    - **You mention that gradient-perturbation methods can avoid spectrum-based blowups, but (i) private first-order methods still suffer on ill-conditioned data [3,4] and (ii) there are techniques for privately estimating $X^T X$ that avoid these blow-ups $[5,6]$, although when used within a sufficient-statistics approach for regression they may incur higher sample complexity than necessary [see 7 for discussion].**
>
> We thank the reviewer for pointing out these issues. In the revision we have updated Sections 1.1 and 2.1 to discuss these issues
>
>    - We explicitly state that $(\varepsilon, \delta)$-DP mechanisms also admits an advanced composition that asymptotically matches zCDP, and clarify that we adopt zCDP purely for technical convenience.
>
>    -  We acknowledge that (i)there exists DP techniques for estimating gram matrices that mitigate spectrum-dependent blow-ups, although these approaches typically require more samples when used within a sufficient-statistics–based regression pipeline, and (ii)private first-order gradient methods still depend heavily on condition number.

---

> > ### Comment · Reviewer_Gj3i · 2025-11-24
> >
> > Thank you for your response. I retain my positive impression of the paper.

---

### Official Review · Reviewer_sNPP · 2025-11-03

**Soundness:** 3
**Presentation:** 3
**Contribution:** 3
**Rating:** 4
**Confidence:** 3

**Summary:**

This paper studies differentially private instrumental variables (IV) regression and proposes **DP-2S-GD**, a two-stage gradient-descent algorithm that enforces Renyi differential privacy through clipping and Gaussian noise injection (in a usual DP-GD style but with two different sets of parameters). The method provides end-to-end privacy guarantees for the classical two-stage least squares (2SLS) setting for IVaR. The authors prove a non-asymptotic convergence bound that decomposes optimization error, sampling error, and noise due to privacy. They also characterize a privacy-iteration trade-off and show how privacy budgets affect convergence behavior. Experiments on synthetic data and the Angrist–Evans dataset support the theoretical predictions, demonstrating that the estimator remains stable and competitive under useful privacy levels.

**Strengths:**

The paper fills a clear gap by proposing a differentially private version of instrumental variable regression. The formulation is simple and technically consistent, combining a two-stage gradient procedure with zCDP-based privacy guarantees. The analysis provides non-asymptotic convergence bounds for $\beta$ (the regression parameter of interest) that decompose statistical, optimization, and privacy errors, making the results partially interpretable. The privacy proof is correct and applies zCDP effectively to handle iterative updates. The experiments, though limited, support the theoretical claims and clearly show the predicted privacy–accuracy and overfitting behavior (from Figure 2). The writing is clear, notation consistent, and the argument proceeds logically from setup to results.

**Weaknesses:**

**Untuned step-sizes and incomplete convergence analysis (major weakness)**

The paper provides only upper bounds for the step sizes \(\eta\) and \(\alpha\) ). These conditions ensure convergence in principle but leave the question of the optimal step-sizes open. As a result, the convergence theorem lacks interpretability: there are too many definitions, and not all of them depend on just problem dependent parameters. This limitation gives the impression that the analysis has not been fully worked out. A sharper result would express convergence explicitly in terms of well-defined problem constants and privacy noise scales. Since this is the main contribution of the paper, and the algorithm itself is not very surprising, I am inclined to reject the paper in its current form. I would like to see a fully worked out, non-asymptotic convergence guarantee before I can recommed acceptance. If it is hard to tune all the parameters for a high probability guarantee, can the authors atleast provide one for an in expectation result?

**Limited baselines**

The experiments lack comparison with simple private 2SLS variants, such as perturbing sufficient statistics or applying output perturbation (which were indeed discussed in the paper while providing a survey of related methods). Including such baselines would help quantify the benefit of the proposed algorithm beyond intuitive reasoning.

**Non-private performance gap**

Even without privacy, the estimator remains slower than classical 2SLS, introducing an additional factor in the rate. The paper acknowledges this but does not explain what causes this.

**Questions:**

1. Can the authors report the privacy levels used in all experiments in terms of the standard $(\epsilon, \delta)$-DP, in addition to $\rho$-zCDP, so that readers can directly compare privacy strengths with prior work?

2. Can the step sizes be tuned or derived in a way that yields an explicit non-asymptotic convergence guarantee depending only on problem parameters (e.g., Lipschitz constants or curvature), rather than through broad stability bounds?

3. Would it be possible to consider a stochastic variant of the algorithm, where each optimization step uses a new sample so that $T$ and $n$ are coupled? This might lead to a simpler and more interpretable result, since the trade-offs between optimization and generalization would coincide. It could also remove the “overfitting” behavior seen in Figure 2, where the error decreases and then rises as $T$ grows. This is not a major issue, but it could better bring other aspects such as the utility-privacy trade-off on the centerstage.

4. The motivation for the IVaR setup remains somewhat unclear. Can the authors give concrete examples of real situations where one can actually observe or construct the full dataset $(z, x, y)$ required for the two-stage regression? In settings where the true causal graph is unknown, how can one justify the validity of Assumptions 1 and 2? Some discussion or examples of practical data sources would help clarify this premise.

---

> ### Author Response · Authors · 2025-11-20
>
> We sincerely thank the reviewer for your valuable comments on our paper. We appreciate the opportunity to address your concerns.
>
> **W1. Untuned step-sizes and incomplete convergence analysis (major weakness)**
>
> We would like to emphasize that our convergence guarantee (Theorem 3.1) applies to any fixed step sizes satisfying condition (3), and the resulting bound is already a finite-sample, non-asymptotic guarantee up to a constant factor. Moreover, one can verify that the contraction rate $\kappa(\tau)$ is minimized when $\eta=\frac{2}{(1+\delta(\tau))^2+(1-\delta(\tau))^2},$ and $ \alpha = \frac{2}{\bar{\gamma}(\tau)+\underline{\gamma}(\tau)}.$ These choices produce an approximately optimal contraction bound. We have added Remark 3.3 and included additional empirical results in Appendix G.1.
>
> Regarding the presentation of the bounds, we absorb the problem-specific constants $\boldsymbol{\beta}, \boldsymbol{\Theta}, \sigma_z, \sigma_1, \sigma_2$ into the overall scaling factor, as carrying these terms explicitly would make the final expression unnecessarily cumbersome. Instead, we emphasize the dependence on $(n, p,q,\tau,T,\rho_1,\rho_2)$, which are the main parameters of interest to practitioners. We acknowledge that the definitions of $\delta(\tau), \bar{\gamma}(\tau), \underline{\gamma}(\tau)$ are somewhat involved, but they are necessary to capture the randomness in data and establish the non-asymptotic bound. We have added several explanations in Theorem 3.1 to clarify the roles of these quantities.
>
> **W2. Limited baselines**
>
> We highlight that our work is, to the best of our knowledge, the first to develop a differentially private algorithm specifically designed for IV regression. Consequently, there are no existing DP counterparts of 2SLS that can serve as direct baselines. In particular, adapting sufficient statistics perturbation method into 2SLS itself would be a non-trivial work, as the closed-form solution of 2SLS estimator involves several nested matrix multiplications and inversions (see equation (45)), making it difficult to obtain tractable sensitivity bounds or derive clean utility guarantees. A similar challenge arises for output perturbation, since the global sensitivity of the 2SLS estimator does not admit a simple form and its stability properties are significantly more complex than in standard OLS. These technical obstacles further motivate our gradient-based design, which enables a much more transparent privacy and utility analysis.
>
> **W3. Non-private performance gap**
>
> Indeed, even in the absence of privacy noise and as $T\to\infty$, there remains a performance gap between the gradient-descent estimator $\boldsymbol{\beta}^{(T)}$ and the closed-form 2SLS estimator $\hat{\boldsymbol{\beta}}$. As an intuitive explanation, the key insight is that the closed-form 2SLS estimator solves both stages exactly using sample moments, whereas the gradient-descent procedure has to approximate the second-stage moment condition. In the population, the optimality condition for 2SLS ensures $\mathbb{E}[\mathbf{z}(\mathbf{y}-\mathbf{x}^\top\boldsymbol{\beta}^\star)]=0$. However, in the finite-sample gradient-descent iteration, the update direction will eventually depend on $\frac{1}{n}\mathbf{Z}^\top(\mathbf{Y}-\mathbf{Z}\hat{\boldsymbol{\Theta}}\hat{\boldsymbol{\beta}}),$ which involves the empirical residual $\mathbf{r}:=\mathbf{Y}-\mathbf{Z}\hat{\boldsymbol{\Theta}}\hat{\boldsymbol{\beta}}$.
>
>
> Specifically, in our proof of Theorem 1, we have the error decomposition given in equation (29). The additional factor in the convergence rate arises from the slower decay of the residual term $\mathbf{r}:=\mathbf{Z}\hat{\boldsymbol{\Theta}}\hat{\boldsymbol{\beta}}-\mathbf{Y}$, which first appears in the third term of the last line of equation (12). In particular, the quantity $\|\tfrac{1}{n}\mathbf{Z}^\top\mathbf{r}\|$ converges at order $\tfrac{\sqrt{pq}(\tau+\log(pq))}{\sqrt{n}}$ (see Lemma D.8), which is slower than the convergence rate of the closed form 2SLS estimator $\hat{\boldsymbol{\beta}}$ (see Lemma D.7). Our experiments in Appendix G.3 justifies the existence of this gap.
>
> Therefore, we believe that this performance gap persists regardless of refinements in the analysis, and is an inherent limitation of gradient-based approximations to 2SLS. Nonetheless, the gradient-descent formulation enables rigorous sensitivity control via gradient clipping, which in turn allows us to establish non-asymptotic  utility guarantees—something that is difficult to obtain for the closed-form estimator.

---

> ### Author Response · Authors · 2025-11-20
>
> **Q1. Can the authors report the privacy levels used in all experiments in terms of the standard
> -DP, in addition to
> -zCDP, so that readers can directly compare privacy strengths with prior work?**
>
> Thank you for the suggestion. In our revised manuscript, we have reported the corresponding $(\epsilon,\delta)$ values for each $\rho\in\{0.1,1,10\}$ used in our experiments, representing strong, medium, and weak privacy levels. Specifically, for each $\rho$ we compute $\epsilon$ via the conversion formula $\epsilon = \rho + 2\sqrt{\rho \log(1/\delta)}$ with a fixed $\delta=10^{-5}$.
>
> **Q2. Can the step sizes be tuned or derived in a way that yields an explicit non-asymptotic convergence guarantee depending only on problem parameters (e.g., Lipschitz constants or curvature), rather than through broad stability bounds?**
>
> We would like to reiterate that our convergence guarantee is already a **non-asymptotic bound** that holds for any fixed step sizes satisfying condition (3), up to a scaling factor. These conditions arise naturally from the classical requirement $(0<\text{step size}<\frac{2}{L})$, where L denotes the appropriate Lipschitz constant of the gradient. At the population level, the Lipschitz constant for first stage regression is $L_1=\lambda_{\max}(\mathbb{E}[\mathbf{zz}^\top])=1$ (according to assumption 2), and the second stage is $L_2=\lambda_{\max}(\mathbb{E}[\boldsymbol{\Theta}^\top\mathbf{zz}^\top\boldsymbol{\Theta}])=\lambda_{\max}(\boldsymbol{\Theta}^\top\mathbb{E}[\mathbf{zz}^\top]\boldsymbol{\Theta})=\|\boldsymbol{\Theta}\|^2$. However, when working with finite samples, these population quantities must be replaced by their empirical counterparts, requiring high-probability bounds on $L_1$ and $L_2$. This leads directly to our sample-level step-size conditions in (3). For a detailed discussion of how these population- and sample-level conditions correspond, please refer to Remark 3.2.
>
> **Q3. Would it be possible to consider a stochastic variant of the algorithm, where each optimization step uses a new sample so that $T$ and $n$ are coupled?  This could potentially mitigate the "overfitting" behavior in Figure 2, where the error decreases and then rises as $T$ grows.**
>
> We appreciate the suggestion of considering a stochastic variant of the algorithm. While this is an interesting direction for future work, we expect some challenges. First, the two-stage nature of the IV regression problem implies that stochasticity in the first stage would propagate into the second stage in a more intricate manner, especially when fresh samples are used at each iteration, which typically leads to larger variance. Second, the tail behavior in Figure 2 arises from the accumulation of privacy noises over multiple iterations (see Lemma D.4), which would still be present in a stochastic setting. Addressing these challenges would require more careful design to balance the privacy-utility trade-off and is beyond the scope of the current paper.
>
> **Q4. The motivation for the IVaR setup and concrete examples. How can one justify the validity of Assumptions 1 and 2?**
>
> We thank the reviewer for raising this important point. Instrumental variable regression is a central tool in econometrics and the social sciences for estimating causal effects in the presence of endogeneity. Endogeneity is ubiquitous in real-world applications, which arises whenever there is a confounder that affects both the regressor and the outcome. These confounders are often unobserved or difficult to measure. In such cases, standard regression methods yield biased estimates, and IV techniques are specifically designed to correct for this issue.
>
> Regarding the validity of assumptions 1 and 2, we first emphasize that Assumption 1 is standard assumptions in IV literature. In general, instrument relevance and instrument exogeneity cannot be verified purely from statistical tests, and typically rely on economic or causal reasoning supported by diagnostic checks. For assumption 2, (i) can be easily satisfied in practice through standardization of instruments, which is also applied as a pre-processing step in our experiment, and (ii) imposes sub-Gaussian tails, a mild condition that underpins most modern non-asymptotic analyses, which is appropriate when the noise terms are centered  and not excessively heavy-tailed.
>
> We now include two classic empirical IV studies in our revised menuscript. The first example (Angrist and Evans, 1998) analyzes the causal effect of children bearing (x) on mothers' labor supply (y), using the indicator for whether the first two children are of the same sex (z) as an instrument. The second example(Card 1993) examines the causal effect of years of education (x) on earnings (y), using proximity and to colleges and parents' educational levels (z) as instruments for years of schooling. The details are attached in Appendix G.4 and G.5.

---

### Official Review · Reviewer_hM7v · 2025-11-10

**Soundness:** 3
**Presentation:** 3
**Contribution:** 3
**Rating:** 6
**Confidence:** 2

**Summary:**

The paper proposes a differentially private estimator for (a specific formulation of) instrumental-variable regression. The paper suggests noisy gradient descent, in which the key parameters $\Theta$ (a matrix) and $\beta$ (a vector) are updated at each round using a noisy gradient measurement. THe paper gives natural conditions under which this algorithm provably converges (and also provides bounds on its error in estimating the vector $\beta$).

The algorithm's convergence is also evaluated empirically on synthetic and real data.

**Strengths:**

* The paper addresses a widely-used statistical method
* The paper appears technically sound. Though I did not check the proofs carefully, the authors present and discuss their results clearly, and the results and proofs seem plausible.

**Weaknesses:**

* The final bounds were hard to interpret, partly because they necessarily involves quite a few different parameters. It would be helpful to provide some baselines, both in the form of nonprivate methods, and in the form of alternate approaches to designing DP algorithms. For example, a few of the questions I couldn't answer as a reader were:

    * What level of noise is acceptable? That is, how are the results of DP IVaR supposed to be used, and when does the noisy version meet the needs of application? Even if the answer is context dependent, providing one context in which the overall usefulness of the answer can be evaluated would be helpful.

    * How does the proposed method compare to a more straightforward perturbation of the sufficient statistics for the model (presumably the covariance matrix of the concatenation $(z,x,y)$).

    * When is DP IVaR more useful than DP OLS on its own? As far as I know, IVaR and OLS are normally used in tandem. However, presumably the noise levels have to be low enough to for them to be meaningfully compared. When is that possible? In what parameter ranges do the results of the paper imply the validity of such a comparison?

* It would be good to compare the bounds in this paper to those implied by other work on DP nonconvex optimization.

**Questions:**

* Privacy notions: Remark 3.8 states that, when $\Theta^{(t)}$ is not released, "setting $\rho=\infty$ implies that no noise $\Xi^{(t)}$ needs to be injected in the first state, and we can simply return $\{\beta^{(t)}\}$ under privacy budget $\rho_2$." I don't see why that's true---the hidden $\\Theta^{(t)}$ values encode information about the sensitive data and propagate it throughout the computation. At the very least the statement deserves a more careful formulation and explanation. (Also, why would $\beta$ be useful without $\Theta$? The latter seems necessary to interpret the former as a predictor of $y$)

* What are the relevant facts about the standard 2S-LS estimator for this paper? It is mentionedand compared to the 2S-GD estimator in Remark 3.7, but the algorithm and its convergence rate are not precisely described, which makes the comparison hard to follow.

---

> ### Author Response · Authors · 2025-11-21
>
> We sincerely thank the reviewer for your valuable comments on our paper. We appreciate the opportunity to address your concerns.
>
> **W1.1. What level of noise is acceptable? That is, how are the results of DP IVaR supposed to be used, and when does the noisy version meet the needs of application? Even if the answer is context dependent, providing one context in which the overall usefulness of the answer can be evaluated would be helpful.**
>
> The goal of our algorithm is to provide a differentially private estimator for the IV regression problem that achieves a utility comparable to the standard non-private 2SLS estimator while satisfying rigorous privacy guarantees. In our algorithm, the user only need to provide the dataset $(\mathbf{Z},\mathbf{X},\mathbf{Y})$, specify the privacy budgets $\rho_1,\rho_2$ and number of iterations $T$. Then the levels of noise $\lambda_1,\lambda_2$ are determined by equation (4), which is set to ensure the privacy level $\rho:=\rho_1+\rho_2$ can be achieved.
>
> As a practical guideline, $\rho=0.1$ can be considered as strong privacy, $\rho=1$ is considered as moderate privacy, and $\rho=10$ is considered as weak privacy.  We have added this guideline in Section 4 of the revised manuscript.
>
>
> **W1.2. How does the proposed method compare to a more straightforward perturbation of the sufficient statistics for the model (presumably the covariance matrix of the concatenation $(z, x, y)$).**
>
> Although sufficient statistics perturbation is conceptually straightforward, implementing it for 2SLS is technically challenging due to the nested matrix operations involved in the estimator. This complexity makes it difficult to derive tight sensitivity bounds and utility guarantees. In contrast, our gradient-based approach allows for more transparent sensitivity control via gradient clipping, enabling rigorous privacy and utility analysis.
>
> Specifically, our method offer the following advantages: (i) Stability under weak instruments: When $Z^{\top} X$ is ill-conditioned, adding Gaussian noise directly to this matrix can dramatically amplify variance or even make the inversion unstable. DP-2S-GD never inverts a noisy Gram matrix; instead, it works with per-sample gradients and hence is more stable in weak-instrument settings. (ii) Stage-wise privacy and interpretability: Our analysis separates the contribution of each stage through ($\rho_1, \rho_2$) and yields an error decomposition where first-stage and second-stage privacy noise appear explicitly. Perturbing sufficient statistics mixes these sources of error, making it harder to reason about how privacy budget should be allocated between stages.
>
> Although direct perturbation of 2SLS sufficient statistics is difficult to analyze, other sufficient-statistics-based frameworks can produce a private 2SLS estimator. As Reviewer Gj3i suggests, the FriendlyCore method (Tsfadia et al., 2022) provides a computationally feasible way to get the private 2SLS estimator using subsample-and-aggregate approach. In the updated paper, we provide quantitative comparison between FriendlyCore and our proposed approach in Appendix F.2. Additional details can be found in our response to Reviewer Gj3i's first question.
>
>
> **W1.3. When is DP IVaR more useful than DP OLS on its own? As far as I know, IVaR and OLS are normally used in tandem. However, presumably the noise levels have to be low enough to for them to be meaningfully compared. When is that possible? In what parameter ranges do the results of the paper imply the validity of such a comparison?**
>
> The key distinction between IV regression and OLS is that OLS requires the exogeneity condition $\mathbb{E}[\varepsilon \mid x]=0$. IV regression is specifically designed for settings where this condition fails—typically due to omitted variables, measurement error, or simultaneity—leading OLS (and thus any DP-OLS variant) to be inconsistent, regardless of its noise levels.  In contrast, a DP version of IV regression remains appropriate because it targets the correct causal parameter under endogeneity. DP-IVaR is therefore the preferred approach when: (1) credible instruments are available and a standard IV design is justified (e.g., randomized encouragement, policy shocks, geographic instruments), and (2) the underlying data are sensitive, so releasing either first-stage predictions or the final treatment-effect estimate must satisfy formal differential-privacy guarantees.
>
> **W2. It would be good to compare the bounds in this paper to those implied by other work on DP nonconvex optimization.**
>
> We thank the reviewer for this suggestion. To the best of our knowledge, there is no prior work on differentially private algorithms specifically designed for IV regression, so a direct comparison is not available.
>
> As discussed in W1.2, there are some computational approaches that can be adapted to produce private 2SLS estimators. Please refer to Appendix F.2 in the revised manuscript for details.

---

> ### Author Response · Authors · 2025-11-21
>
> **Q1. Privacy notions: Remark 3.8 states that, when $\Theta^{(t)}$ is not released, "setting $\rho=\infty$ implies that no noise $\Xi^{(t)}$ needs to be injected in the first state, and we can simply return $\beta^{(t)}$ under privacy budget $\rho_2$." I don't see why that's true---the hidden $\Theta^{(t)}$ values encode information about the sensitive data and propagate it throughout the computation. At the very least the statement deserves a more careful formulation and explanation. (Also, why would $\beta$ be useful without $\Theta$ ? The latter seems necessary to interpret the former as a predictor of $y$)**
>
> We would like to clarify that the primary goal of IV regression is to obtain a consistent estimate of the causal effect of the endogenous variable $\mathbf{X}$ on the outcome $\mathbf{Y}$. The parameter of interest is $\boldsymbol{\beta}$, which quantifies this causal relationship. The first-stage parameter $\boldsymbol{\Theta}$ is an intermediate quantity used to construct the fitted values $\hat{\mathbf{X}} = \mathbf{Z}\hat{\boldsymbol{\Theta}}$, which are then employed in the second stage to estimate $\boldsymbol{\beta}$. Although $\boldsymbol{\Theta}$ describes how the instruments relate to the endogenous regressor, it is not itself the focus of inference.Therefore, in many applications, releasing $\boldsymbol{\beta}^{(t)}$ alone is sufficient. When $\boldsymbol{\Theta}^{(t)}$ are not released, the first-stage updates do not need to be privatized, allowing us to set $\rho_1=\infty$ and avoid adding noise $\boldsymbol{\Xi}^{(t)}$ in the first stage.
>
> **Q2. What are the relevant facts about the standard 2S-LS estimator for this paper? It is mentionedand compared to the 2S-GD estimator in Remark 3.7, but the algorithm and its convergence rate are not precisely described, which makes the comparison hard to follow.**
>
> We appreciate this remark. The current draft assumes familiarity with classical 2SLS and does not restate its form clearly enough. In the revised manuscript, we have added a brief description of the standard 2SLS estimator in Section 2.2 for clarity. In the first stage of 2SLS, we solve
>
> $\hat{\boldsymbol{\Theta}}=\arg \min _{\boldsymbol{\Theta}} \frac{1}{n}\|\mathbf{X}-\mathbf{Z}\boldsymbol{\Theta}\|^2=\left(\mathbf{Z}^{\top} \mathbf{Z}\right)^{-1} \mathbf{Z}^{\top} \mathbf{X},$
>
> and in the second stage we solve
>
> $\hat{\boldsymbol{\beta}}_{2SLS}=\arg \min _{\boldsymbol{\beta}} \frac{1}{n}\|\mathbf{Y}-\hat{\mathbf{X}}\boldsymbol{\beta}\|^2=(\hat{\boldsymbol{\Theta}}^\top\mathbf{Z}^\top\mathbf{Z}\hat{\boldsymbol{\Theta}})^{-1}\hat{\boldsymbol{\Theta}}^\top\mathbf{Z}^\top\mathbf{Y}.$
>
> 2SLS admits a closed-form solution, and its non-asymptotic error rate is stated in Lemma D.7.
>
> **Reference**
> Eliad Tsfadia, Edith Cohen, Haim Kaplan, Yishay Mansour, and Uri Stemmer. Friendlycore: Practical differentially private aggregation. In International Conference on Machine Learning, pp.21828–21863. PMLR, 2022

---

### Meta-Review · Area_Chair_J5jV · 2026-01-07

**Summary:**

The paper proposes a differentially private algorithm for instrumental variable regression using a two-stage gradient descent approach, providing non-asymptotic convergence bounds. It fills a significant gap in the literature by enabling privacy-preserving causal inference with provable utility guarantees.

**Reviewer Concerns:**

The authors effectively addressed the lack of baselines by comparing against FriendlyCore and clarified step-size tuning in the revision. The performance gap vs. non-private 2SLS is accepted as an inherent trade-off for gradient-based privacy control rather than a flaw.

**Reviewer Scores:**

Gj3i: Retained a strong score of 8, finding the method a clear contribution and the inclusion of the FriendlyCore baseline satisfactory.

Cx28: Raised score to 6 after the authors clarified the motivation, step-sizes, and acknowledged the linearity limitations as acceptable scope.

sNPP: Would likely raise their score to at least a 5 or 6, as their "major weakness" regarding step-size tuning and the request for baselines were directly addressed in the revision.

hM7v: Would likely maintain or slightly increase their score of 6, as the clarifications regarding noise levels and utility vs. OLS addressed their specific interpretability concerns.

---

### Decision · Program_Chairs · 2026-01-26

Accept (Poster)